# Selection rules in symmetry-broken systems by symmetries in synthetic dimensions

Matan Even Tzur [ID] [1✉], Ofer Neufeld [ID] [1,2], Eliyahu Bordo [ID] [1], Avner Fleischer [ID] [3] & Oren Cohen [ID] [1]

Selection rules are often considered a hallmark of symmetry. Here, we employ symmetry-breaking degrees of freedom as synthetic dimensions to demonstrate that symmetry-broken systems systematically exhibit a specific class of symmetries and selection rules. These selection rules constrain the scaling of a system's observables (non-perturbatively) as it transitions from symmetric to symmetry-broken. Specifically, we drive bi-elliptical high harmonic generation (HHG), and observe that the scaling of the HHG spectrum with the pump's ellipticities is constrained by selection rules corresponding to symmetries in synthetic dimensions. We then show the generality of this phenomenon by analyzing periodically-driven (Floquet) systems subject to two driving fields, tabulating the resulting synthetic symmetries for $(2+1)$D Floquet groups, and deriving the corresponding selection rules for high harmonic generation (HHG) and other phenomena. The presented class of symmetries and selection rules opens routes for ultrafast spectroscopy of phonon-polarization, spin-orbit coupling, symmetry-protected dark bands, and more.

[1] Solid State Institute and Physics Department, Technion-Israel Institute of Technology, Haifa 3200003, Israel. [2] Max Planck Institute for the Structure and Dynamics of Matter and Center for Free-Electron Laser Science, Hamburg 22761, Germany. [3] Raymond and Beverly Sackler Faculty of Exact Science, School of Chemistry and Center for Light-Matter-Interaction, Tel Aviv University, Tel-Aviv 6997801, Israel. ✉email: Matanev@campus.technion.ac.il

Selection rules are a significant consequence of symmetries, appearing throughout science. For example, point-group symmetries forbid electronic transitions in solids and molecules[1], the existence of electric and magnetic moments[2], and harmonic generation in perturbative nonlinear optics[3]. In periodically driven (Floquet[4]) systems, symmetries that involve space and time (denoted dynamical symmetries, DSs) result in selection rules for various phenomena. Examples include symmetry-induced dark states and transparency[5–7], forbidden harmonics and polarization restrictions in HHG[8–10], as well as forbidden photoionization channels[11]. Symmetries and selection rules are widely used for controlling and measuring one another. On one hand, a system's symmetry is often "engineered" to achieve a desirable selection rule[12–17]. On the other hand, in symmetry-broken systems, the selection rules are replaced with selection rule *deviations* that can be used to extract information about the broken symmetry and the symmetry-breaking perturbation. In nonlinear optics for instance, selection rule deviations are used as a background-free gauge of molecular symmetry[18], orientation[19], and chirality[20–22], electric currents[23], Berry curvature[24], topological phase transitions[25], and more[26]. However, to date, periodically driven systems with broken symmetry, and selection rule deviations, have only been analyzed ad hoc[18–21,23–27], mostly via perturbation theory (which is often limited to analyzing selection rule deviations up to 2nd order). As a result, there is limited insight into the system's full dynamical behavior.

Here we present a class of symmetries and selection rules which naturally appear in systems that exhibit broken DSs. We construct these symmetries by exploiting symmetry-breaking degrees of freedom as synthetic dimensions. We consider a general system that exhibits some DS $\hat{X}$, and show that when a perturbation breaks that DS, it imposes an alternative symmetry of the form $\hat{X} \cdot \hat{\zeta}$, where $\hat{\zeta}$ is an operation in synthetic space that acts only on the symmetry-breaking degrees of freedom (i.e., on the perturbation). We term the class of these alternative symmetries "real-synthetic symmetries" because of their composite structure. Instead of the standard selection rules from the symmetry $\hat{X}$ (which are broken), we show that the system exhibits a specific class of selection rules that restrict scaling of observables with respect to the perturbation. As an example, we focus on HHG in Floquet systems subject to two driving fields, where the DS $\hat{X}$ corresponds to the symmetry of the material sample and a 1st driving laser, and a 2nd field breaks $\hat{X}$ but imposes $\hat{X} \cdot \hat{\zeta}$ symmetry in synthetic space, which consequently constrains the scaling of the HHG spectral yield. The paper is organized as follows: we begin with a simple numerical example of real-synthetic symmetries and their corresponding selection rules. Then, we present an experimental investigation of real-synthetic symmetries in HHG driven by bi-chromatic bi-elliptical pumps. We observe selection rules that manifest as restricted scaling laws of the harmonic orders and polarization states as the system transitions from the symmetric to the symmetry broken state. Finally, we rigorously tabulate all real synthetic symmetries and their corresponding HHG selection rules in $(2 + 1)$D Floquet systems subject to two driving fields (ATI selection rules are derived in section VI of the Supplementary Information).

## Results
In order to introduce real-synthetic symmetries, we first remind the difference between HHG in symmetric and symmetry-broken systems. In HHG[28–30], high-order harmonics are emitted due to interaction between a strong pump laser and a medium. When the system (pump and medium) exhibits a DS, the harmonic spectrum exhibits selection rules in the form of forbidden harmonics and polarization restrictions. If the DS is broken, selection rule deviations (e.g., forbidden harmonics) are generated. Figure 1 illustrates examples of the scenario depicted above; with "intact" and with broken dynamical reflection symmetry. In the three examples presented in the left panel, the DS forbids even harmonic generation polarized along the $\hat{x}$ axis due to a dynamical reflection symmetry denoted by $\hat{Z}_i = \hat{\tau}_2 \cdot \hat{\sigma}_i$, where $\hat{\tau}_2$ is a half-cycle time translation and $\hat{\sigma}_i$ is a spatial reflection relative to $\hat{i}$ axis (so that $\hat{\sigma}_x$ is the operation $y \rightarrow -y$)[10]. Conversely, the dynamical reflection symmetry is broken in the systems in the right panel due to application of voltage to the sample, orientation of the molecular ensemble, or by an additional laser pulse. Consequently, the systems on the right panel emit even harmonics along the $\hat{x}$ axis– selection rule deviations—as shown in the schematic spectrum. While all the perturbations break the same dynamical reflection symmetry $\hat{Z}_i$, each one imposes a different real-synthetic symmetry $\hat{Z} \cdot \hat{\zeta}$, where $\hat{\zeta}$ acts on the perturbation degrees of freedom in the Hamiltonian: e.g., the applied voltage direction, the degree of orientation of the molecular ensemble, or the polarization direction of the symmetry breaking laser pulse (marked red in Fig. 1). As will be shown below, symmetry operations of the form $\hat{X} \cdot \hat{\zeta}$ impose selection rules that constrain the scaling of the emission in a non-perturbative manner, valid to all orders in the perturbation strength (without invoking perturbation theory at any stage). Below we focus on symmetry breaking by a perturbing field (bottom right column in Fig. 1).

**Simple example**. In this section, we present a simple numerical example of a real-synthetic symmetry and corresponding HHG selection rules. We consider a Ne atom irradiated by two laser beams, with frequencies $\omega = 2\pi/T$ and $3\omega$, polarized along the $\hat{x}$ and $\hat{y}$ axes, respectively (see Fig. 2a, top row, for the Lissajous curve and mathematical expression for the driving field). Due to the spherical symmetry of the Ne atom, the DS of the system is determined by the driving field—it exhibits a 2nd order rotational DS $\hat{C}_2$, defined by $\hat{C}_N = \hat{\tau}_N \cdot \hat{R}_N$. Here, $\hat{\tau}_N$ is a $T/N$ time translation and $\hat{R}_N$ is a $2\pi/N$ rotation within the polarization plane of the laser[10]. The top row of Fig. 2a illustrates the invariance of Lissajous curve of the driving field under the $\hat{C}_2$ operation. As a result, even harmonic generation is forbidden, and the HHG spectrum exhibits odd-only harmonics (Fig. 2b, blue curve). At this point we intentionally deform the Lissajous curve of the driving field such that it does not exhibit any DS by adding a third beam with a relative amplitude $\lambda$, frequency $4\omega$, and $\hat{x}$ axis polarization. The symmetry broken driving field is given by

$$\mathbf{F}(t, \lambda) = (\sin(\omega t) + \lambda \cos(4\omega t))\hat{x} + \sin(3\omega t + \pi/7)\hat{y} \quad (1)$$

and the Hamiltonian of the symmetry broken system is given by

$$\hat{H} = \frac{-\nabla^2}{2} + V(\mathbf{r}) + \mathbf{r} \cdot \mathbf{F}(t, \lambda) \quad (2)$$

where $V(\mathbf{r})$ represents the atomic potential of the Ne atom. For $\lambda \neq 0$, the driving field does not exhibit $\hat{C}_2$ DS because the term $\lambda x \cos(4\omega t)$ changes sign under the $\hat{C}_2$ operation. In order to revive a symmetry in the symmetry-broken Hamiltonian, we incorporate the symmetry breaking DOF $\lambda$ as a synthetic dimension. The synthetic dimension operation $\hat{\zeta}(\lambda) = -\lambda$ inverts the effect of $\hat{C}_2$ on the symmetry breaking term. Figure 2a illustrate this symmetry operation – to recover the initial Lissajous curve, both $\hat{C}_2$ and $\hat{\zeta}$ operations need to act on the field. To obtain the corresponding harmonic generation selection rules, we employ the invariance of the emitted field

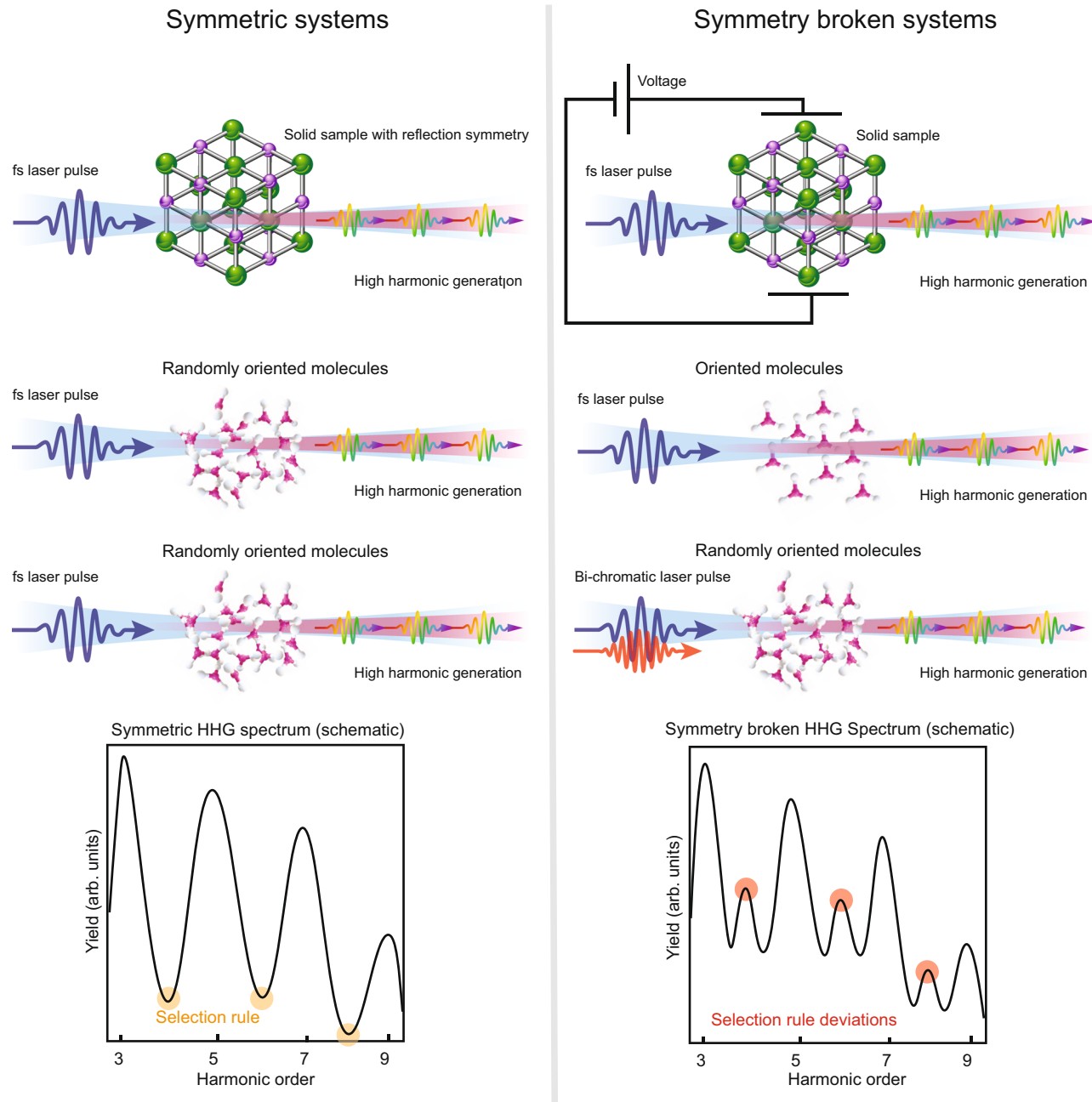

**Fig. 1 Dynamical symmetry and dynamical symmetry breaking in periodically driven systems.** On the left, a linearly polarized laser pulse interacts with either a solid or a randomly oriented ensemble of molecules, so that the system exhibits a dynamical reflection symmetry. The DS imposes a selection rule that forbids even harmonic generation along the $\hat{x}$ axis, as illustrated in the schematic spectrum in the bottom left corner. On the right, the dynamical reflection symmetry is broken either by a DC voltage applied to the solid sample, orientation of the molecular ensemble, or an additional laser field with a different frequency. Because the DS is broken, selection rule deviations in the form of even harmonics appear.

$\mathbf{E}_{\text{HHG}}(t,\lambda)$ under the symmetry operation:

$$\mathbf{E}_{\text{HHG}}(t,\lambda) = \hat{C}_2 \cdot \hat{\zeta}\mathbf{E}_{\text{HHG}}(t,\lambda) = \hat{R}_2\mathbf{E}_{\text{HHG}}\left(t+T/2,-\lambda\right) \quad (3)$$

To reformulate this selection rule for harmonic amplitudes, we employ a Fourier transform $\mathbf{E}_{\text{HHG}}(t,\lambda) = \sum_n \mathbf{E}_n(\lambda)e^{in\omega t}$, where $\mathbf{E}_n(\lambda)$ is the complex amplitude of the n'th harmonic. The operation $\hat{R}_2$ transforms a vector $\mathbf{E}_n(\lambda)$ as $\hat{R}_2\mathbf{E}_n(\lambda) = -\mathbf{E}_n(\lambda)$, and $\hat{\tau}_2 e^{in\omega t} = e^{in\omega(t+T/2)} = (-1)^n e^{in\omega t}$. Consequently, the $\hat{C}_2 \cdot \hat{\zeta}$ selection rule for harmonic amplitude $n$ is:

$$\mathbf{E}_n(\lambda) \equiv (-1)^{n+1}\mathbf{E}_n(-\lambda) \quad (4)$$

That is, the amplitude of each even (odd) harmonic is an odd (even) function of $\lambda$. Importantly, this analytical result is correct for any value of $\lambda$ (i.e., it is non perturbative) because it is a direct result of a symmetry-based selection rule rather than a perturbative expansion. We demonstrate the different parity of even and odd harmonic responses to the perturbation by numerically solving the time-dependent Schrodinger equation (TDSE) for a model Ne atom irradiated by field in Eq. 1 (see section I of the Supplementary Information). Harmonic amplitudes are obtained by Fourier transforming the time dependent dipole acceleration. Figure 2c, d show the numerically exact scaling of the harmonic amplitudes with $\lambda$ over the range between

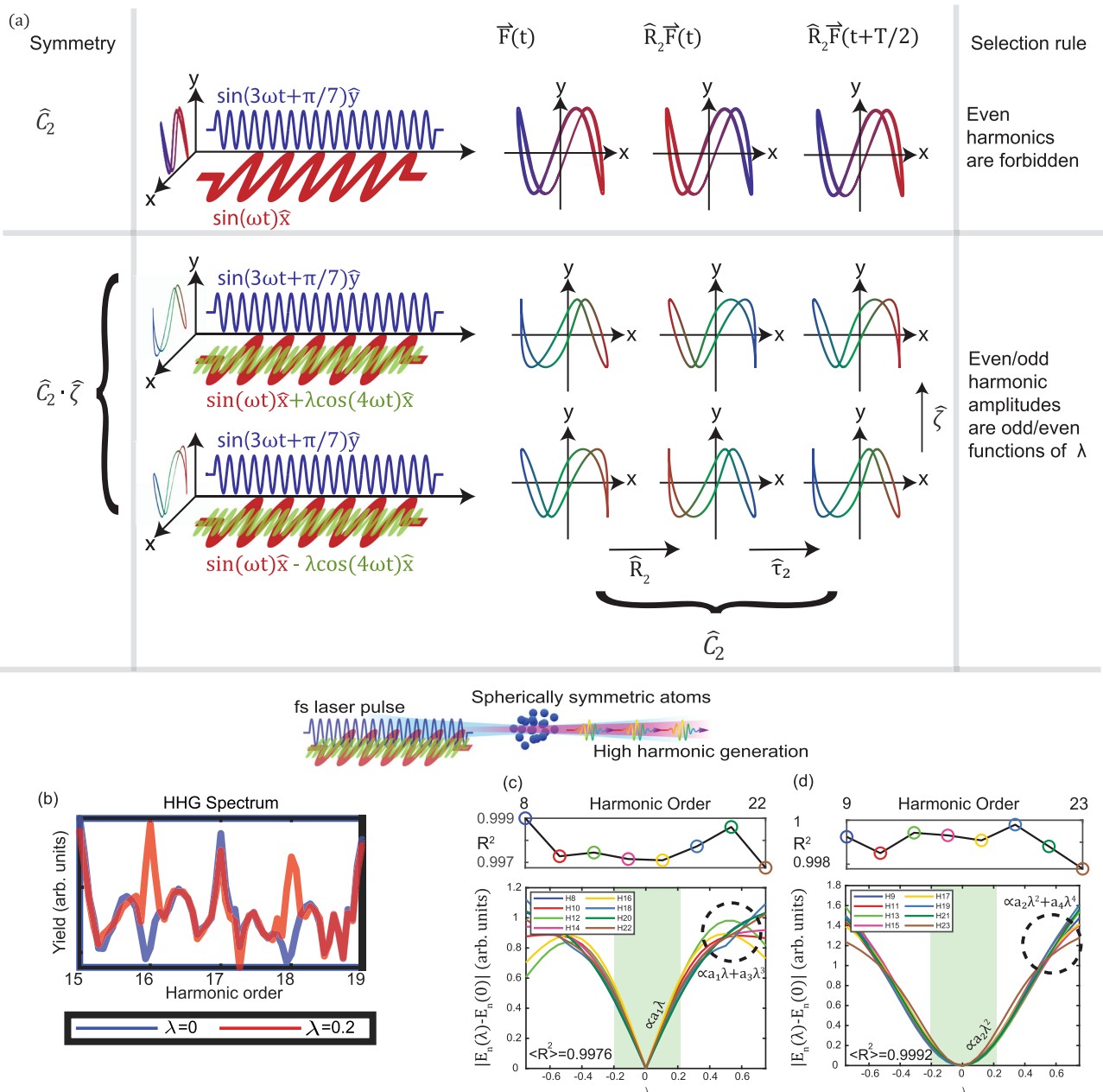

**Fig. 2 HHG selection rules in a system with broken $\hat{C}_2$ symmetry. a** For $\lambda = 0$ (top row), the driving field exhibits $\hat{C}_2$ DS (top row). For $\lambda \neq 0$ (bottom row), the Lissajous curve of the driving field is invariant under $\hat{C}_2 \cdot \hat{\zeta}$ where $\hat{\zeta}(\lambda) = -\lambda$ **b** HHG spectrum for $\lambda = 0$ (blue, even harmonic forbidden) and $\lambda = 0.2$ (red, all harmonic orders allowed). The phase of each Lissajous curve is illustrated by a color gradient that transforms under temporal translations. When an $\hat{x}$-polarized field of frequency $4\omega$ and relative amplitude $\lambda$ perturbs a Ne atom driven by a $\hat{C}_2$ symmetric field, the DS is reduced to real-synthetic symmetry, which coerces **c** $2q$ harmonics of to scale oddly with $\lambda$ whereas **d** $2q + 1$ harmonics scale evenly with $\lambda$. Even (odd) harmonic amplitudes are fit to a linear (quadratic) model in the range $|\lambda| \leq 0.2$ (green shaded), and the average $R^2$ value is presented in the top right of each figure. Individual $R^2$ values are presented above each subfigure. For $|\lambda| > 0.2$, cubic/quadratic contributions deform the lowest order linear/quadratic scaling.

$\lambda = -0.75$ to $\lambda = 0.75$. Fitting even (odd) harmonic amplitudes to a linear (quadratic) $\lambda$-dependence, results in good agreement within the range $|\lambda| \leq 0.2$ (green shaded region; see individual and average $R^2$ values in Fig. 2(c, d)), corresponding to the analytical predictions of odd/even scaling with $\lambda$.

**Experimental investigation of selection rules by real-synthetic DS.** Next, we experimentally explore real-synthetic symmetries and their corresponding selection rules in HHG driven with a bi-chromatic bi-elliptical $\omega_0 - 1.95\omega_0$ field. When this frequency

ratio is employed for HHG, non-integer harmonics are generated, corresponding to different emission channels of $n_1$ fundamental photons and $n_2$ photons of frequency $1.95\omega_0$[31,32]. The non-integer frequency ratio results in a spectral separation of emission channels that would otherwise overlap. As will be shown below, this spectral separation allows us to observe real-synthetic symmetries and selection rules without resolving the polarization of the HHG spectrum.

In our set-up[33] (Fig. 3a), a bi-chromatic laser pulse (40 $fs$ FWHM) with frequencies $\omega_0 - 1.95\omega_0$ (corresponding to the wavelengths 800 nm and 410 nm, respectively) is passed through an achromatic

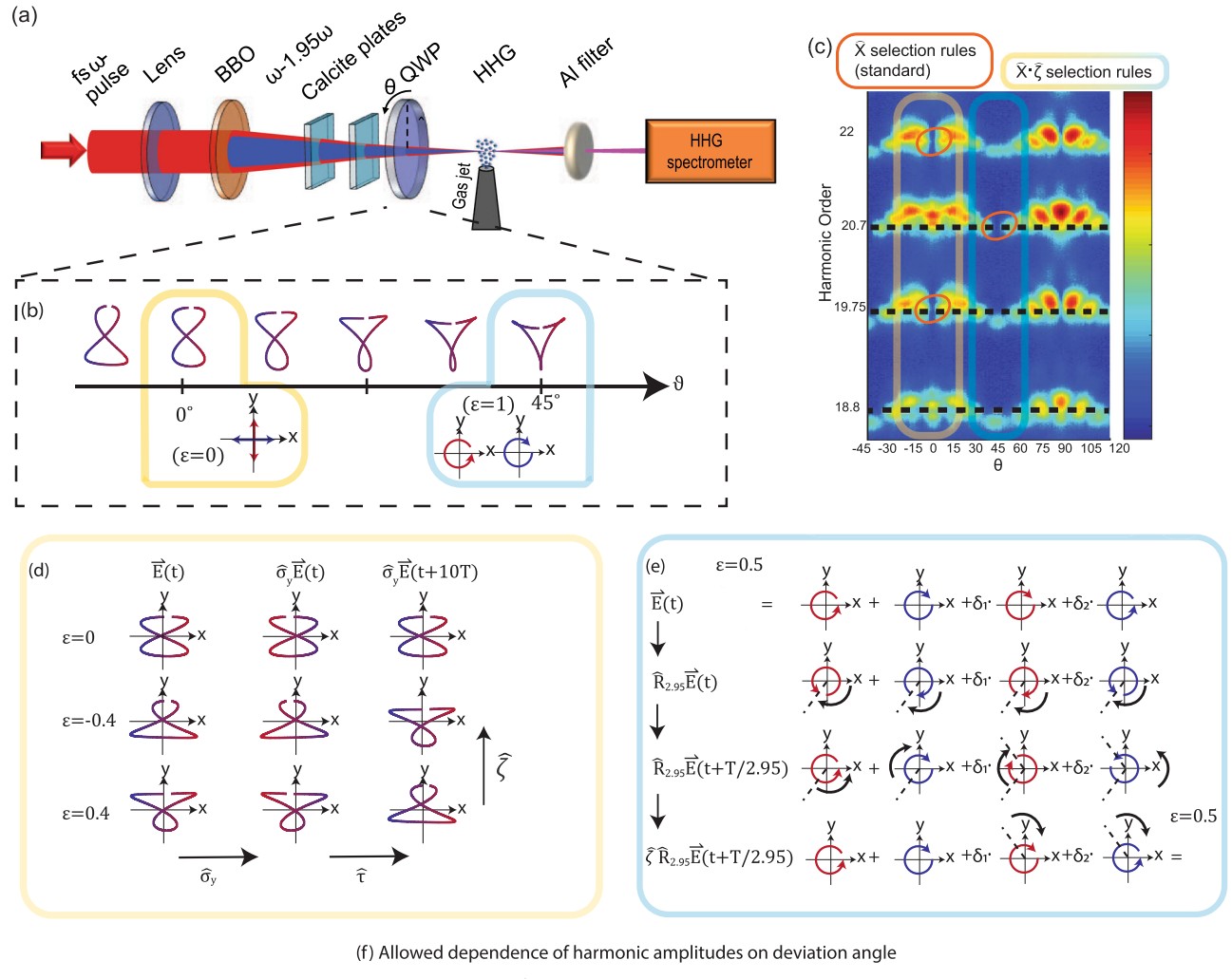

**Fig. 3 Real-synthetic symmetries in bi-elliptical HHG. a** Illustration of the experimental setup[29] **b** Lissajous curves of the driving field for different values of the QWP angle $\theta$, in the temporal window $0 < t < T_{\omega 0}$. The phase of each Lissajous is illustrated by a color gradient. The driving field exhibits a dynamical reflection symmetry at $\theta = 0°$ (yellow) and a dynamical rotation symmetry at $\theta = 45°$ (blue) **c** the measured HHG spectrum as a function of the QWP angle $\theta$. **d** Illustration of the symmetry operations $\hat{Z} \cdot \tilde{\zeta}_{\hat{Z}}$ on the Lissajous curves of the driving field for $\epsilon = 0, \pm 0.4$. **e** Illustration of the symmetry operation $\hat{C} \cdot \tilde{\zeta}_{\hat{C}}$ on the Lissajous curve of the driving field for $\epsilon = 0.5$. To visualize the operation of $\hat{C} \cdot \tilde{\zeta}_{\hat{C}}$ on the field, it is broken to 4 circularly polarized components. **f** Lowest order allowed in the scaling of H20.7, H19.75, and H18.8 with the deviations from $\theta = 0°$ and $\theta = 45°$.

(f) Allowed dependence of harmonic amplitudes on deviation angle

| Harmonic order | H18.8 | H19.75 | H20.7 |
|---|---|---|---|
| $|\theta\text{-}0°|$ | Quadratic | Linear | Quadratic |
| $|\theta\text{-}45°|$ | Cubic | Quadratic | Linear |

zero-order quarter-wave plate (QWP). The rotation angle $\theta$ of the QWP controls the ellipticity $\epsilon(\theta)$ of the pumps (Supplementary Information section V), resulting in the following field:

$$\mathbf{F}(t, \epsilon) = \sqrt{\frac{1}{1+\epsilon^2}} \Re\left\{ e^{i\omega_0 t}\left(i\epsilon\hat{x} + \hat{y}\right) + \Delta e^{1.95 i\omega_0 t}\left(i\hat{x} - \epsilon\hat{y}\right)\right\} \quad (5)$$

Equation (5) describes two counter rotating elliptically polarized beams of ellipticity $\epsilon(\theta)$, at frequencies $\omega_0 - 1.95\omega_0$ where $\omega_0 \equiv 2\pi/T_{\omega_0}$, and $\Delta$ is the two-color amplitude ratio. For $\theta = 0$, the ellipticity is $\epsilon = 0$ and the field is in a "cross-linear" configuration. For $\theta = 45°$, the pump ellipticities are $\epsilon = 1$ and the field is in a "bi-circular" configuration. Figure 3b shows Lissajous curves of the driving field for different values of $\theta$. We note that Fig. 3b depicts the Lissajous curves in the temporal window between 0 and $T_{\omega_0}$, while the periodicity of the bi-chromatic field is $T =$

$20T_{\omega_0}$ (the complete Lissajous are given in the Supplementary Information, section V). We further emphasize that the Floquet frequency of this system is given by $\omega = 2\pi/T$ and not $\omega_0 = 2\pi/T_{\omega_0}$, and it is the Floquet frequency $\omega$ that should be used when applying the general theory outlined in the next section.

The bi-chromatic beam is focused onto a supersonic jet of argon gas at an intensity of $2 \times 10^{14} \text{W/cm}^2$ at the focus, where 10% of the intensity is in the redshifted SH driver (i.e., $\Delta = 1/\sqrt{10}$). The $\hat{y}$-polarization component of the HHG spectrum is measured by a polarizing XUV spectrometer. The measured HHG spectrum (Fig. 3c) exhibits two types of selection rules, which are imposed either by standard DSs, or by real-synthetic DSs. Firstly, for $\theta = 0°$ and $\theta = 45°$, the driving field exhibits standard HHG selection rules in the form of forbidden harmonics due to dynamical reflection and rotation symmetries, respectively[10]. These selection rules determine that even harmonic generation is forbidden along the $\hat{y}$ axis (the

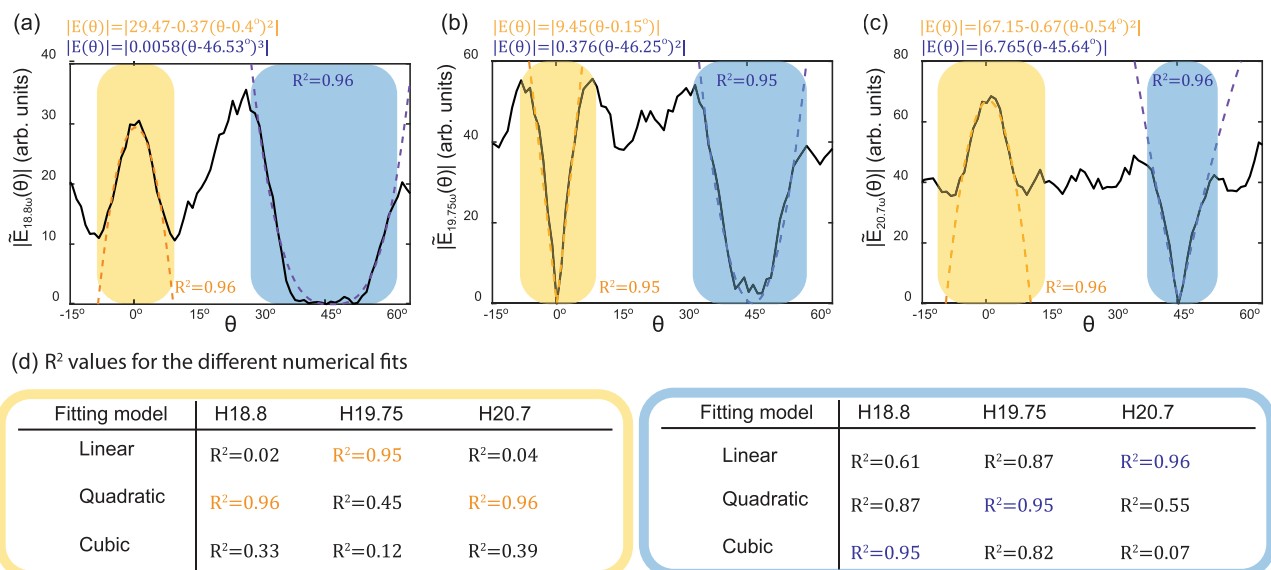

**Fig. 4 Experimental observation of selection rules in a symmetry broken system due to real-synthetic symmetries. a–c** Deviations of harmonic amplitudes from standard selection rules as a function of QWP angle $\theta$ for harmonic orders 18.8, 19.75, and 20.7. The scaling of each harmonic amplitude was fit to three models within the yellow & blue shaded regions, and the best fits appear in the corresponding color above each subfigure. **d** $R^2$ values for all numerical fits. The best fits for each harmonic are marked in yellow in blue.

measurement axis) for $\theta = 0°$ and $3n$ harmonic generation is forbidden for $\theta = 45°$ for integer $n$ (Fig. 3c, marked in orange). As $\theta$ detunes from $0°$ and $45°$, these DSs are broken by the polarization components of the pump, playing the role of a perturbation. In the next paragraph, we will identify explicitly the real-synthetic symmetries imposed by these polarization components and obtain their corresponding selection rules, which constrain the spectral response of the emission. We will show that these selection rules are consistent with the observed scaling of the HHG spectrum as $\theta$ detunes from the high symmetry points $0°$ and $45°$, for 3 exemplary frequency components ($18.8\omega_0, 19.75\omega_0, 20.7\omega_0$, Fig. 3c). Additional spectral components are analyzed in the Supplementary Information, section IV.

As $\theta$ detunes from $0°$, the dynamical reflection symmetry is broken, and instead, the symmetry $\hat{Z}_y \cdot \hat{\zeta}$ is imposed. Here, $\hat{Z}_y$ is the operation $\{x \rightarrow -x; t \rightarrow t + T/2 = t + 10T_{\omega_0}\}$ and $\hat{\zeta}$ is the operation $\{\epsilon \rightarrow -\epsilon\}$. Notably, the operation $\hat{Z}_y$ is defined as $\hat{\tau}_2 \cdot \hat{\sigma}_y$ where $\hat{\sigma}_y$ is a spatial reflection relative to the $\hat{x}$ axis, and $\hat{\tau}_2$ is $T/2$ where $T = 20T_{\omega_0}$ is the period of the bi-chromatic field, $T_{\omega_0}$ is the period of the fundamental field. Figure 3d shows the action of the composite symmetry operation $\hat{Z}_y \cdot \hat{\zeta}$ on the Lissajous curve of the driving field. Employing the invariance of the emission under the symmetry operation, we obtain $\mathbf{E}_{HHG}(t, \epsilon) = \hat{\sigma}_y \mathbf{E}_{HHG}(t + 10T_{\omega_0}, -\epsilon)$. This condition coerces harmonic amplitudes 20.7 and 18.8 (19.75) to be even (odd) functions of $\epsilon$ (and $\theta$) along the $\hat{y}$ axis.

As $\theta$ is detuned from $45°$, the dynamical rotation symmetry is broken. To obtain the real-synthetic symmetry associated with the broken dynamical rotation symmetry, we consider the following general field, formulated with circularly polarized vectors:

$$\mathbf{E}(t) = \Re \frac{i}{2}\left\{\eta\underbrace{(\hat{e}_R e^{i\omega_0 t} + \hat{e}_L e^{1.95 i\omega_0 t})}_{\hat{C}_{2.95} symmetric} - \delta_1 \hat{e}_L e^{i\omega_0 t} + \delta_2 \hat{e}_R e^{1.95\omega_0 t}\right\}$$

(6)

Here, $\hat{e}_{L/R} = \hat{x} \pm i\hat{y}$ where $\hat{x}$ and $\hat{y}$ are basis vectors polarized along the $x$ and $y$ axes, respectively. The parameter $\eta$ is the amplitude of an $\omega_0 - 1.95\omega_0$ bi-circular field exhibiting $\hat{C}_{2.95} = \hat{R}_{2.95} \cdot \hat{\tau}_{2.95}$ DS, where $\hat{R}_{2.95}$ is a $2\pi/2.95$ spatial rotation and $\hat{\tau}_{2.95}$ is a $T_{\omega_0}/2.95$ time translation. The parameters $\delta_{1,2}$ are the complex amplitudes of two circularly polarized symmetry breaking fields which result in selection rule deviations as the symmetry is broken. The physical field in our experiment is represented by $\eta = 1 + \epsilon$ and $\delta_{1,2} = 1 - \epsilon \propto |\theta - 45°|$. The field $\eta(\hat{e}_R e^{i\omega_0 t} + \hat{e}_L e^{1.95 i\omega_0 t})$ exhibits $\hat{C}_{2.95}$ DS (the relations $\hat{R}_{2.95}\hat{e}_{L/R} = e^{\pm 2\pi i/2.95}\hat{e}_{L/R}$ and $\hat{\tau}_{2.95}e^{in\omega_0 t} = e^{2\pi in/2.95}e^{in\omega_0 t}$ are useful for verifying it). In contrast, the fields of amplitudes $\delta_{1,2}$ do not, hence they are symmetry breaking. Instead, these field components are symmetric under the operation $\hat{C}_{2.95} \cdot \hat{\zeta}$, where $\hat{\zeta}$ phase shifts the symmetry breaking field components $\delta_1 \underset{\hat{\zeta}}{\rightarrow} \delta_1 e^{-2 \times 2\pi i/2.95} = \delta_1 e^{0.95 \times 2\pi i/2.95}$, $\delta_2 \underset{\hat{\zeta}}{\rightarrow} \delta_2 e^{0.95 \times 2\pi i/2.95}$.

Figure 3e illustrates how each circularly polarized components of the driving field is transformed separately by the operation $\hat{C}_{2.95} \cdot \hat{\zeta}$. The resulting selection rule for the harmonic amplitudes is $\mathbf{E}_{HHG}(t, \delta_1, \delta_2) = \hat{R}_{2.95}\mathbf{E}_{HHG}(t + T/2.95, \delta_1 e^{0.95 \times 2\pi i/2.95}, \delta_2 e^{-0.95 \times 2\pi i/2.95})$. By expanding $\mathbf{E}_{HHG}(t, \delta_1, \delta_2)$ to a power series in $\delta_1^a \delta_2^b \bar{\delta}_1^c \bar{\delta}_2^d$ (bar represents complex conjugate) and taking $\delta_{1,2} \propto |\theta - 45°|$, we obtain that the symmetry $\hat{C}_{2.95} \cdot \hat{\zeta}$ forbids linear contributions in the scaling of H20.7 and quadratic contributions in the scaling of H19.75 (Supplementary Information, section III). Similarly, it forbids linear and quadratic contributions to the scaling of H18.8, hence it scales cubically with the deviation angle $|\theta - 45°|$. The table in Fig. 3f summarizes the results of this section, listing the lowest allowed orders in the scaling of each of the harmonic amplitudes 18.8, 19.75 and 20.7 with $|\theta|$ and $|\theta - 45°|$.

Figure 4a–c show the measured harmonic amplitudes of spectral components $n = 18.8, 19.75$, and $20.7$ as a function of the waveplate angle, $\theta$, obtained by integrating the measured signal (Fig. 3c) in a range of $n\omega_0 \pm 0.0225\omega_0$ and taking the square root. Each harmonic amplitude curve was fitted to three models of the

form $|a_m(\theta - \theta_0)^m|$, for $m = 1, 2, 3$ (a constant term was allowed for initially allowed harmonics). The obtained $R^2$ values of these fits are summarized in the table in Fig. 4d. The formula and $R^2$ values of the best fit to each harmonic amplitude, as well as overlays of the numerical fits over the measured harmonic amplitudes, are shown in Fig. 3a–c. Comparing the obtained $R^2$ values in Fig. 4d with the predicted scaling (Fig. 3f), we observe that fits to the predicted lowest order allowed contributions resulted with $R^2 > 0.95$ for all the six examined cases, while all other fits were significantly smaller. These results demonstrate the experimental observation of selection rules by real-synthetic DS, which could consequently be used for applications.

Finally, we note that the observed scaling of the harmonic amplitudes around $\theta = 45°$ can also be obtained via emission-channel analysis[32,33] that relies on conservation of energy, parity, and spin, (Supplementary Information, section IV). In contrast, the scaling around $\theta = 0°$ does not have an analogue conservation law derivation.

**General theory**. In this section we classify real synthetic symmetries imposed by dressing laser fields, using Floquet group theory. That is, we consider a Floquet system that initially exhibits some DS $\hat{X}$ to be subject to an external laser field (so-called dressing field), that transforms its DS $\hat{X}$ to a real-synthetic symmetry $\hat{X} \cdot \hat{\zeta}$, and tabulate the corresponding selection rules by Floquet group theory.

We start by considering a *general* Floquet system with period $T = 2\pi/\omega$, and a DS denoted by $\hat{X}$. A Floquet system $\hat{H}_0(t) = \hat{H}_0(t + T)$ exhibits the DS $\hat{X}$ if $[\hat{\mathcal{H}}_f, \hat{X}] = 0$ where $\hat{\mathcal{H}}_f \equiv \hat{H}_0 - i\partial_t$ is the Floquet Hamiltonian. The operation $\hat{X}$ is a (2+1)D spatio-temporal symmetry, jointly imposed by the symmetries of the target material and a first driving laser (or by any other periodic excitation of the system[34,35]). The operations $\hat{X}$ were comprehensively tabulated within the framework of Floquet group theory[10], and for completeness, are given with their corresponding HHG selection rules in Table 1. In Table 1, $\hat{T}$ is the time-reversal operation ($t \to -t$), $\hat{R}_N$ are $2\pi/N$ spatial rotations, $\hat{\tau}_N$ are $T/N$ time translations, $\hat{\sigma}_i$ is a reflection relative to the vector $\hat{i}$, and $\hat{L}_b$ is the scaling operation $\hat{y} \to b\hat{y}$ where $\hat{y}$ is a basis vector parallel to the $y$-axis.

We consider the $\hat{X}$ symmetric Floquet system to be perturbed by a perturbation $\hat{W}$:

$$\hat{H} = \hat{H}_0 + \hat{W} \quad (7)$$

$\hat{W}$ breaks the symmetry $\hat{X}$ such that $\hat{X}^\dagger \hat{H}_0 \hat{X} = \hat{H}_0$, but $\hat{X}^\dagger \hat{H} \hat{X} \neq \hat{H}$. Although $\hat{X}$ is broken, it may still be exploited to formulate a symmetry of the form $\hat{X} \cdot \hat{\zeta}_{\hat{X}}$ in the symmetry broken

system, where $\hat{\zeta}_{\hat{X}}$ operates on the internal degrees of freedom of $\hat{W}$ denoted by the vector $\mathbf{Q}$, while leaving $\hat{H}_0$ unaffected. The operation $\hat{\zeta}_{\hat{X}}$ are derived by solving the equation $\hat{W} = (\hat{X} \cdot \hat{\zeta}_{\hat{X}})^\dagger \hat{W}(\hat{X} \cdot \hat{\zeta}_{\hat{X}}) \equiv \hat{X} \cdot \hat{\zeta}_{\hat{X}}[\hat{W}]$, where the square brackets indicate that the composite operation $\hat{X} \cdot \hat{\zeta}_{\hat{X}}$ transforms the operator $\hat{W}$. In the examples above, $\mathbf{Q}$ is a vector containing the complex polarization components of the symmetry breaking fields, which is acted on by $\hat{\zeta}$. The selection rule for the optical emission is obtained by employing the invariance of the emission under the symmetry operation, that is $\mathbf{E}_{HHG}(t, \mathbf{Q}) = \hat{X} \cdot \hat{\zeta}_{\hat{X}} \mathbf{E}_{HHG}(t, \mathbf{Q}) = \hat{X}\mathbf{E}_{HHG}(t, \hat{\zeta}[\mathbf{Q}])$.

We now focus on the case where $\hat{W}$ represents an additional laser whose amplitude and polarization are given by the complex vector $\mathbf{Q} = (q_x, q_y)$

$$\hat{W} = \Re\{\mathbf{Q}re^{is\omega t}\} \quad (8)$$

Here, $\omega$ is the fundamental frequency of the symmetric Floquet system and $s\omega = 2\pi s/T$ is the frequency of the symmetry breaking field $\hat{W}$, and $s$ is a rational number. Since $\hat{X}$ is a symmetry of $\hat{H}_0$, and $\hat{\zeta}_{\hat{X}}$ only operates on $\hat{W}(\mathbf{Q})$ by definition, the symmetry condition is

$$\hat{W} = \Re\left\{\hat{\zeta}_{\hat{X}}[\mathbf{Q}]\hat{X}[re^{is\omega t}]\right\} \quad (9)$$

For example, if $\hat{X} = \hat{T}$, this equation becomes $\Re\{\mathbf{Q}re^{is\omega t}\} = \Re\{\hat{\zeta}_{\hat{X}}[\mathbf{Q}]re^{-is\omega t}\}$, which is fulfilled by the complex conjugation operation $\hat{\zeta}_{\hat{T}}[\mathbf{Q}] = \bar{\mathbf{Q}}$. If $\hat{X} = \hat{Z}_y$ ($\{x \to -x, t \to T/2\}$), Eq. (9) reads $\Re\{\mathbf{Q}re^{is\omega t}\} = \Re\{\hat{\zeta}_{\hat{X}}[\mathbf{Q}][(-1)^s \hat{\sigma}_x re^{is\omega t}]\}$, which is solved by $\hat{\zeta}_{\hat{Z}_y} = (-1)^s \hat{\sigma}_y^{(\mathbf{Q})}$, where the $\mathbf{Q}$ superscript indicates that $\hat{\sigma}_y^{(\mathbf{Q})}$ operates in the synthetic $\mathbf{Q}$ space ($\{q_x \to -q_x\}$). Table 2 shows operations $\hat{\zeta}_{\hat{X}}$ that solve Eq. (9) for all other Floquet group theory[10] symmetries $\hat{X}$ (derived in section II of the Supplementary Information).

The real-synthetic symmetries, $\hat{X} \cdot \hat{\zeta}_{\hat{X}}$, result in selection rules on various physical phenomena. Particularly, the selection rule for the emitted harmonic light (denoted by $\mathbf{E}_{HHG}(t, \mathbf{Q})$) can be obtained using the invariance of a time dependent observable under the symmetry operation[10], i.e. $\mathbf{E}_{HHG}(t, \mathbf{Q}) = \hat{X}\mathbf{E}_{HHG}(t, \hat{\zeta}[\mathbf{Q}])$. Notably, this equation also holds for other observables, i.e. $\mathbf{o}(t, \mathbf{Q}) = \hat{X}\mathbf{o}(t, \hat{\zeta}[\mathbf{Q}])$ for a general $\mathbf{o}(t)$. However, $\mathbf{E}_{HHG}$ and $\mathbf{o}$ may transform differently under $\hat{X}$ and therefore adhere to different selection rules. For example, $\mathbf{E}_{HHG}(t, \mathbf{Q})$ transforms as the dipole moment expectation value hence it changes sign under $\hat{R}_2$, whereas the expectation value for squared x-axis position $x^2(t, \mathbf{Q})$ does not. Detailed examples of the transformation of the

**Table 1 Floquet group theory and harmonic generation selection rules in (2+1)D.**

| Floquet group symmetry $\hat{X}$ | Harmonic generation selection rule |
|---|---|
| $\hat{T}$ | Linearly polarized only harmonics. They may be polarized along any axis. |
| $\hat{Q} = \hat{T} \cdot \hat{R}_2$ | Linearly polarized only harmonics. They may be polarized along any axis. |
| $\hat{G} = \hat{T} \cdot \hat{\tau}_2 \cdot \hat{R}_2$ | Linearly polarized only harmonics. They may be polarized along any axis. |
| $\hat{Z}_y = \hat{\tau}_2 \cdot \hat{\sigma}_y$ | Linearly polarized only harmonics, even harmonics are polarized along the reflection axis, and odd harmonics are polarized orthogonal to the reflection axis. |
| $\hat{D}_y = \hat{T} \cdot \hat{\sigma}_y$ | Elliptically polarized harmonics with major/minor axis corresponding to the reflection axis. |
| $\hat{H}_y = \hat{T} \cdot \hat{\sigma}_{\bar{y}}$ | Elliptically polarized harmonics with major/minor axis corresponding to the reflection axis. |
| $\hat{C}_N = \hat{\tau}_N \cdot \hat{R}_N$ | (±) circularly polarized $Nq \pm 1$ harmonics, $q \in \mathbb{N}$, all other orders forbidden |
| $\hat{C}_{N,M} = \hat{\tau}_N \cdot \hat{R}_{N,M} = \hat{\tau}_N \cdot (\hat{R}_N)^M$ | (±) circularly polarized $Nq \pm M$ harmonics, $q \in \mathbb{N}$, all other orders forbidden |
| $\hat{e}_{N,M} = \hat{\tau}_N \cdot \hat{L}_b \cdot \hat{R}_{N,M} \cdot \hat{L}_{1/b}$ | (±) elliptically polarized $Nq \pm M$ harmonics, $q \in \mathbb{N}$, with an ellipticity b, all other orders forbidden. |

**Table 2 Real-synthetic symmetries and harmonic generation selection rules in (2+1)D symmetry broken systems.**

| $\hat{X}$ | $\hat{\zeta}_{\hat{X}}(\mathbf{Q})$ | Harmonic generation selection rule |
|---|---|---|
| $\hat{T}$ | $\bar{\mathbf{Q}}$ | $E_{nx}^{(abcd)}, E_{ny}^{(abcd)} \in \mathbb{R}$ |
| $\hat{Q}$ | $-\bar{\mathbf{Q}}$ | $E_{nx}^{(abcd)}, E_{ny}^{(abcd)} \in i^{1+a+b+c+d}\mathbb{R}$ |
| $\hat{G}$ | $(-\mathbf{1})^{\mathbf{1}+s}\bar{\mathbf{Q}}$ | $E_{nx}^{(klhj)}, E_{ny}^{(klhj)} \in i^{n+1+(s+1)(a+b+c+d)}\mathbb{R}$ |
| $\hat{Z}_y$ | $\begin{pmatrix} (-1)^{s+1} & 0 \\ 0 & (-1)^s \end{pmatrix}\begin{pmatrix} q_x \\ q_y \end{pmatrix}$ | $n + (s+1)(a+c) + s(b+d) = 2q \Rightarrow$ $E_{nx}^{(abcd)} = 0$ $n + (s+1)(a+c) + s(b+d) = 2q + 1 \Rightarrow$ $E_{ny}^{(abcd)} = 0$ |
| $\hat{D}_y$ | $\begin{pmatrix} -1 & 0 \\ 0 & 1 \end{pmatrix}\begin{pmatrix} \bar{q}_x \\ \bar{q}_y \end{pmatrix}$ | $E_{nx}^{(abcd)} \in i^{1+a+c}\mathbb{R}$ $E_{ny}^{(abcd)} \in i^{a+c}\mathbb{R}$ |
| $\hat{H}_y$ | $\begin{pmatrix} (-1)^{s+1} & 0 \\ 0 & (-1)^s \end{pmatrix}\begin{pmatrix} \bar{q}_x \\ \bar{q}_y \end{pmatrix}$ | $E_{nx}^{(abcd)} \in i^{n+1+(s+1)(a+c)+s(b+d)}\mathbb{R}$ $E_{ny}^{(abcd)} \in i^{n+(s+1)(a+c)+s(b+d)}\mathbb{R}$ |
| $\hat{C}_{NM}$ | $e^{-\frac{i2\pi s}{N}}\hat{R}^{(\mathbf{Q})}{}_{N,M}\cdot\mathbf{Q}$ | $E_{Rn}^{(abcd)}$ is forbidden unless $mod(n - M(a-b-c+d) - s(a+b-c-d) - M, N) = 0$ $E_{Ln}^{(klhj)}$ is forbidden unless $mod(n - M(a-b-c+d) - s(a+b-c-d) + M, N) = 0$ |
| $\hat{e}_{NM}$ | $e^{-\frac{i2\pi s}{N}}\hat{L}^{(\mathbf{Q})}{}_{1/b}\cdot\hat{R}^{(\mathbf{Q})}{}_{N,M}\cdot\hat{L}^{(\mathbf{Q})}{}_b\cdot\mathbf{Q}$ | $E_{-n}^{(abcd)}$ is forbidden unless $mod(n - M(a-b-c+d) - s(a+b-c-d) - M, N) = 0$ $E_{+n}^{(abcd)}$ is forbidden unless $mod(n - M(a-b-c+d) - s(a+b-c-d) + M, N) = 0$ |

$\mathbf{E}_{\text{HHG}}$ under $\hat{X}$ are given in the Supplementary Information and in ref. [10]. The condition $\mathbf{E}_{\text{HHG}}(t, \mathbf{Q}) = \hat{X}\mathbf{E}_{\text{HHG}}(t, \hat{\zeta}[\mathbf{Q}])$ is a non-perturbative restriction (selection rule) on the response of the dipolar emission of the system as the symmetry is broken, and in principle, it is valid beyond the radius of convergence of a particular perturbative expansion. However, for practical application of these rules, it is instructive to reformulate them as selection rules on the expansion coefficients of a perturbative expansion. We emphasize that we do not employ perturbation theory, but rather reformulate non-perturbative selection rules in a perturbative language. To do this, we write

$$\mathbf{E}_{\text{HHG}}(t, \mathbf{Q}) = \sum_n \mathbf{E}_n(\mathbf{Q})e^{in\omega t} \qquad (10)$$

$$\boldsymbol{E}_n(\mathbf{Q}) \equiv \sum_{a,b,c,d=0}^{\infty} \begin{pmatrix} E_{nx}^{(abcd)} \\ E_{ny}^{(abcd)} \end{pmatrix} q_x^a q_y^b \bar{q}_x^c \bar{q}_y^d$$

Here, $\omega$ is the fundamental frequency of the perturbed Floquet system, $\mathbf{E}_n(\mathbf{Q})$ is the complex amplitude of the n'th harmonic, $q_{x,y}$ are the complex polarization components of the symmetry breaking field (Eq. (8)), $\bar{q}_{x,y}$ are their complex conjugates, $E_{nx}^{(abcd)}, E_{ny}^{(abcd)}$ are expansion coefficients, and $a, b, c, d$ are non-negative integers. The non-perturbative selection rule of $\hat{X}\cdot\hat{\zeta}_{\hat{X}}$ may be translated to a selection rule on $E_{nx/y}^{(abcd)}$ to all orders in $a, b, c$ and $d$, or alternatively, to selection rules on the expansion coefficients of any other perturbative expansion. For example, we found above that for $\hat{X} = \hat{T}$, the synthetic dimensions operation is $\hat{\zeta}_{\hat{T}}[\mathbf{Q}] = \bar{\mathbf{Q}}$. Hence, $\mathbf{E}_{\text{HHG}}(t, \mathbf{Q}) = \mathbf{E}_{\text{HHG}}(-t, \bar{\mathbf{Q}})$, which implies that the expansion coefficients $E_{nx/y}^{(abcd)}$ must all be real. Table 2 shows these selection rules for all Floquet group symmetries (derived in the Supplementary Information, section II). Similar rules were also derived for the ATI spectrum (see SI, section VI).

The rules presented in Table 2 are consistent with the numerical example presented above where the perturbation is monochromatic ($\hat{C}_2$ symmetry breaking). We emphasize that the conditions of our experiment involve a bi-chromatic symmetry breaking perturbation. In this case, the bi-chromatic perturbation implies that the synthetic operations act in a higher dimensional space, transforming each color of the perturbation separately. Then, the selection rules are derived in the same manner but with a more elaborate series expansion. In the general case of a laser with two colors, we may write

$$\hat{W} = \Re\left\{\mathbf{Q_1}\cdot\mathbf{r}e^{is_1\omega t} + \mathbf{Q_2}\cdot\mathbf{r}e^{is_2\omega t}\right\} \qquad (11)$$

where $s_{1,2}$ determine the color of each perturbation, and $\mathbf{Q}_{1,2}$ are their complex amplitudes, and $s_1\omega, s_2\omega$ and $\omega$ are mutually commensurate frequencies. Now, the parameter space that defines $\hat{W}$ is given by $\{\mathbf{Q}_1, \mathbf{Q}_2\}$, and the synthetic dimensions operation is given by $\hat{\zeta} = \hat{\zeta}_1\cdot\hat{\zeta}_2$ where $\hat{\zeta}_i$ operates only on $\mathbf{Q}_i$ (i = 1, 2). Here, $\hat{\zeta}_{1,2}$ are the operations tabulated in Table 2 corresponding to $s_{1,2}$ respectively. The corresponding selection rule for the emission is given by $\mathbf{E}_{\text{HHG}}(t, \mathbf{Q}_1, \mathbf{Q}_{,2}) = \mathbf{E}_{\text{HHG}}(t, \hat{\zeta}_1[\mathbf{Q}_1], \hat{\zeta}[\mathbf{Q}_2])$, which can be translated to selection rules on the coefficients of a series expansion, in a manner identical to the one presented above. The process of concatenating the symmetry operations tabulated in Table 2 and deriving the corresponding selection rules is not limited to bi-chromatic perturbations and one may directly extend it to obtain the real-synthetic symmetries and selection rules associated with a polychromatic perturbation. Finally, we emphasize that the only necessary condition for this construction is that the system exhibits a broken-symmetry, and that there exists a unitary/anti-unitary solution for $\hat{\zeta}_{\hat{x}}$ [10].

## Discussion

To summarize, we have demonstrated that systems that are traditionally regarded as symmetry-broken, systematically exhibit a specific class of symmetries and selection rules through synthetic dimensions. These determine how the

system's observables scale as the system transitions out of its original symmetric state, showing the role of the broken symmetries in the dynamics. We have tabulated these symmetries for periodically driven Floquet systems subject to two driving fields, and derived the corresponding selection rules for HHG, ATI and more. We observed experimentally that the scaling of the HHG spectrum is consistent with selection rules rooted in synthetic dimensions by driving HHG with a bi-chromatic, bi-elliptical laser field. We highlight that our theory is a non-perturbative and applies to all orders of the perturbation's strength. We further emphasize that real-synthetic symmetries and their associated selection rules are general concepts, relevant to all systems with a broken symmetry in real-space and time. For example, one (or both) of the lasers that we have employed may be replaced by a different periodically oscillating (or static) element (either extrinsic or intrinsic), e.g., spin–orbit coupling strengths[36] (see section V in the Supplementary Information), Floquet dark bands[6] (section VI in the Supplementary Information), or lattice excitations[34,35,37]. Specifically, by reformulating them as effective gauge fields, the derived symmetries and selection rules (Table 2) can be directly applied to dynamical symmetry breaking by phonons and magnons, opening opportunities for all-optical time-resolved spectroscopy (and control) of their dynamics. Overall, the presented approach provides a unified framework for the analysis of symmetry-broken systems, complementary to perturbation theory, hence we expect it to be used throughout science and engineering.

## Data availability

Data presented in Figs. 3 and 4 have been deposited in a Zenodo repository at https://zenodo.org/record/5977834#.Yf-dlOpBwkk. Any other data supporting the findings of this study are available from the corresponding author upon reasonable request.

## Code availability

The code supporting the findings of this study are available from the corresponding author upon reasonable request.

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

## Acknowledgements

This work was supported by the Israel Science Foundation (Grant No. 1781/18). O.N. gratefully acknowledges the support of the Adams Fellowship Program of the Israel Academy of Sciences and Humanities, support from the Alexander von Humboldt foundation, and support from a Schmidt Science Fellowship.

## Author contributions

M.E.T. and O.C. initiated this research direction. M.E.T, O.N, E.B., A.F., and O.C. made substantial contributions to all aspects of this work. O.C. supervised the project. M.E.T. wrote the paper.

## Competing interests

The authors declare no competing interests.

**Additional information**

