## [Peer Review File · Nature Communications]

REVIEWER COMMENTS

Reviewer #1 (Remarks to the Author):

In this paper, the authors derive a new set of selection rules for higher harmonic generation and above threshold ionization. The key difference from conventional selection rules, which depend only on symmetries of the system being probed, is that the polarization of the laser is treated as a separate degree of freedom (referred to by the authors as a “synthetic dimension”) for which the goal is to constrain combined polarization-real space properties of the output field. Based on this idea, they experimentally demonstrate that certain higher harmonic terms in the angular dependence of input polarization are suppressed when expanding around high symmetry axes.

The work seems correct, though almost all of the actual calculations are pushed to a very technical supplement and build on a detailed Floquet symmetry classification from a previous work such that I can't claim to have checked their results line by line. Whether it meets the criteria for Nature Communications is a bit trickier of a question – I lean towards no. On the one hand, the results are rather general, as they essentially argue for an explicit supplementing of the symmetry by controllable terms in the drive Hamiltonian (mostly an electric field treated in the unscreened electron approximation, though argued to be more general). On the other hand, I don't find any of the specific cases that they solve to be particularly interesting, and they don't really give any convincing arguments that this generalized symmetry has any meaningful use. The experimental example is very simple theoretically, and indeed they really only measure symmetries of the bichromatic drive and not of the argon gas itself. I suspect that many of these Taylor series showing various missing powers of theta could be derived from direct Taylor series of the time-dependent fields without any reference to this symmetry argument. Furthermore, the paper is written in a very confusing manner. I had to read it through approximately three times before even understanding the gist, and more so to understand technical details. Independent of relevance to Nature Communications, I strongly urge the authors to rewrite the paper in such a way that specific examples of this extended symmetry and their relevance to experiment are shown, instead of generic arguments whose relevance is impossible to assess.

Besides this overall concern, I have the following specific notes:

- 1) I find it confusing the refer to p as the polarization, given that we are considering nonlinear response, e.g., powers $p_{\{x,y\}}^n$ for $n > 1$. More accurately, this is the incoming (weak) electric field for which the x and y components are treated as independent scalars.
- 2) Calling the method non-perturbative is confusing, given that it is precisely a $k + l + h + j$ order perturbation in p which is considered for a given component. If the idea is that one would be able to make a general constraint on response as a function of symmetries of p without direct reference to each term in the Taylor series, that argument should be made more clearly.
- 3) In Eq. 4 of the main text, it is confusing to call the input field E because this has been used for the output electric field throughout the paper, including in deriving the selection rules throughout the supplement.
- 4) I'm not a fan of the term “synthetic dimensions” in referring to the polarization degrees of freedom, as they do not represent dynamical degrees of freedom. Instead, I would just say “polarization degrees of freedom” or, more generally, “parametric degrees of freedom” to capture the (unproven) statement that in principle you could also have symmetries related to the spin-orbit coupling parameters.

Reviewer #2 (Remarks to the Author):

In this work, the authors propose the concept of real-synthetic symmetries as a tool to derive selection rules in strong field phenomena (such as high-harmonic generation and ionization). For a system with a certain symmetry that is subject to a symmetry breaking perturbation, a real-synthetic symmetry is a operation formed by composing the original symmetry, with an additional operation that acts on the perturbation degrees of freedom, such that the perturbed system remains invariant. Existence of these real-synthetic symmetries constrains the physical response, imposing selection

rules. The method is non-perturbative in nature, going beyond derivation of selection rules based on low order perturbation theories.

The authors construct (in the supplementary material) the real-synthetic symmetry operations for a series of different symmetries and table corresponding selection rules for both high-harmonic generation and above-threshold ionization.

The authors then apply the general method to high-harmonic generation for a bi-chromatic field and contrast theoretical predictions based on real-synthetic selection rules with experimental results, obtaining excellent agreement. Importantly, a quartic response (which would be hard to obtain with perturbation theory) to a symmetry breaking term is predicted and observed experimentally.

The paper is very interesting and the notion of non-perturbative selection rules based on real-synthetic symmetries will most likely attract attention and be useful to the strong-field physics community.

However, I believe the paper has some problems with the presentation that should be addressed. I list them in the following:

- 1) In page 6 before equation 4, it is stated that a bi-chromatic field with incommensurate frequencies is considered. The two frequencies ω and 1.95ω . Since $1.95 = 39/20$, the two frequencies are actually commensurate and the field in Eq. (4) is periodic in time, with a period of $20 \cdot 2\pi/\omega$.
- 2) The definition of the time translations is confusing. Is $T = 2\pi/\omega$, the period of the ω -field and $T' = 20T = 20 \cdot 2\pi/\omega$ the period of the field in Eq. (4)? This is never clearly stated, which obfuscates the whole presentation.
- 3) It is said that the σ_x operator is a "reflection relative to the Cartesian basis vector \hat{x} ". Does this mean a $x \rightarrow -x$ exchange (mirror symmetry with respect to the y - z plane)?
- 4) In the main text, $C_{\{59, 20\}}$ is said to represent a $2\pi \cdot 20/59$ rotation composed by a time translation by $T'/20$. However, in the SM the symmetry $C_{\{N, M\}}$ is defined as a $2\pi \cdot M/N$ rotation composed with a time-translation by the period divided by N . These two definitions seem to be at odds. Should the $C_{\{59, 20\}}$ symmetry of the main text actually be a $C_{\{20, 59\}}$?
- 5) In table I, besides the HHG selection rules, selection rules for above-threshold ionization are also listed. However, the ϕ coefficients are only defined in the SM. I believe that either the ϕ coefficients are defined in the main text or, table I should only list the HHG selection rules.
- 6) In order to improve clarity of the method, more of the details in Section VI of the supplementary material should be provided in the main text.
- 7) In the supplementary material, below Eq. (II.13) it is said "is only nonzero if n and $(a + c)$ are of the same parity", but c was previously set to 0. So what does this statement mean?
- 8) In page 18 of SM, I assume that T symmetry is time-reversal symmetry. Is this correct?
- 9) In Section II of the SM, symmetries D and H are mentioned (but not defined). Are these the same as D_y and H_y of Section I?
- 10) In Section IV of the SM, details on the numerical solution of the time dependent Schrodinger equation are provided. It is stated that "The high harmonic generation (HHG) spectra for the different cases analyzed in the paper were obtained by numerically solving the time dependent Schrodinger equation (TDSE) for an atom irradiated by a laser field, in the length gauge". Does this refer to the

results shown in Fig. III.1? This should be made more explicit.

11) In Section V of the SM, which discusses real-synthetic symmetries in a spin-orbit coupled system, it is said that S_4 is a $2\pi/4$ rotation in "synthetic spin space". However, from Eq. (V.5) it seems to be a rotation of the Pauli matrices and therefore a rotation of the usual spin degrees of freedom. Why is it then referred to as a synthetic spin space? Is it synthetic in the same sense of "real-synthetic symmetry"?

Provided these issues are addressed, I believe this work warrants publication in Nature Communications.

Finally, I have a question that I would like for the authors to answer. The whole concept of real-synthetic symmetries, appears to rely on the fact that the frequency of the perturbation is commensurate with the frequency of the driving field of the "unperturbed system", such that the perturbed system is still periodic in time and Floquet theorem can be used. What if this is not the case and the frequency of the perturbation is incommensurate? Could approximate selection rules be obtained by studying commensurate approximants? Could the approximate selection rules be improved by considering higher order (longer period) commensurate approximants? Commensurate approximants are employed in the study of electronic systems subject to quasi-periodic perturbations and I wonder if the same could be made in the context of perturbed Floquet systems.

Reviewer #3 (Remarks to the Author):

In this manuscript, the authors present a novel and valuable group-theoretical framework for handling symmetry-breaking perturbations to a system that obeys a dynamical symmetry of Floquet type. The authors' framework unifies a number of existing and (currently) independent and disparate approaches, clarifies which aspects are universal and which aspects are specific to perturbation-theory approaches, and should provide a solid platform on which to build future work, particularly regarding symmetry-breaking spectroscopies.

The core idea of the framework is that, in general, if one has a system which is originally symmetric and then has its symmetry broken, the symmetry-breaking perturbation generally has one (or more) controlling parameters; and, moreover, the broken symmetry operation can typically be restored by a suitable transformation on that controlling parameter. By treating this controlling parameter as a 'synthetic' dimension, the authors are able to provide a full symmetry to the (expanded) system, which provides clean and universal selection rules for how an experimental observable can depend on the symmetry-breaking parameter.

The results presented in this paper are extremely strong and valuable, and they definitely merit publication in Nature Communications. However, the presentation of the results requires significant work before the paper can be accessible to the broad audience of this journal (and, in particular, to the audiences that stand to benefit from the authors' results). As such, I feel that major revisions are required to the manuscript before it can be published.

In short, I found the text extremely hard to read, mostly due to the structure of the presentation but also due to some of the choices regarding the details of the exposition. The presentation is extremely mathematical and dry, with very little that the reader can use to hang on to and create an intuitive picture of the material. This can only have the effect of losing most readers early on, in such a way that the manuscript fails to communicate its core message to its intended audience. Because of this, I would recommend that the authors re-build the manuscript from the ground up, with a didactical approach in mind.

To be frank, I feel that what is required is that the authors explain the framework in detail, in person,

to a small number (say, five?) of masters- or upper-undergraduate-level students, take the strategies that work in those explanations, and use them as the backbone to restructure the paper. It is hard to give more concrete advice, but I can provide the following major and minor comments:

Major comments

1. The core idea of the framework (which I have tried to enunciate in a compact but clear fashion in the second paragraph of this report) is very hard to glean from the abstract and introduction of the paper, and this makes it extremely hard to get started with the paper.
2. The text dearly needs a clear example that the reader can use as a test case to which the full-blown formalism can be applied: a concrete, manipulatable, visualizable instance that can be used to understand the (very) abstract general formalism. This should ideally be presented *before* the general formalism, so the overall ideas can be presented in a more intuitive way, and the abstractions of the general formalism can then be overlaid on the models built using the more intuitive concrete instance.

The authors already have two options of specific cases that can be used for this initial instance: the bi-elliptical HHG configuration in the second half of the current main text, and the 'tutorial example' that's currently buried in §III of the supplement. I feel that the latter is a better choice, as the geometry is simpler and the point is more easily conveyed.

3. The mathematical presentation requires more attention to detail, and the notation feels, at times, inconsistent. In particular, in equations such as $W = X \cdot \zeta_X W$, it is not clear whether X and ζ are operators that are *multiplying* W , or super-operators that are acting on it. The notation as written suggests the former, but the formalism as it's actually employed implies the latter. The authors should make sure that their notation is crystal clear in separating these two aspects.

4. The main text of this paper must be self-contained: in particular, it cannot rely on references to external papers (and specifically to the authors' previous work in ref. [8]) for definitions of any symbols or notations that it employs. This is particularly noticeable in Table 1, which in its current state just lists multiple undefined symbols (symmetry operations) and is therefore not actionable.

As an offshoot of this: the symbols corresponding to ATI (and particularly $\tilde{\phi}_{\mathbf{k},n}^{(abcd)}$) *must* be defined in the main text if the corresponding selection rules are reported in the main text. I personally feel that they are not really required, and that it is perfectly reasonable for the authors to move the entire discussion of ATI to the supplement (or to a separate paper) and keep the discussion of the main text focused on harmonic generation.

5. Many of the revisions suggested in this report imply an increase in the length of the main text -- sometimes significantly so. If the authors feel that there is some overall need for brevity, I would suggest a thorough reconsideration. There is no requirement to fit the text in under a fixed number of pages, and clarity is a very high price to pay for conciseness.

6. The manuscript claims that the authors' framework provides "system-independent scaling laws [...] valid to all orders in the strength of the perturbation". I disagree with the way the results are being represented here. The authors' framework provides *selection rules* -- sometimes quite restrictive ones -- on the Taylor series of the observables with respect to the strength of the perturbation, but the analysis cannot be used to conclude that a single term suffices within that Taylor series. It will typically be the case that the leading term will dominate at low perturbation strength, but that is obviously contingent on the validity of the relevant order of perturbation theory. In general, multiple terms can coexist (say, a linear term and a cubic one), providing a nontrivial shape to the response which cannot be predicted from the authors' formalism.

7. In the formal presentation of the results (the paragraph around eq. (1)), I found it confusing whether the presentation was for *generic* configurations or for some specific case (of laser-driven dynamics). Does the formalism presented here apply to the spin-orbit coupling mentioned shortly afterwards? If it doesn't, then shouldn't the formalism be reformulated to an even more abstract one that does cover all of the intended cases of applicability? (note that often an increase in abstraction can actually increase clarity.) This complete-generality case can then be further narrowed to the given form of W for the specific case of laser-driven systems as required.

8. In the form of W presented in eq. (1), it's unclear what happens if the presentation contains more than one colour. Does s play any important role in the resulting calculations? (if so, it wasn't immediately obvious to me, perhaps due to the lack of actionable examples where this could be seen in action.) If not, then shouldn't W be replaced by a more arbitrary and general form?

This is particularly important because the main example, in the second half of the manuscript, *does* contain a polychromatic perturbation, which is introduced without accounting for what extra difficulties that implies.

9. Where does the equation $W = X \cdot \zeta_X W$ come from, and what does it actually mean? Perhaps my confusion is mostly just caused by unclear notation, but this key touchstone of the formalism is introduced without much explanation of its origin and its significance.

10. On a similar note, the authors state "The operation ζ_X may be systematically derived by solving the equation $W = X \cdot \zeta_X W$ " ... but they never really explain how that equation can be solved. I found the given examples insufficient to understand the procedure and what's really at stake here, both for practical matters (how is this done in practice?) as well as more theoretical aspects (is the solution always guaranteed? is it unique?).

11. The role played by the specific form of the harmonic response (and particularly how it transforms under X) is unclear in the current text. What happens with other observables?

12. It is unclear to me why the authors chose to represent the bielliptical fields used in the second half of the paper as a $w:1.95w$ frequency relation. The detuning of the second harmonic by 2.5% was a key component of the original experiment on bicircular fields (ref. 11) but it does not really add anything here, since the spectrum (fig. 1(b)) is never separated into individual sub-channels in the way done in ref. 11. The cost of doing this is a significant price in terms of the clarity of the presentation, for very little real gain. The presentation in the main text should use a $w:2w$ frequency ratio, and avail itself of all of the corresponding gains in simplicity of exposition. If required, the authors can mention that the experiment (for practical experimental reasons?) used a slightly different ratio, and perhaps present the analysis using the $w:1.95w$ ratio in the supplement. But I see no reason for the current choices.

13. In the analysis of bielliptical fields, it is extremely unclear at the start of the presentation what $\mathbf{p} = (p_x, p_y)$ should be understood as within this context. This introduces a disconnect between the first and second halves of the manuscript which significantly damages the readability and cohesion of the text.

Moreover, once the analysis gets going, p is used in a much more hand-wavy way than as originally used in the definition of W in eq. (1). If the synthetic dimension is going to be a generic parameter, that's fine by me, but it does need to be correctly represented from the beginning.

14. In a similar vein, the p plane depicted in fig. 1(d) is completely unclear to me. This needs to be defined much more clearly and much more prominently. The space defined by the geometry shown there is definitely extremely valuable -- which implies that it should be included in the paper, but just adds to the need for a full definition and a clear explanation.

15. The authors claim that their analysis is independent of any use of perturbation theory. However, when actually faced with the concrete case of how their bi-elliptical configuration behaves as θ is varied away from 0° and 45° , the authors immediately resort to representing the fields to first order in the deviation of θ from these points, which significantly undercuts the results.

16. The fitting procedure to a polynomial model mentioned in p. 7 needs to be explained in significantly more detail, and specific attention needs to be paid to the uncertainties that result in the fit coefficients produced by the procedure as arising both from the experimental uncertainties in the underlying data as well as the confidence-interval estimations produced by the fit itself.

This is particularly relevant when the authors state "in accordance with the analytically-derived selection rule that forbids quadratic contributions (because the coefficient for θ^2 is ~ 200 weaker than the θ coefficient)". This is inappropriate: either the predicted coefficient for the quadratic scaling is consistent with zero, or it isn't. As reported, it isn't. If the accuracy of the experiment is such that the reported fit coefficient is actually negligible, this needs to be argued and justified in full.

17. One alarming bit of (mis-)notation: when the authors say " $H_{19.75}$ scales as $9.5(\theta - 0.1) + 0.05(\theta - 0.1)^2$ ", what is "0.1"? is it 0.1 radians, or 0.1° ? The inconsistent handling of the degree symbol, both in the text and in fig. 1, is not acceptable in published literature.

(And, linking back to my previous comment -- what is this offset, and why was it produced? what kind of fit was used that implied such a form? why was that form used?)

Minor comments

18. The text makes significant use of Floquet theory, including the concept of the Floquet Hamiltonian (which is rather esoteric to uninitiated readers), throughout. The authors should include a reference to a suitable textbook-level introduction.

19. In the citation block 10-15 in the introduction, the authors could consider including [PRL 112, 135502 (2014)], which presents the same symmetries used in nonlinear-optical harmonic generation at a vastly different length scale.

20. The authors' use of the inverted caret for the symmetry operation over the synthetic dimension is over-complicated notation for no good reason. As a guide, if the general reader cannot give a clearly-recognizable name to the notation (other than "zeta with a funny hat"?), then it should be changed. In some situations, this type of complex notation is unavoidable (i.e. when all other avenues have been exhausted), but this is not the case here.

21. The use of the indices k, l, h, j , in eq. (3) and afterwards, is extremely confusing. Why those letters? why not a consecutive set?

22. I find eq. (4) essentially unreadable. This is partly due to the choice of vector notation (as opposed to e.g. a matrix used to represent the QWP?) and partly due to the lack of symbolic values for field strengths. The notation would be much clearer if F_1 and F_2 were used instead of the mysterious factors of $\sqrt{10}$.

23. In fig. 1(f-h), what is the signal that's being plotted? Is it a lineout? an integral over a frequency range? if so, what is the range?

24. As a brief comment regarding formatting: I can understand (but don't really agree with) the motivation for providing reviewer copies in double spacing (i.e. to allow for easier annotation). However, this does not apply to the bibliography, and providing the references section in (a) double spacing with (b) extremely large font size, makes no sense at all. The only result is a waste of paper on the side of the reviewer.

25. Similarly, providing the supplementary information in double-spaced format just produces a waste of paper on the side of the reader. The SI is already long and unwieldy enough. For the sake of the reader, the final version of the SI should be single-spaced. And I would argue that for the sake of the reviewer, the review copies should be single-spaced as well.

26. The supplementary material requires a thorough round of spell-checking, proof-reading and copy-editing. There's multiple format mishaps and various typos that should be fixed before the SI is published.

27. In the introduction of the SI, it mentions that section IV contains the details of numerical simulations -- which have not yet been mentioned!

28. In figure III.1 in the SI, is there a line joining the crosses? Or are the crosses adjacent to each other? It's impossible to tell, and that makes it hard to tell how the plots should be interpreted.

29. In §IV of the SI, the text mentions that "The ground state was found by representing [the] field-free Hamiltonian in matrix form on the cartesian spatial grid" without mentioning how the derivatives in the kinetic energy operator were handled.

30. In §V of the SI, the form of the hamiltonian in eq. (V.1) is not suitably justified with reference to suitable literature, and it is hard to match against existing resources. The authors should provide appropriate references and explain in more depth the origin and handling of this hamiltonian. Similarly, the symmetry operations in eq. (V.4) are not defined with enough detail for the reader to be able to understand and interpret them correctly.

31. In §V of the SI, the authors find good selection rules for the emission produced by each electron of well-defined crystal quasimomentum \mathbf{k} , but they do not explain how those individual emissions are collected together and what that combination procedure does to the emission. This procedure depends on the situation (i.e. perturbative vs nonperturbative harmonic generation) and needs to be handled carefully. More generally, it is not clear how the authors connect the previous part of the section to any experimental observables.

(And, taking these two together, a case can be made that this section should be spun out and made to form the nucleus of a separate publication, but that's a choice for the authors to make.)

32. In p. 37 (!) of the SI, after equation (VII.3), the authors write $[C_6, W, W] \neq 0$ -- yet another formulation of the symmetry operations and their relationship to the Hamiltonian! This emphasizes the need for a consistent and clearly handled use of notation.

As mentioned above, the presentation of the results requires a significant overhaul, probably including an almost complete rebuild of the manuscript. There were further smaller details, but it only makes sense for the peer-review process to address them once the manuscript is closer to publishable form.

I hate to be the bearer of bad news -- particularly for a paper with results as valuable as these -- but I

really think that if the presentation issues I have mentioned are not addressed, the paper will become significantly less usable by its target audience and its impact will become vastly reduced compared to what the results really deserve.

We thank all referees for their considerable effort in reviewing our work and for all their comments. Your reviews were helpful and valuable. Following, the reviews, we revised the MS significantly, and rebuilt it from the ground up, as you reviewer 3 suggested. According to feedback from our colleagues, the revised paper is significantly easier to read and conveys the message much more smoothly. We have tried to avoid exotic notation, and to make things as simple and didactic as possible.

The structure of the revised MS is the following: we begin with a figure to quickly remind what selection rules and selection rules deviations are. Then move to the “tutorial example”, which was in the SI of the original version. Then we present the experiment, introducing more details and extending the conceptual framework, while still treating a specific problem and avoiding abstraction. After the numerical and experimental investigations were presented, we develop the general theory and relate it to the specific examples. In addition, we transferred the discussion of ATI to the SI. The SI was significantly revised and organized.

Finally, we corrected an error. The first non-zero term of H18.8 around WP angle 45 deg is cubic and not quartic.

Below, we address the comments of the referees point by point:

Reviewer #1 (Remarks to the Author):

In this paper, the authors derive a new set of selection rules for higher harmonic generation and above threshold ionization. The key difference from conventional selection rules, which depend only on symmetries of the system being probed, is that the polarization of the laser is treated as a separate degree of freedom (referred to by the authors as a “synthetic dimension”) for which the goal is to constrain combined polarization-real space properties of the output field. Based on this idea, they experimentally demonstrate that certain higher harmonic terms in the angular dependence of input polarization are suppressed when expanding around high symmetry axes.

The work seems correct, though almost all of the actual calculations are pushed to a very technical supplement and build on a detailed Floquet symmetry classification from a previous work such that I can't claim to have checked their results line by line. Whether it meets the criteria for Nature Communications is a bit trickier of a question – I lean towards no. On the one hand, the results are rather general, as they essentially argue for an explicit supplementing of the symmetry by controllable terms in the drive Hamiltonian (mostly an electric field treated in the unscreened electron approximation, though argued to be more general). On the other hand, I don't find any of the specific cases that they solve to be particularly interesting, and they don't really give any convincing arguments that this generalized symmetry has any meaningful use. The experimental example is very simple theoretically, and indeed they really only measure symmetries of the bichromatic drive and not of the argon gas itself. I suspect that many of these Taylor series showing various missing powers of theta could be derived from direct Taylor series of the time-dependent fields without any reference to this symmetry argument.

We believe that our MS meets the criteria for publication in Nature Communications because it presents a truly general and useful new concept-symmetry breaking perturbations systematically impose new class of symmetries that result with observable selection rules. It is relevant to so many perturbative systems in Nature and technology. For example, in future publications we shall apply it to analyzing polarization moments of molecules and photonic structures (the concept is not limited to Floquet systems).

In the paper, we present numerous examples for the generality of the concept, analytical and numerical calculations, and experimental observation of six analytical selection rules, in the context of HHG.

Furthermore, the paper is written in a very confusing manner. I had to read it through approximately three times before even understanding the gist, and more so to understand technical details. Independent of relevance to Nature Communications, I strongly urge the authors to rewrite the paper in such a way that specific examples of this extended symmetry and their relevance to experiment are shown, instead of generic arguments whose relevance is impossible to assess.

Thank you. We revised the MS significantly. According to feedback from our colleagues, the revised paper is significantly easier to read and conveys the message much more smoothly.

Besides this overall concern, I have the following specific notes:

1) I find it confusing to refer to p as the polarization, given that we are considering nonlinear response, e.g., powers $p_{\{x,y\}}^n$ for $n > 1$. More accurately, this is the incoming (weak) electric field for which the x and y components are treated as independent scalars.

We have changed p to q throughout the paper, to avoid this confusion.

2) Calling the method non-perturbative is confusing, given that it is precisely a $k + l + h + j$ order perturbation in p which is considered for a given component. If the idea is that one would be able to make a general constraint on response as a function of symmetries of p without direct reference to each term in the Taylor series, that argument should be made more clearly.

Thank you. This is precisely our claim – the constraints imposed by the real synthetic symmetry are non-perturbative, and do not depend on a particular perturbative expansion.

We make this claim more clearly in the revised manuscript:

"

The condition $E_{HHG}(t, Q) = \hat{X}E_{HHG}(t, \hat{\zeta}(Q))$ is a non-perturbative restriction (selection rule) on the response of the dipolar emission of the system as the symmetry is broken, and in principle, it is valid beyond the radius of convergence of a particular perturbative expansion. However, for practical application of these rules, it is instructive to reformulate them as selection rules on the expansion coefficients of a perturbative expansion. We emphasize that we do not employ perturbation theory, but rather reformulate non-perturbative selection rules in a perturbative language. To do this, we write...

"

3) In Eq. 4 of the main text, it is confusing to call the input field E because this has been used for the output electric field throughout the paper, including in deriving the selection rules throughout the supplement.

We have changed the incoming field in the manuscript and SI from E to F .

4) I'm not a fan of the term "synthetic dimensions" in referring to the polarization degrees of freedom, as they do not represent dynamical degrees of freedom. Instead, I would just say "polarization degrees of freedom" or, more generally, "parametric degrees of freedom" to capture the (unproven) statement that in principle you could also have symmetries related to the spin-orbit coupling parameters.

Parametric degrees of freedom are widely referred to as synthetic dimensions in the literature. Of course, this is a matter of perspective and taste. However, we feel that the terminology is well placed here, because the parametric degrees of freedom facilitate the observation of higher dimensional physics, i.e., symmetries in higher dimensions. To put this in context, we would like to

refer your attention to the recent review by Eran Lustig and Mordechai Segev, entitled "Topological photonics in synthetic dimensions" (Adv. Opt. Photon. 13, 426-461, 2021). In this review, the literature usage of the term "synthetic dimensions" was neatly organized into three categories, the first of which are parametric degrees of freedom:

"

... As mentioned earlier, the non-spatial degrees of freedom can take several forms in photonics. Currently, the implementation of synthetic dimensions in photonics can be cast into three main classes. The first class, which may be called "parametric synthetic dimensions," is using a parameter of the Hamiltonian, which, when varied (either continuously or in discrete steps), can constitute a "parametric synthetic dimension." However, if the parameter is constant or is varied adiabatically—it lacks a kinetic term. The absence of a kinetic term means that there is no transport in this parametric synthetic dimension. Such systems are simpler to implement compared to other approaches, and they do allow for observing higher dimensional physics in lower dimensional systems, which made them a useful tool. For example, by mapping a 1D lattice to a slice in a 2D model, topological Thouless pumping was demonstrated in photonics experiments [61], and more recently higher dimensional topological pumps were demonstrated in both optics and cold atoms [72,73]. In fact, a recent theoretical proposal suggested systems simulating 6D topological pumps [74]. Other aspects related to topology, such as Fermi arcs and Weyl points, were also demonstrated by exploiting parameter space [75]...

...The second class of non-spatial degrees of freedom may be called "eigenstates ladder." It does have a kinetic term in the Hamiltonian that results in propagation along the synthetic dimension...

...Finally, the third approach for introducing a synthetic dimension has to do with discrete dynamics in a synthetic space of "time-bins"...

"

We feel that our work naturally fits in the 1st category and captured the spirit of the term "synthetic dimensions". The polarization vector/ other parameter facilitates the realization of high dimensional dynamical symmetries through tunable parameters.

Regarding the spin-orbit coupling, we show explicitly in section V of the revised SI that spin-orbit coupling strengths impose real synthetic symmetries and corresponding selection rules on HHG.

Reviewer #2 (Remarks to the Author):

In this work, the authors propose the concept of real-synthetic symmetries as a tool to derive selection rules in strong field phenomena (such as high-harmonic generation and ionization). For a system with a certain symmetry that is subject to a symmetry breaking perturbation, a real-synthetic symmetry is a operation formed by composing the original symmetry, with an additional operation that acts on the perturbation degrees of freedom, such that the perturbed system remains invariant. Existence of these real-synthetic symmetries constrains the physical response, imposing selection rules. The method is non-perturbative in nature, going beyond derivation of selection rules based on low order perturbation theories.

The authors construct (in the supplementary material) the real-synthetic symmetry operations for a series of different symmetries and table corresponding selection rules for both high-harmonic generation and above-threshold ionization.

The authors then apply the general method to high-harmonic generation for a bi-chromatic field and contrast theoretical predictions based on real-synthetic selection rules with experimental results, obtaining excellent agreement. Importantly, a quartic response (which would be hard to obtain with perturbation theory) to a symmetry breaking term is predicted and observed experimentally.

The paper is very interesting and the notion of non-perturbative selection rules based on real-synthetic symmetries will most likely attract attention and be useful to the strong-field physics community.

However, I believe the paper has some problems with the presentation that should be addressed. I list then in the following:

1) In page 6 before equation 4, it is stated that a bi-chromatic field with incommensurate frequencies is considered. The two frequencies ω and 1.95ω . Since $1.95 = 39/20$, the two frequencies are actually commensurate and the field in Eq. (4) is periodic in time, with a period of $20 \cdot 2\pi/\omega$.

Thank you. The word "incommensurate" was deleted from the manuscript.

2) The definition of the time translations is confusing. Is $T = 2\pi/\omega$, the period of the ω -field and $T' = 20T = 20 \cdot 2\pi/\omega$ the period of the field in Eq. (4)? This is never clearly stated, which obfuscates the whole presentation.

Yes, $T = 2\pi/\omega$, the period of the ω -field and $T' = 20T = 20 \cdot 2\pi/\omega$ the period of the field in Eq. (4). However, in the revised version of the manuscript we avoid reformulating the problem in terms of ω' and T' , because we understand that it was confusing.

3) It is said that the σ_x operator is a "reflection relative to the Cartesian basis vector \hat{x} ". Does this mean a $x \rightarrow -x$ exchange (mirror symmetry with respect to the $y-z$ plane)?

All of the Floquet group theory discussed in the paper are in $(2+1)D$, so the symmetries are defined in the xy plane. The operation $\hat{\sigma}_i$ is a reflection relative

to the \hat{t} vector. This means that $\hat{\sigma}_x$ is the operation $y \rightarrow -y$ and $\hat{\sigma}_y$ is the operation $x \rightarrow -x$. This is now written clearly in the revised manuscript, where we describe all building blocks of Floquet group theory (in the first paragraph of the subsection “General theory”).

4) In the main text, $C_{\{59, 20\}}$ is said to represent a $2\pi \cdot 20/59$ rotation composed by a time translation by $T/20$. However, in the SM the symmetry $C_{\{N, M\}}$ is defined as a $2\pi \cdot M/N$ rotation composed with a time-translation by the period divided by N . These two definitions seem to be at odds. Should the $C_{\{59, 20\}}$ symmetry of the main text actually be a $C_{\{20, 59\}}$?

The $C_{\{N,M\}}$ symmetry is defined as a T/N time translation and a $2\pi \cdot M/N$ spatial rotation. According to this definition, the operation in the main text should in fact be $C'_{\{59,20\}}$.

However, this formulation was confusing, and in the revised paper we refrain from using it. In the revised paper, we identify the symmetry as $\hat{C}_{2.95}$, which is a time translation by $T/2.95=T/59$, and a rotation by $2\pi/2.95=20 \cdot 2\pi/59$.

5) In table I, besides the HHG selection rules, selection rules for above-threshold ionization are also listed. However, the phi coefficients are only defined in the SM. I believe that either the phi coefficients are defined in the main text or, table I should only list the HHG selection rules.

The discussion of ATI was moved to the SI. Table 1 in the original manuscript became Table 2 in the revised manuscript. Table 2 is now focused on HHG, and does not include the phi coefficients.

6) In order to improve clarity of the method, more of the details in Section VI of the supplementary material should be provided in the main text.

This section was moved to the main text, in order to improve the clarity of the paper and make it easier to grasp.

7) In the supplementary material, below Eq. (II.13) it is said “is only nonzero if n and $(a + c)$ are of the same parity”, but c was previously set to 0. So what does this statement mean?

Setting c to zero was not actually necessary. It was set to zero because p was real. However, the series expansion we employ is consistent as it is even for real valued p . This only implies that the expansion coefficients have the symmetry $a \leftrightarrow c$. Hence, we removed the constraint $c=0$. The revised SI reads:

To illustrate what this restriction means physically, we consider an example. When ATI is driven by an $\omega - 2\omega$ cross linear driving field,

$$E(t) = \begin{pmatrix} \sin(\omega t) \\ \cos(2\omega t) \end{pmatrix}$$

the system exhibits \hat{Z}_y DS. Thus, along the \hat{y} axis (i.e. when the detector is situated on the polarization axis of the 2ω pump field), only even order photoelectron peaks are allowed⁶. If the system is perturbed by the field $\lambda \cos(2\omega t) \hat{x}$, the symmetry is broken and odd order

photoelectron peaks are generated along the \hat{y} axis. We plug in $s = 2, b = d = 0$, and obtain the following restriction on the photoemission spectrum

$$\tilde{\Phi}_{\alpha, k_n}^{(a0c0)} = \tilde{\Phi}_{\alpha, \hat{\sigma}_y k_n}^{(a0c0)} (-1)^{n+a+c}$$

For electrons whose momentum is parallel to the \hat{y} axis (perpendicular to the perturbation),

$$k_n = \hat{\sigma}_y k_n$$

and thus

$$\tilde{\Phi}_{\alpha, k_n}^{(a0c0)} = \tilde{\Phi}_{\alpha, k_n}^{(a0c0)} (-1)^{n+a+c}$$

i.e. $\tilde{\Phi}_{\alpha, k_n}^{(abcd)}$ is only nonzero if n and $(a + c)$ are of the same parity. Thus, along the \hat{y} axis, even ordered photoelectron peaks scale as even orders of the perturbation's amplitude and odd order photoelectron peaks scale as odd powers of the perturbation's amplitude.

“

8) In page 18 of SM, I assume that T symmetry is time-reversal symmetry. Is this correct?

Yes, this is correct. In the revised manuscript we explicitly define all symmetry operations in the main text.

9) In Section II of the SM, symmetries D and H are mentioned (but not defined). Are these the same as D_y and H_y of Section I?

Thank you. This is correct. We have proofread and double checked the SI and corrected this error. The subscript “y” is now included.

10) In Section IV of the SM, details on the numerical solution of the time dependent Schrodinger equation are provided. It is stated that “The high harmonic generation (HHG) spectra for the different cases analyzed in the paper were obtained by numerically solving the time dependent Schrodinger equation (TDSE) for an atom irradiated by a laser field, in the length gauge”. Does this refer to the results shown in Fig. III.1? This should be made more explicit.

We added the following sentence to section I of the revised SI:

“

In this section we describe the numerical procedure for the calculation of the HHG spectra presented in in Figure 2 of the main text.

“

11) In Section V of the SM, which discusses real-synthetic symmetries in a spin-orbit coupled system, it is said that S₄ is a 2pi/4 rotation in “synthetic spin space”. However, from Eq. (V.5) it seems to be a rotation of the Pauli matrices and therefore a rotation of the usual spin degrees of freedom. Why is it then referred to as a synthetic spin space? Is it synthetic in the same sense of “real-synthetic symmetry”?

Thank you. We have proofread the SI and made numerous corrections. This terminology is inaccurate and was removed from this sections. The synthetic dimension in this example is the spin-orbit coupling strength γ .

Provided these issues are addressed, I believe this work warrants publication in

Nature Communications.

Finally, I have question that I would like for the authors to answer. The whole concept of real-synthetic symmetries, appears to rely on the fact that the frequency of the perturbation is commensurate with the frequency of the driving field of the “unperturbed system”, such that the perturbed system is still periodic in time and Floquet theorem can be used. What if this is not the case and the frequency of the perturbation is incommensurate? Could approximate selection rules be obtained by studying commensurate approximants? Could the approximate selection rules be improved by considering higher order (longer period) commensurate approximants? Commensurate approximants are employed in the study of electronic system subject to quasi-periodic perturbations and I wonder if the same could be made in the context of perturbed Floquet systems.

This is true that the Floquet group classification of real-synthetic symmetries, presented in our manuscript, is reliant on the fact that the frequencies are commensurate. This is a general requirement of any Floquet theory. However, incommensurate frequencies can be handled by Floquet theories in one of two ways (that we are aware of) :

- 1. The first option is exactly what you suggested – "round off" the incommensurate frequencies and obtain a commensurate approximation, with a long period. Reformulate your system as a Floquet system with the common frequency as the fundamental frequency, and employ the Floquet theory of your choosing – Floquet group theory, Floquet perturbation theory, real synthetic symmetries, etc. To estimate a priori the accuracy of this approximation is a challenging task. One has to estimate how many cycles of the long period are "enough" for the system to be considered truly periodic. The answer will probably depend on the quasi energy and different parameters of the system. One option is that several cycles of the common periodic are needed, another is that a few cycle of each incommensurate frequency are enough. Depending on the specifics of the problem, anything in between is also possible. The easiest approach for practical applications is just to try the approximation and compare it to the experiment/ simulation.**
- 2. There is an exact approach that does not rely on a commensurate approximation. This approach will appear in a separate publication from our group. A system subject to incommensurate drives, although not periodic in real space time, is periodic in an extended space with two temporal dimensions, and further, conclusions about the real space dynamics can be drawn from the physics in this extended "fake" space. Explicitly, if the temporal dependence of the system is given by**

$$e^{iet} + e^{i\pi t}$$

One can reformulate the problem in a space with two temporal dimensions (t1,t2), so that the temporal dependence of the system becomes

$$e^{iet_1} + e^{i\pi t_2}$$

In this extended space, the system is periodic with a vector period of $(\frac{2\pi}{e}, \frac{2\pi}{\pi})$, and all Floquet theories can be consistently employed. Finally, after all desired analytical conclusions about the extended space wave function $|\psi(r, t_1, t_2)\rangle$ one can take $t_1 = t_2$ and obtain the physical wave function $|\psi(r, t_1)\rangle = |\psi(r, t_1, t_1)\rangle$. Adding temporal dimensions and tracing them out after the solution was first done in context of standard Floquet systems (J. Chem. Phys. 99, 4590 (1993); <https://doi.org/10.1063/1.466058>). The extension to incommensurate frequencies, which we described here in short, will appear in a separate publication from our group.

Finally, we wish to emphasize that the concept of real-synthetic symmetries and their selection rules is NOT limited to Floquet systems, and one can employ the concept developed in our work to other fields of physics, e.g., solid state physics.

Reviewer #3 (Remarks to the Author):

In this manuscript, the authors present a novel and valuable group-theoretical framework for handling symmetry-breaking perturbations to a system that obeys a dynamical symmetry of Floquet type. The authors' framework unifies a number of existing and (currently) independent and disparate approaches, clarifies which aspects are universal and which aspects are specific to perturbation-theory approaches, and should provide a solid platform on which to build future work, particularly regarding symmetry-breaking spectroscopies.

The core idea of the framework is that, in general, if one has a system which is originally symmetric and then has its symmetry broken, the symmetry-breaking perturbation generally has one (or more) controlling parameters; and, moreover, the broken symmetry operation can typically be restored by a suitable transformation on that controlling parameter. By treating this controlling parameter as a 'synthetic' dimension, the authors are able to provide a full symmetry to the (expanded) system, which provides clean and universal selection rules for how an experimental observable can depend on the symmetry-breaking parameter.

The results presented in this paper are extremely strong and valuable, and they definitely merit publication in Nature Communications. However, the presentation of the results requires significant work before the paper can be accessible to the broad audience of this journal (and, in particular, to the audiences that stand to benefit from the authors' results). As such, I feel that major revisions are required to the manuscript before it can be published.

In short, I found the text extremely hard to read, mostly due to the structure of the presentation but also due to some of the choices regarding the details of the exposition. The presentation is extremely mathematical and dry, with very little that the reader can use to hang on to and create an intuitive picture of the material. This can only have the effect of losing most readers early on, in such a way that the manuscript fails to communicate its core message to its intended audience. Because of this, I would recommend that the authors re-build the manuscript from the ground up, with a didactical approach in mind.

To be frank, I feel that what is required is that the authors explain the framework in detail, in person, to a small number (say, five?) of masters- or upper-undergraduate-level students, take the strategies that work in those explanations, and use them as the backbone to restructure the paper. It is hard to give more concrete advice, but I can provide the following major and minor comments:

Major comments

1. The core idea of the framework (which I have tried to enunciate in a compact but clear fashion in the second paragraph of this report) **is very hard to glean from the abstract and introduction of the paper**, and this makes it extremely hard to get started with the paper.

Thank you. We have significantly revised the abstract, introduction, and the structure of the paper as a whole. The manuscript now begins with the

concept, moves to a simple numerical example (“tutorial example” in the original MS), then moves on to the experimental observation, and finally, to the general theory.

2. The text clearly needs a clear example that the reader can use as a test case to which the full-blown formalism can be applied: a concrete, manipulatable, visualizable instance that can be used to understand the (very) abstract general formalism. This should ideally be presented *before* the general formalism, so the overall ideas can be presented in a more intuitive way, and the abstractions of the general formalism can then be overlaid on the models built using the more intuitive concrete instance.

The authors already have two options of specific cases that can be used for this initial instance: the bi-elliptical HHG configuration in the second half of the current main text, and the 'tutorial example' that's currently buried in §III of the supplement. I feel that the latter is a better choice, as the geometry is simpler and the point is more easily conveyed.

We have included the “tutorial example” in the body of the main text. This is followed with the experiment, and only afterwards, the full-blown formalism is presented with the table and the general power series expansion.

3. The mathematical presentation requires more attention to detail, and the notation feels, at times, inconsistent. In particular, in equations such as $W = X \cdot \zeta_X W$, it is not clear whether X and ζ are operators that are *multiplying* W , or super-operators that are acting on it. The notation as written suggests the former, but the formalism as it's actually employed implies the latter. The authors should make sure that their notation is crystal clear in separating these two aspects.

The equation is now rewritten as $\widehat{W} = (\widehat{X} \cdot \widehat{\zeta}_X)^\dagger \widehat{W} (\widehat{X} \cdot \widehat{\zeta}_X) \equiv \widehat{X} \cdot \widehat{\zeta}_X [\widehat{W}]$ throughout the manuscript and the supplementary information, and the operators $\widehat{\zeta}$ and \widehat{X} are acting on \widehat{W} (transforming it). We have added explicit examples of this in the revised manuscript, as well as the following clarifying paragraph:

“

We consider the \widehat{X} symmetric Floquet system to be perturbed by a perturbation \widehat{W} :

$$\widehat{H} = \widehat{H}_0 + \widehat{W} \quad (1)$$

\widehat{W} breaks the symmetry \widehat{X} such that $\widehat{X}^\dagger \widehat{H}_0 \widehat{X} = \widehat{H}_0$, but $\widehat{X}^\dagger \widehat{H} \widehat{X} \neq \widehat{H}$. Although \widehat{X} is broken, it may still be exploited to formulate a symmetry of the form $\widehat{X} \cdot \widehat{\zeta}_X$ in the symmetry broken system, where $\widehat{\zeta}_X$ operates on the internal degrees of freedom of \widehat{W} , while leaving \widehat{H}_0 unaffected. The operation $\widehat{\zeta}_X$ are derived by solving the equation $\widehat{W} = (\widehat{X} \cdot \widehat{\zeta}_X)^\dagger \widehat{W} (\widehat{X} \cdot \widehat{\zeta}_X) \equiv \widehat{X} \cdot \widehat{\zeta}_X [\widehat{W}]$, where the square brackets indicate that the composite operation $\widehat{X} \cdot \widehat{\zeta}_X$ transforms the operator \widehat{W} . In the examples above, Q is a vector containing the complex polarization components of the symmetry breaking fields, which is acted on by $\widehat{\zeta}$. The selection rule for the optical emission is obtained by employing the invariance of the emission under the symmetry operation, that is $E_{HHG}(t, Q) = \widehat{X} \cdot$

$$\widehat{\zeta}_X E_{HHG}(t, Q) = \widehat{X} E_{HHG}(t, \widehat{\zeta}[Q]).$$

We now focus on the case where \widehat{W} represents an additional laser whose amplitude and polarization are given by the complex vector $Q = (q_x, q_y)$

$$\widehat{W} = \Re\{Q \cdot r e^{is\omega t}\} \quad (2)$$

Here, ω is the fundamental frequency of the symmetric Floquet system and $s\omega = 2\pi s/T$ is the frequency of the symmetry breaking field \widehat{W} . Since \widehat{X} is a symmetry of \widehat{H}_0 , and $\widehat{\zeta}_{\widehat{X}}$ only operates on $\widehat{W}(Q)$ by definition, the symmetry condition is

$$\widehat{W} = \Re\{\widehat{\zeta}_{\widehat{X}}[Q] \cdot \widehat{X}[r e^{is\omega t}]\} \quad (3)$$

For example, if $\widehat{X} = \widehat{T}$, this equation becomes $\Re\{Q \cdot r e^{is\omega t}\} = \Re\{\widehat{\zeta}_{\widehat{T}}[Q] \cdot r e^{-is\omega t}\}$, which is fulfilled by the complex conjugation operation $\widehat{\zeta}_{\widehat{T}}[Q] = \bar{Q}$. If $\widehat{X} = \widehat{Z}_y$ ($\{x \rightarrow -x, t \rightarrow T/2\}$), Eq.(9) reads $\Re\{Q \cdot r e^{is\omega t}\} = \Re\{\widehat{\zeta}_{\widehat{Z}_y}[Q] \cdot [(-1)^s \widehat{\sigma}_x r e^{is\omega t}]\}$, which is solved by $\widehat{\zeta}_{\widehat{Z}_y} = (-1)^s \widehat{\sigma}_y^{(Q)}$, where the Q superscript indicates that $\widehat{\sigma}_y^{(Q)}$ operates in the synthetic Q space ($\{q_x \rightarrow -q_x\}$). Table 2 shows operations $\widehat{\zeta}_{\widehat{X}}$ that solve Eq.(9) for all other Floquet group theory9 symmetries \widehat{X} (derived in section II of the SI).

“

4. The main text of this paper must be self-contained: in particular, it cannot rely on references to external papers (and specifically to the authors' previous work in ref. [8]) for definitions of any symbols or notations that it employs. This is particularly noticeable in Table 1, which in its current state just lists multiple undefined symbols (symmetry operations) and is therefore not actionable.

In the revised manuscript, we have made the text as self-contained as possible, including explicit walkthroughs of the derivation of the symmetries and selection rules. In addition, we have included the definitions of all symmetry operations in a separate table in the main text (Table 1 in the revised manuscript). All building blocks of Floquet group theory were explicitly laid out in the revised text:

“

A Floquet system $\widehat{H}_0(t) = \widehat{H}_0(t + T)$ exhibits the DS \widehat{X} if $[\widehat{\mathcal{H}}_f, \widehat{X}] = 0$ where $\widehat{\mathcal{H}}_f \equiv \widehat{H}_0 - i\partial_t$ is the Floquet Hamiltonian. The operation \widehat{X} is a (2+1)D spatio-temporal symmetry, jointly imposed by the symmetries of the target material and a first driving laser (or by any other periodic excitation of the system^{30,31}). The operations \widehat{X} were comprehensively tabulated within the framework of Floquet group theory9, and for completeness, are given with their corresponding HHG selection rules in Table 1. In Table 1, \widehat{T} is the time-reversal operation ($t \rightarrow -t$), \widehat{R}_N are $2\pi/N$ spatial rotations, $\widehat{\tau}_N$ are T/N time translations, $\widehat{\sigma}_i$ is a reflection relative to the vector \hat{i} , and \widehat{L}_b is the scaling operation $\hat{y} \rightarrow b\hat{y}$.

“

As an offshoot of this: the symbols corresponding to ATI (and particularly

$\tilde{\phi}_{\mathbf{k},n}(\mathbf{abcd})$ *must* be defined in the main text if the corresponding selection rules are reported in the main text. I personally feel that they are not really required, and that it is perfectly reasonable for the authors to move the entire discussion of ATI to the supplement (or to a separate paper) and keep the discussion of the main text focused on harmonic generation.

We have moved all of the discussion of ATI to the supplementary material file.

5. Many of the revisions suggested in this report imply an increase in the length of the main text -- sometimes significantly so. If the authors feel that there is some overall need for brevity, I would suggest a thorough reconsideration. There is no requirement to fit the text in under a fixed number of pages, and clarity is a very high price to pay for conciseness.

Our initial constraint was indeed brevity. We increased the length of the paper, allowing for smaller steps to be taken in the presentation, and providing additional examples & illustrations in the body of the text.

6. The manuscript claims that the authors' framework provides "system-independent scaling laws [...] valid to all orders in the strength of the perturbation". I disagree with the way the results are being represented here. The authors' framework provides *selection rules* -- sometimes quite restrictive ones -- on the Taylor series of the observables with respect to the strength of the perturbation, but the analysis cannot be used to conclude that a single term suffices within that Taylor series. It will typically be the case that the leading term will dominate at low perturbation strength, but that is obviously contingent on the validity of the relevant order of perturbation theory. In general, multiple terms can coexist (say, a linear term and a cubic one), providing a nontrivial shape to the response which cannot be predicted from the authors' formalism.

Thank you. This is indeed correct. We have deleted the phrase “..system independent scaling laws...” from the manuscript.

7. In the formal presentation of the results (the paragraph around eq. (1)), I found it confusing whether the presentation was for *generic* configurations or for some specific case (of laser-driven dynamics). Does the formalism presented here apply to the spin-orbit coupling mentioned shortly afterwards? If it doesn't, then shouldn't the formalism be reformulated to an even more abstract one that does cover all of the intended cases of applicability? (note that often an increase in abstraction can actually increase clarity.) This complete-generality case can then be further narrowed to the given form of W for the specific case of laser-driven systems as required.

Thank you. The discussion you refer to (around Eq.(1)) was moved to the last section of the paper, after the tutorial example and the experiment. Now, it is under the subsection “Floquet group theory classification”. This subsection starts with a general discussion without restricting the form of \widehat{W} (e.g. it includes the spin-orbit case). After describing how the symmetry is detected and the selection rule is derived, we narrow the discussion to a laser driven dynamics in Eq.(9).

8. In the form of \widehat{W} presented in eq. (1), it's unclear what happens if the presentation contains more than one colour. Does s play any important role in the resulting calculations? (if so, it wasn't immediately obvious to me, perhaps due to the lack of actionable examples where this could be seen in action.) If not, then shouldn't \widehat{W} be replaced by a more arbitrary and general form?

This is particularly important because the main example, in the second half of the manuscript, *does* contain a polychromatic perturbation, which is introduced without accounting for what extra difficulties that implies.

When more than two colors are included in \widehat{W} , the dimensionality of the synthetic space is increased, so that $\hat{\zeta}$ can operate on each of the colors separately. We have made several revisions to the original manuscript, so that this point would be clear.

Firstly, in the revised MS, the experiment is presented prior to the general formalism.

In the current presentation of the experiment, we explicitly address the two colors by formulating a synthetic dimensions operation that separately acts on the complex amplitudes of each of the colors. This is also illustrated in Figure 3, where it is shown step by step how a real-synthetic operation acts on each of the 4 components of the driving field (RCP ω , LCP 1.95ω , LCP ω , RCP 1.95ω). The components LCP ω , RCP 1.95ω are acting as the perturbation, and are phase shifted by a different amount by the synthetic dimensions operation.

We have also addressed this explicitly in the section about the general formalism.

“

The rules presented in Table 2 are consistent with the numerical example presented above where the perturbation is monochromatic (\widehat{C}_2 symmetry breaking). We emphasize that the conditions of our experiment involve a bi-chromatic symmetry breaking perturbation. We have seen explicitly that in that case the bi-chromatic perturbation implies that the synthetic operations act in a higher dimensional space, transforming each color of the perturbation separately. Then, the selection rules are derived in the same manner but with a more elaborate series expansion. In the general case of a laser with two colors, we may write

$$\widehat{W} = \Re\{Q_1 \cdot re^{is_1\omega t} + Q_2 \cdot re^{is_2\omega t}\} \quad (4)$$

where $s_{1,2}$ determine the color of each perturbation, and $Q_{1,2}$ are their complex amplitudes. Now, the parameter space that defines \widehat{W} is given by $\{Q_1, Q_2\}$, and the synthetic dimensions operation is given by $\hat{\zeta} = \hat{\zeta}_1 \cdot \hat{\zeta}_2$ where $\hat{\zeta}_i$ operates

only on Q_i ($i=1,2$). Here, $\hat{\zeta}_{1,2}$ are the operations tabulated in Table 2 corresponding to $s_{1,2}$ respectively. The corresponding selection rule for the emission is given by $E_{HHG}(t, Q_1, Q_2) = E_{HHG}(t, \hat{\zeta}_1(Q_1), \hat{\zeta}_2(Q_2))$, which can be translated to selection rules on the coefficients of a series expansion, in a manner identical to the one presented above. The process of concatenating the symmetry operations tabulated in Table 2 and deriving the corresponding selection rules is not limited to bi-chromatic perturbations and one may directly extend it to obtain the real-synthetic symmetries and selection rules associated with a polychromatic perturbation.

“

9. Where does the equation $W = X \cdot \zeta_X W$ come from, and what does it actually mean? Perhaps my confusion is mostly just caused by unclear notation, but this key touchstone of the formalism is introduced without much explanation of its origin and its significance.

Firstly, allow us to state in words what the equation $W = X \cdot \zeta_X W$ really means. This equation says that \widehat{W} is invariant when transformed by the operation $\widehat{X} \cdot \hat{\zeta}_X$. The operation $\widehat{X} \cdot \hat{\zeta}_X$ operates on \widehat{W} (it does not multiply it, it operates on it). The reason that the solution to this equation is a symmetry of the perturbed symmetry, is because \widehat{X} is a symmetry of \widehat{H}_0 , and $\hat{\zeta}_X$ only operates on the parameters of \widehat{W} .

In more detail:

We begin with a symmetric Hamiltonian with a symmetry \widehat{X} so that $\widehat{X}^+ \widehat{H}_0 \widehat{X} = \widehat{H}_0$.

We break the symmetry of this Hamiltonian by adding a general perturbation \widehat{W} that does not comply with the symmetry condition, $\widehat{X}^+ \widehat{W} \widehat{X} \neq \widehat{W}$.

However, there is still a broken symmetry in the system. This is a resource that can be exploited. To exploit the broken symmetry, we seek symmetries of the form $\widehat{X} \cdot \hat{\zeta}_X$ in the perturbed system. Here, $\hat{\zeta}_X$ only operates on the internal parameters of the perturbation \widehat{W} , leaving \widehat{H}_0 unaffected.

Mathematically, this translates to the condition $\widehat{X} \cdot \hat{\zeta}_X[\widehat{H}] = \widehat{H}$ (where we added the square brackets $[\]$ to emphasize that $\widehat{X} \cdot \hat{\zeta}_X$ acts on \widehat{H}):

$$\widehat{H} = \widehat{H}_0 + \widehat{W}$$

$$\widehat{X} \cdot \hat{\zeta}_X[\widehat{H}] = \widehat{X} \cdot \hat{\zeta}_X[\widehat{H}_0 + \widehat{W}] = \widehat{X}[\widehat{H}_0] + \widehat{X} \cdot \hat{\zeta}_X[\widehat{W}] = \widehat{H}_0 + \widehat{X} \cdot \hat{\zeta}_X[\widehat{W}]$$

So, in order for $\widehat{X} \cdot \hat{\zeta}_X$ to be a symmetry of the system, $\widehat{X} \cdot \hat{\zeta}_X[\widehat{W}] = \widehat{W}$.

A symmetry of this form is not guaranteed to exist for every \widehat{W} , but for the case where \widehat{W} represents a laser field, it is guaranteed. This is a key result of our explicit classification (Table 1 in the original manuscript, and Table 2 in the revised manuscript).

We have added the following paragraph to the manuscript:

“

We consider the \hat{X} symmetric Floquet system to be perturbed by a perturbation \hat{W} :

$$\hat{H} = \hat{H}_0 + \hat{W} \quad (5)$$

\hat{W} breaks the symmetry \hat{X} such that $\hat{X}^\dagger \hat{H}_0 \hat{X} = \hat{H}_0$, but $\hat{X}^\dagger \hat{H} \hat{X} \neq \hat{H}$. Although \hat{X} is broken, it may still be exploited to formulate a symmetry of the form $\hat{X} \cdot \hat{\zeta}_X$ in the symmetry broken system, where $\hat{\zeta}_X$ operates on the internal degrees of freedom of \hat{W} , while leaving \hat{H}_0 unaffected. The operation $\hat{\zeta}_X$ are derived by solving the equation $\hat{W} = (\hat{X} \cdot \hat{\zeta}_X)^\dagger \hat{W} (\hat{X} \cdot \hat{\zeta}_X) \equiv \hat{X} \cdot \hat{\zeta}_X [\hat{W}]$, where the square brackets indicate that the composite operation $\hat{X} \cdot \hat{\zeta}_X$ transforms the operator \hat{W} . In the examples above, Q is a vector containing the complex polarization components of the symmetry breaking fields, which is acted on by $\hat{\zeta}$. The selection rule for the optical emission is obtained by employing the invariance of the emission under the symmetry operation, that is $E_{HHG}(t, Q) = \hat{X} \cdot \hat{\zeta}_X E_{HHG}(t, Q) = \hat{X} E_{HHG}(t, \hat{\zeta}[Q])$.

“

10. On a similar note, the authors state "The operation ζ_X may be systematically derived by solving the equation $W = X \cdot \zeta_X W$ " ... but they never really explain how that equation can be solved. I found the given examples insufficient to understand the procedure and what's really at stake here, both for practical matters (how is this done in practice?) as well as more theoretical aspects (is the solution always guaranteed? is it unique?).

To find $\hat{\zeta}_X$, we first operate with $\hat{X} \cdot \hat{\zeta}_X$ on the perturbation (where $\hat{\zeta}_X$ is still unknown). Then we equate the transformed \hat{W} to the original \hat{W} and solve the equation. To simply illustrate this, we included in the manuscript the solution for $\hat{\zeta}_T$ and $\hat{\zeta}_{Z_y}$ in the case where \hat{W} represents a perturbing laser:

“

Because \hat{X} is a symmetry of \hat{H}_0 , and $\hat{\zeta}_{\hat{X}}$ only operates on $\hat{W}(Q)$ by definition, the symmetry condition is

$$\hat{W} = \Re\{\hat{\zeta}_{\hat{X}}[Q] \cdot \hat{X}[re^{is\omega t}]\} \quad (6)$$

For example, if $\hat{X} = \hat{T}$, this equation becomes $\Re\{Q \cdot re^{is\omega t}\} = \Re\{\hat{\zeta}_{\hat{T}}[Q] \cdot re^{-is\omega t}\}$, which is fulfilled by the complex conjugation operation $\hat{\zeta}_{\hat{T}}[Q] = \bar{Q}$. If $\hat{X} = \hat{Z}_y$ ($\{x \rightarrow -x, t \rightarrow T/2\}$), equation 10 reads $\Re\{Q \cdot re^{is\omega t}\} = \Re\{\hat{\zeta}_{\hat{X}}[Q] \cdot [(-1)^s \hat{\sigma}_x re^{is\omega t}]\}$. This equation is solved by $\hat{\zeta}_{\hat{Z}_y} = (-1)^s \hat{\sigma}_y^{(Q)}$ where the Q superscript indicates that $\hat{\sigma}_y^{(Q)}$ it operates in the synthetic Q space ($\{q_x \rightarrow -q_x\}$). Table 2 shows operations $\hat{\zeta}_{\hat{X}}$ that solve Eq.(10) for all other Floquet group theory8 symmetries \hat{X} .

“

Regarding the existence and uniqueness of a solution, that depends on the specific \hat{W} and \hat{X} . In the case where \hat{W} represents a monochromatic laser field, a unique solution exists for any \hat{X} as shown by our explicit Floquet group classification. We have no guarantee that a unique solution (or any solution) will always exist for any \hat{W} , or that the solution that one finds will always result in an insightful, non-trivial conclusion about the response of the system. But, in our experience, this is most often the case.

11. The role played by the specific form of the harmonic response (and particularly how it transforms under \hat{X}) is unclear in the current text. What happens with other observables?

The optical emission transforms as the dipole moment expectation value. In the SI, we show explicitly how they are transformed under each one of the operations \hat{X} . However, in order to make things clearer, we have specified explicitly that it transforms as the dipole moment expectation value (that is , it transforms as the position operator). We also provide an example of an

operator that transforms in a different way (the squared x-axis position x^2) and contrast them.

We have added the following text in the revised manuscript:

“

the selection rule for the emitted harmonic light (denoted by $E_{HHG}(t, Q)$) can be obtained using the invariance of a time dependent observable under the symmetry operation⁹, i.e. $E_{HHG}(t, Q) = \hat{X}E_{HHG}(t, \hat{\zeta}[Q])$. Notably, this equation also holds for other observables, i.e. $o(t, Q) = \hat{X}o(t, \hat{\zeta}[Q])$ for a general $o(t)$. However, E_{HHG} and o may transform differently under \hat{X} and therefore adhere to different selection rules. . For example, $E_{HHG}(t, Q)$ transforms as the dipole moment expectation value hence it changes sign under \hat{R}_2 , whereas the expectation value for squared x-axis position $x^2(t, Q)$ does not. Detailed examples of the transformation of the E_{HHG} under \hat{X} are given in the SI and in Ref9.

“

12. It is unclear to me why the authors chose to represent the bielliptical fields used in the second half of the paper as a $w:1.95w$ frequency relation. The detuning of the second harmonic by 2.5% was a key component of the original experiment on bicircular fields (ref. 11) but it does not really add anything here, since the spectrum (fig. 1(b)) is never separated into individual sub-channels in the way done in ref. 11. The cost of doing this is a significant price in terms of the clarity of the presentation, for very little real gain. The presentation in the main text should use a $w:2w$ frequency ratio, and avail itself of all of the corresponding gains in simplicity of exposition. If required, the authors can mention that the experiment (for practical experimental reasons?) used a slightly different ratio, and perhaps present the analysis using the $w:1.95w$ ratio in the supplement. But I see no reason for the current choices.

The approximation you suggest (approximating the $\omega - 1.95\omega$ beam as an $\omega - 2\omega$ beam), can only work for a subset of the results. The selection rules that correspond to the broken reflection symmetry (i.e., describe the scaling around $\theta = 0^\circ$) can remain the same if this approximation is made. However, the selection associated with the broken rotational symmetry (i.e., describing the scaling around $\theta = 45^\circ$) cannot be consistently obtained from such an approximation. In fact, within such an approximation, all harmonic orders should have linear and quadratic contributions.

More explicitly, within the $\omega - 2\omega$ approximation, the spectral component $n\omega$ are allowed to have a linearly scaling component with the deviation from 45° if

$$n = 3q \pm (2 \pm 1)$$

and a quadratic component if

$$n = 3q \pm (2 \pm 1) \pm (1 \pm 1)$$

where q is an integer. Obviously, this condition allows for both a linearly and a quadratically scaling component to all harmonic orders. However, if the precise $\omega - 1.95\omega$ treatment carried out, the spectral component $n\omega$ are allowed to have a linearly scaling component if

$$n = 2.95q \pm (2 \pm 1)$$

and a quadratic component if

$$n = 2.95q \pm (2 \pm 1) \pm (1 \pm 1)$$

which are consistent with the observations. Hence, if we take the route of approximating the experiment to $\omega - 2\omega$, we must defer the entire discussion of the broken rotational symmetry to the supplement. Of course, this is possible, and if necessary, we can do so. But for now, we have tried an

alternative approach for the presentation of the broken rotational symmetry, that would hopefully be more easily digestible. We avoid the reformulation of the problem with ω' , and analyze the field as a sum of circularly polarized fields. We feel that this presentation is more intuitive and approachable, but still, it is impossible to avoid complexity altogether in the presentation of this experiment.

We believe that the emergence of robust analytical selection rules alongside this physical complexity is part of the beauty of the results.

13. In the analysis of bielliptical fields, it is extremely unclear at the start of the presentation what $\mathbf{p}=(p_x,p_y)$ should be understood as within this context. This introduces a disconnect between the first and second halves of the manuscript which significantly damages the readability and cohesion of the text.

Moreover, once the analysis gets going, p is used in a much more hand-wavy way than as originally used in the definition of W in eq. (1). If the synthetic dimension is going to be a generic parameter, that's fine by me, but it does need to be correctly represented from the beginning.

We realize that this analysis and presentation were confusing and not easy to grasp. For this reason, we entirely rewritten this section, with a focus on the specifics of this experiment instead of the connection to the general scenario.

14. In a similar vein, the p plane depicted in fig. 1(d) is completely unclear to me. This needs to be defined much more clearly and much more prominently. The space defined by the geometry shown there is definitely extremely valuable -- which implies that it should be included in the paper, but just adds to the need for a full definition and a clear explanation.

We have eliminated this representation from the discussion, and instead, included a simpler representation which is hopefully easier to grasp. This is represented in Figure 3. (b-c).

15. The authors claim that their analysis is independent of any use of perturbation theory. However, when actually faced with the concrete case of how their bi-elliptical configuration behaves as θ is varied away from 0° and 45° , the authors immediately resort to representing the fields to first order in the deviation of θ from these points, which significantly undercuts the results.

Thank you for this good very good comment. We have employed a more precise, less complicated representation of the driving field that does not require us to approximate it using perturbation theory. Now, the driving field is represented directly as a function of the ellipticities of the bi-chromatic drives rather than the QWP angle θ . It is given in Eq.(5) of the revised manuscript by

$$E(t, \epsilon) = \sqrt{\frac{1}{1 + \epsilon^2}} \Re\{e^{i\omega t}(i\epsilon\hat{x} + \hat{y}) + e^{1.95i\omega t}(i\hat{x} - \epsilon\hat{y})\}$$

where the dependence of ϵ on θ is explained in the following paragraph. Then, the selection rules associated with the broken reflection symmetry are fleshed out in terms of either ϵ (around $\theta = 0^\circ$), or $\delta = 1 - \epsilon$ (around $\theta = 45^\circ$). The only perturbative approximation left in this section is the approximation $\delta \propto (\theta - 45^\circ)$, after the selection rules are obtained. This is done for conciseness, and it does not harm the agreement with the data.

16. The fitting procedure to a polynomial model mentioned in p. 7 needs to be explained in significantly more detail, and specific attention needs to be paid to the uncertainties that result in the fit coefficients produced by the procedure as arising both from the experimental uncertainties in the underlying data as well as the confidence-interval estimations produced by the fit itself.

This is particularly relevant when the authors state "in accordance with the analytically-derived selection rule that forbids quadratic contributions (because the coefficient for θ^2 is ~ 200 weaker than the θ coefficient)". This is inappropriate: either the predicted coefficient for the quadratic scaling is consistent with zero, or it isn't. As reported, it isn't. If the accuracy of the experiment is such that

the reported fit coefficient is actually negligible, this needs to be argued and justified in full.

Let us describe the fitting procedure of the data in the manuscript.

Obtaining $|E_{n\omega}(\theta)|$ from the raw data: The data presented in the figures in the data for harmonic amplitudes, which is obtained from a square root over a 10-pixel frequency integral. This is equivalent to integration over the spectral range of $n\omega \pm 0.0225\omega$ where n is the harmonic order discussed (i.e., either 20.7, 19.75 or 18.8).

Fitting the data: For each frequency $n\omega$, two separate fits need to be carried out. One for a θ range around 0° and one for a θ range around 45° . The range around 0° was picked to be $0 \pm 6^\circ$. The range around 45° was picked to be $45^\circ \pm 10^\circ$. The same ranges were used for all harmonics. In the original manuscript, we fit each harmonic amplitude to a general polynomial model. In the revised manuscript, we fit each harmonic amplitude to 3 fitting models of the form $|a_m(\theta - \theta_0)^m|$, for $m = 1, 2, 3$ (a constant term was allowed for initially allowed harmonics). We obtain 6 fits and R^2 values for each harmonic amplitude, for a total of 18 fits. The R^2 values of these fits are summarized in Fig. 3(d) of the revised manuscript. We found that fits to the lowest order analytical contribution result in $R^2 > 0.95$ while fits to all other contributions (analytically forbidden) result in much smaller values. This shows consistency between experiment and theory.

The revised manuscript reads:

“

Figures 4(c-e) show the measured harmonic amplitudes of spectral components $n=18.8, 19.75$ and 20.7 as a function of the waveplate angle, θ , obtained by integrating the measured signal (Fig. 4(b)) in a range of $n\omega \pm 0.0225\omega$ and taking the square root. Each harmonic amplitude curve was fitted to three models of the form $|a_m(\theta - \theta_0)^m|$, for $m = 1, 2, 3$ (a constant term was allowed for initially allowed harmonics). The obtained R^2 values of these fits are summarized in the table in Fig. 4(f). The formula and R^2 values of the best fit to each harmonic amplitude, as well

as overlays of the numerical fits over the measured harmonic amplitudes, are shown in Figs 4(c-e). Comparing the obtained R^2 values in Fig. 4(f) with the predicted scaling (Fig. 3(d)), we observe that fits to the predicted lowest order allowed contributions resulted with $R^2 > 0.95$ for all the six examined cases, while all other fits were significantly smaller.

”

17. One alarming bit of (mis-)notation: when the authors say "H19.75 scales as $9.5(\theta-0.1)+0.05(\theta-0.1)^2$ ", what is "0.1"? is it 0.1 radians, or 0.1°? The inconsistent handling of the degree symbol, both in the text and in fig. 1, is not acceptable in published literature.

(And, linking back to my previous comment -- what is this offset, and why was it produced? what kind of fit was used that implied such a form? why was that form used?)

0.1 are degrees, and this is now corrected in the text and in the figure.

This offset is a small offset we allow for in the fitting procedure, because it is impossible to know a priori what is the exact QWP angles that produced the “most symmetric” field. In addition, this degree of freedom does not make a significant difference in any case, as the offsets the fitting procedure produces are several orders of magnitude smaller the explored angular ranges.

Minor comments

18. The text makes significant use of Floquet theory, including the concept of the Floquet Hamiltonian (which is rather esoteric to uninitiated readers), throughout. The authors should include a reference to a suitable textbook-level introduction.

We have added a citation to Holthaus, M., Floquet engineering with quasienergy bands of periodically driven optical lattices. *J. Phys. B At. Mol. Opt. Phys.* 49, 13001 (2015).

This paper includes a pedagogical introduction to Floquet theory.

"

In periodically driven (Floquet⁴) systems...

"

19. In the citation block 10-15 in the introduction, the authors could consider including [PRL 112, 135502 (2014)], which presents the same symmetries used in nonlinear-optical harmonic generation at a vastly different length scale.

Thank you. We have included the citation of this manuscript in the relevant citation block.

20. The authors' use of the inverted caret for the symmetry operation over the synthetic dimension is over-complicated notation for no good reason. As a guide, if the general reader cannot give a clearly-recognizable name to the notation (other than "zeta with a funny hat"?), then it should be changed. In some situations, this type of complex notation is unavoidable (i.e. when all other avenues have been exhausted), but this is not the case here.

We have changed this to the standard hat symbol throughout the MS and SI.

21. The use of the indices k, l, h, j , in eq. (3) and afterwards, is extremely confusing. Why those letters? why not a consecutive set?

We have made the change throughout the paper and the SI from k, l, h, j to a, b, c, d .

22. I find eq. (4) essentially unreadable. This is partly due to the choice of vector notation (as opposed to e.g. a matrix used to represent the QWP?) and partly due to the lack of symbolic values for field strengths. The notation would be much clearer if F_1 and F_2 were used instead of the mysterious factors of $\sqrt{10}$.

Equation 4 was replaced. Instead of representing the driving field as a function of θ , we now represent it as a function of ϵ , where θ controls ϵ . The factor $\sqrt{10}$ (field ratio of the different coors) was replaced with a symbolic parameter Δ , in accordance with your suggestion. The equation for the driving field now reads:

$$\mathbf{E}(\mathbf{t}, \epsilon) = \sqrt{\frac{1}{1 + \epsilon^2}} \Re\{e^{i\omega t}(i\epsilon\hat{x} + \hat{y}) + \Delta e^{1.95i\omega t}(i\hat{x} - \epsilon\hat{y})\} \quad (7)$$

Following the equation, we include a brief explanation:

“Eq. (5) describes two counter rotating elliptically polarized beams of ellipticity $\epsilon(\theta)$, at frequencies $\omega - 1.95\omega$ where $\omega \equiv 2\pi/T$, and Δ is the two-color amplitude ratio. For $\theta = 0$, the ellipticity is $\epsilon = 0$ and the field is in a “cross-linear” configuration. For $\theta = 45^\circ$, the pump ellipticities are $\epsilon = 1$ and the field is in a “bi-circular” configuration. Figure 3(a) shows Lissajous curves of the driving field for different values of θ .”

23. In fig. 1(f-h), what is the signal that's being plotted? Is it a lineout? an integral over a frequency range? if so, what is the range?

This signal is a square root of an integral over a frequency range of the measurement.

To obtain the harmonic intensity of a particular frequency, we integrate over 10 pixels, which in the resolution of the experiment, are equivalent to 0.045ω .

That is, we integrate over the range $n\omega \pm 0.0225\omega$ where n is one of the harmonic orders we explored. Then, to obtain the harmonic amplitude (presented in the figures) we take the square root of the integrated signal. We have added this information inside the manuscript. :

"

Figures 4(c-e) show the measured harmonic amplitudes of spectral components $n=18.8, 19.75$ and 20.7 as a function of the waveplate angle, θ , obtained by integrating the measured signal (Fig. 4(b)) in a range of $n\omega \pm 0.0225\omega$ and taking the square root.

"

24. As a brief comment regarding formatting: I can understand (but don't really agree with) the motivation for providing reviewer copies in double spacing (i.e. to allow for easier annotation). However, this does not apply to the bibliography, and providing

the references section in (a) double spacing with (b) extremely large font size, makes no sense at all. The only result is a waste of paper on the side of the reviewer.

We are sorry for the inconvenience. We have changed the formatting of the paper to have a smaller font size in the references section, and 1.5 spacing throughout.

25. Similarly, providing the supplementary information in double-spaced format just produces a waste of paper on the side of the reader. The SI is already long and unwieldy enough. For the sake of the reader, the final version of the SI should be single-spaced. And I would argue that for the sake of the reviewer, the review copies should be single-spaced as well.

We are very sorry for your inconvenience. We have changed the SI to be single-spaced.

26. The supplementary material requires a thorough round of spell-checking, proof-reading and copy-editing. There's multiple format mishaps and various typos that should be fixed before the SI is published.

Thank you, we have proofread and spell-checked the SI. The revised SI includes many corrections.

27. In the introduction of the SI, it mentions that section IV contains the details of numerical simulations -- which have not yet been mentioned!

In the revised version of the manuscript, the numerical simulations are included within the main text. In the revised SI, the numerical simulations are discussed in section I.

28. In figure III.1 in the SI, is there a line joining the crosses? Or are the crosses adjacent to each other? It's impossible to tell, and that makes it hard to tell how the plots should be interpreted.

We deleted the crosses in the revised version (now part of the main text of the paper).

29. In §IV of the SI, the text mentions that "The ground state was found by representing [the] field-free Hamiltonian in matrix form on the cartesian spatial grid" without mentioning how the derivatives in the kinetic energy operator were handled. **The kinetic energy term was handled using the finite difference approximation on a 2 dimensional grid. The exact MATLAB syntax used in our code is in the revised SI:**

“

The kinetic energy operator was represented using the finite difference approximation on a two dimensional grid. In MATLAB, it is obtained by the syntax:

```
I = speye(Nx)
e = ones(Nx,1);
kin = (1/5040)*spdiags([-9*e 128*e -1008*e 8064*e -14350*e
8064*e -1008*e 128*e -9*e],[-4 -3 -2 -1 0 1 2 3 4],Nx,Nx)
Tmat = (-1/2)*(1/(dx^2)).*(kron(kin,I)+kron(I,kin))
```

Here, Nx is the number points on the x-axis and dx is the grid spacing. The command speye assigns a sparse identity matrix to I, the command ones assigns an array whose entries are 1, the variable kin is a representation of the operator d^2/dx^2 on a 1D grid, and the variable Tmat is the matrix representing the kinetic energy operator on a two dimensional grid. Kron is the Kronecker tensor multiplication operation.

“

30. In §V of the SI, the form of the Hamiltonian in eq. (V.1) is not suitably justified with reference to suitable literature, and it is hard to match against existing resources. The authors should provide appropriate references and explain in more depth the origin and handling of this Hamiltonian. Similarly, the symmetry operations in eq. (V.4) are not defined with enough detail for the reader to be able to understand and interpret them correctly.

We provided the appropriate references in the SI:

“

This model was discussed in the supplementary section of Ref⁸ in the context of HHG in spin-orbit coupled systems, as well as in supplementary Refs⁹⁻¹¹ in other physical contexts.

....

8. Lysne, M., Murakami, Y., Schüler, M. & Werner, P. High-harmonic generation in spin-orbit coupled systems. *Phys. Rev. B* 102, 081121(R) (2020).
9. Pareek, T. P. Anisotropic spin and charge transport in presence of spin-orbit interaction. *Phys. Rev. B - Condens. Matter Mater. Phys.* 66, 1-4 (2002).
10. Pareek, T. P. & Bruno, P. Magnetic scanning tunneling microscopy with a two-terminal nonmagnetic tip: Quantitative results. *Phys. Rev. B - Condens. Matter Mater. Phys.* 63, 1-5 (2001).
11. Mireles, F. & Kirczenow, G. Ballistic spin-polarized transport and Rashba spin precession in semiconductor nanowires. *Phys. Rev. B - Condens. Matter Mater. Phys.* 64, 24426 (2001).

“

We have also cited reference 7 when introducing the symmetry operation that includes an operation on the spin degree of freedom:

“

The driven Rashba Hamiltonian exhibits the symmetry⁸:

“

31. In §V of the SI, the authors find good selection rules for the emission produced by each electron of well-defined crystal quasimomentum \mathbf{k} , but they do not explain how those individual emissions are collected together and what that combination procedure does to the emission. This procedure depends on the situation (i.e. perturbative vs nonperturbative harmonic generation) and needs to be handled carefully. More generally, it is not clear how the authors connect the previous part of the section to any experimental observables.

(And, taking these two together, a case can be made that this section should be spun out and made to form the nucleus of a separate publication, but that's a choice for the authors to make.)

Thank you. This is an important point. Indeed, we have calculated the selection rules for the emission of a single electron in a well-defined crystal momentum, instead of the selection rule of the total emission given by the total current density. We followed supplementary reference 7, which discussed HHG and HHG selection rules in the spin orbit model we explore. It was shown there that the selection rules of the single electron emission (i.e., those of $v(k, \omega)$) are consistent with a numerical calculation of the total emission (i.e., $J(\omega)$ which includes a sum over k). It would be more physically meaningful and correct to derive the selection rules for $J(\omega)$ and not $v(k, \omega)$, and this is indeed possible using the same methodology. However, to stay consistent with ref. 7., we prefer to remain with the existing treatment, and to explicitly clarify the difference between J and v in writing:

“

Next, we derive the HHG selection rules. The standard selection rules due to DS \hat{X} was derived in Ref8. We extend their treatment to real-synthetic symmetries of the form $\hat{X} \cdot \hat{\zeta}$. The HHG spectrum is given by the expression $|i\omega J_i(\omega)|^2$ where $|J_i(\omega)|$ is Fourier transform of the charge current, and the magnetization current was neglected. The charge current is given by $J_i(t) = N_k^{-1} \sum_k Tr[\rho(t)v_i(k, t)]$ in the $i = x, y, z$ direction. Here, $\rho(t)$ is the time dependent charge density and $v_i(k, t)$ is the charge velocity. Generally, to obtain selection rules on the complete harmonic emission, one needs operate on $J_i(t)$ with $\hat{X} \cdot \hat{\zeta}$ and employ its invariance under the symmetry operation. While this is possible, Ref8 showed that for this model, the selection rules are well approximated by the emission of a single electron in a well-defined crystal momentum k . Hence, for simplicity, we derive the selection rules for $v(k, t)$ and not $J(t)$.

“

32. In p. 37 (!) of the SI, after equation (VII.3), the authors write $[C_6, W, W] \neq 0$ - yet another formulation of the symmetry operations and their relationship to the Hamiltonian! This emphasizes the need for a consistent and clearly handled use of notation.

Thank you. This was an error. The correct expression is $[\hat{C}_6, \hat{W}] \neq 0$. This is corrected in the revised SI.

As mentioned above, the presentation of the results requires a significant overhaul, probably including an almost complete rebuild of the manuscript. There were further smaller details, but it only makes sense for the peer-review process to address them once the manuscript is closer to publishable form.

I hate to be the bearer of bad news -- particularly for a paper with results as valuable as these -- but I really think that if the presentation issues I have mentioned are not addressed, the paper will become significantly less usable by its target audience and its impact will become vastly reduced compared to what the results really deserve.

We thank the reviewer his very detailed comments, many very good suggestions that guided us to improve the paper.

** See Nature Research's author and referees' website at www.nature.com/authors for information about policies, services and author benefits.

Our flexible approach during the COVID-19 pandemic

If you need more time at any stage of the peer-review process, please do let us know. While our systems will continue to remind you of the original timelines, we aim to be as flexible as possible during the current pandemic.

This email has been sent through the Springer Nature Tracking System NY-610A-NPG&MTS

Confidentiality Statement:

This e-mail is confidential and subject to copyright. Any unauthorised use or disclosure of its contents is prohibited. If you have received this email in error please notify our Manuscript Tracking System Helpdesk team at <http://platformsupport.nature.com> .

Details of the confidentiality and pre-publicity policy may be found here <http://www.nature.com/authors/policies/confidentiality.html>

Privacy Policy | Update Profile

REVIEWER COMMENTS

Reviewer #2 (Remarks to the Author):

In this new version of the manuscript, the authors have significantly restructured the text, greatly improving its readability and clarity. Furthermore, I believe that they have adequately address all the issues I raised in the first review round.

However, I still believe that the employed notation for period and the frequency of the system is still confusing, as the same symbols have different meanings in each section. Case in point, in the new section "Experimental investigation of selection rules by real-synthetic DS", T is used to denote the period of the reference omega-field and T' is the true period of the bi-chromatic field. However, in section "General Theory" T is said to be the period of the Floquet system. Unless I am missing something, since in the experimental example, the perturbation does not alter the period of the Floquet system (the perturbation is change in polarization), we would have that T' of the section "Experimental investigation of selection rules by real-synthetic DS" corresponds to T of "General Theory". This is quite confusing and should be clarified.

If I understand correctly, there are 3 different periods relevant to the present formalism:

- 1) the (true) period of the unperturbed Floquet system;
- 2) the period of a reference field (the omega-field);
- 3) the (true) period of the Floquet perturbed system (which can be distinct from the period of the unperturbed system).

The text and notation should make a clear distinction between these 3 different periods and at present fails to do so. A possibility would be to use T_0 for the period of the unperturbed Floquet system, T_W for the perturbed Floquet system, and T_ω for the reference omega-field. The notation of angular frequency should also reflect this, using ω_0 , ω and ω_W , respectively. This is just a suggestion and any other notation that clearly distinguishes the 3 periods would be adequate, provided it is used consistently throughout the whole text.

I have a few other question related to this problem.

- 1) In Eq. 8, is the variable s a rational number? From the previous version of the manuscript, I understood as much, but in the current version, that is not so clear. Please clarify this.
- 2) In Eq. 10, ω represents the fundamental frequency of the reference field, of the unperturbed Floquet system or of the perturbed Floquet system? I would expect it to be the fundamental frequency of the perturbed Floquet system. But this should be made clear.

I have a few suggestions:

- a) I think that immediately above or below eq. 5, it should be clarified that the two fields are commensurate and state what is the periodicity of this bi-chromatic field, instead of just saying that further ahead in the text.
- b) It would be helpful to clarify the meaning of "dressing laser fields" in the first line of the section "General Theory".

All these questions/suggestions are minor issues. While these should be addressed by the authors, I can already recommend the paper to be published on Nature Communications.

I would also like to thank the authors for the detailed answer regarding my question on the use of quasi-periodic approximants. I look forward for their upcoming publication on incommensurate frequencies.

Reviewer #3 (Remarks to the Author):

See attached report in PDF format.

The manuscript is vastly improved, and it is much closer to publishable form. The authors have successfully implemented the recommendation to rebuild the manuscript from the ground up, and as a result the manuscript is now readable and accessible and it presents a solid argument for its theses which will be easily approachable by its intended audience. I continue to think that the strength of the core results contained in this manuscript is sufficient to warrant publication in Nature Communications, and the improvements in presentation bring the paper very close to publishability. However, that said, there are still several critical issues remaining which must be addressed before publication.

Major comments

1. I am unable to reproduce the details of the authors’ handling of the DS of the bielliptical field. It is possible that this is because the conventions in use have not been clearly communicated. I presume the sign conventions are such that $\hat{R}_{2.95}\mathbf{r}_{\pm} = e^{i2\pi/2.95}\mathbf{r}_{\pm}$ and $\hat{t}_{2.95}e^{in\omega t} = e^{in2\pi/2.95}e^{in\omega t}$, as this is the only way I can see which will retain the form of the first half of the field in eq. (6). (These sign conventions, including those two specific actions, should be included after that equation, to avoid ambiguity.)
Under this assumption, the action of the DS operator on the symmetry-breaking terms is then $\hat{C}_{2.95}\delta_1\mathbf{r}_+e^{i\omega t} = \delta_1e^{(1+1)2\pi i/2.95}\mathbf{r}_+e^{i\omega t}$ for the first term and $\hat{C}_{2.95}\delta_2\mathbf{r}_-e^{1.95i\omega t} = \delta_2e^{(-1+1.95)2\pi i/2.95}\mathbf{r}_-e^{1.95i\omega t}$ for the second one. This then implies that the action of the synthetic component of the new symmetry on the second term is correctly inferred as $\hat{\zeta}(\delta_2) = \delta_2e^{-0.95\times 2\pi i/2.95}$, but for the first term the phase $e^{+0.95\times 2\pi i/2.95}$ seems to be incorrect – the same calculation indicates that the action should be $\hat{\zeta}(\delta_1) = \delta_1e^{-2\times 2\pi i/2.95}$. This raises two possibilities:
 - a. The derivation I have just made is incorrect, in which case the correct version should be laid out much more clearly in the manuscript. This entails giving unambiguous definitions in the main text, and a more detailed walkthrough in the SI.
 - b. The derivations in the paper need to be corrected.
2. I am unconvinced by the claims, at the end of the paragraph below equation (4), that even and odd harmonics scale linearly and quadratically, which are presented as categorical and unequivocal. To me, what this says is that the range of values of λ which is used for this calculation is too restricted. Given the structure of the problem, there is guaranteed to be a point in which λ is big enough that the pure linear/quadratic scalings are no longer operative, and the behaviour gives way to generic odd/even behaviour, starting with the inclusion of cubic/quartic terms. (Indeed, this is baked into the manuscript’s assertion that the selection rules under consideration are independent of perturbation theory.) I think the manuscript’s point would be carried across much more clearly if the range of λ in use here were expanded enough for that point to be reached.
3. I am unconvinced by the authors’ response regarding the use of the $\omega : 1.95\omega$ frequency ratio in the theoretical description of their experiment with bielliptical fields, which I initially raised as comment 12 of my first report.
 - a. As a first observation: there is a key missing piece of information regarding the experimental configuration, namely, the duration of the pulses used in the experiment. As a reader, the closest one can get is by following the manuscript’s

reference 29 (which is indicated as essentially the same configuration), where the duration of the initial pump laser is given as 27fs. This amounts to 10 cycles of the fundamental beam, which has some severe implications: the “primed” dynamical symmetry (reflection on the y axis coupled with a time translation by 10T) does not have enough time to establish itself, and indeed translating in time by 10T does not give the same system – it takes the configuration completely out of the duration of the pulse. This is in stark contrast with the authors’ assertion (in their reply to my initial comment 12) that the reflection symmetry would be retained but the rotation symmetry would break.

- b. Generally, I think the core of the issue concerns the resolution of the experimentally observed spectrum (figure 4b) into individual subchannels as done in the original 2014 experiment. For the claimed frequency ratio, the allowed channels for harmonic ~ 19 split into a “base” channel at $18.7\omega = 7\omega + 6 \times 1.95\omega$ (allowed at full symmetry) together with the perturbation-driven subchannel at harmonic frequencies $18.75\omega = 9\omega + 5 \times 1.95\omega$ and $18.8\omega = 11\omega + 4 \times 1.95\omega$ (and so on) forbidden at full symmetry, with the HO18.75 subchannel possibly forbidden depending on the configuration. If the SH detuning from 2ω to 1.95ω is experimentally important, then one would expect to see all of the allowed subchannels present and clearly resolved in the measured spectra. However, I do not think this is the case: the presented spectra do show noticeable shifts in frequency depending on the value of θ , but these are not enough to clearly separate into individual subchannels.

- c. This leads me to the authors’ assertion, in their reply to the initial comments, that if the simpler frequency ratio were used, the results in the paper would break, and the selection rules would change.

I am unable to follow the authors’ calculations in their reply, mostly because I do not understand the action of the DS as I mentioned in comment 1 above; I also do not understand the meaning of the multiple \pm signs used.

Nevertheless, it is very strange to claim that such a small change in how the situation is analysed will bring about such a massive change in the observed selection rules (i.e. the allowed scalings w.r.t. the perturbation for the chosen lines). The authors claim that the allowed scalings would “change” from cubic/quartic to linear/ quadratic, and I find this implausible. Instead, it is much more likely that each line consists of multiple subchannels (as laid out above) and that each line contains one subchannel with linear/quadratic scaling, and that the authors are interested in additional subchannels which are detuned from that simpler linear/quadratic one. If this is the case, then it should be explained in depth, which implies adding a subsection in the SI exploring in detail the scaling of all the different subchannels of at least one chosen line of interest.

- d. As a minor, related question: at the end of p. 7, where the integration ranges are specified as $n\omega \pm 0.0225\omega$, is n an integer (as the notation would lead one to assume), or are we meant to infer $n = 18.8$ (etc.)?

4. I feel that the Lissajous figures reported in figure 3 do not accurately represent the fields in use. The Lissajous figures currently depicted correspond to the non-detuned frequency ratio $\omega:2\omega$, but if indeed the redshifting of the SH is essential, as the authors claim, then the Lissajous figures change to look something like the following,

with the relative phase of the two fields slipping slightly on each period. I appreciate the need for simplicity, but this needs to be reported correctly (i.e. it should be at least indicated in the text that the Lissajous figures as shown consist only of a cut lasting a single period; I would still recommend in that case to include depictions of the full figures in the SI).

Minor comments

5. The reference block [27,28] used for HHG is not particularly informative for readers who are unfamiliar with HHG. I would recommend adding a generic review here; a good example is *Contemp. Phys.* 59, p. 47 (2018).
6. In the first paragraph of the Results section, “broken in the systems in the left panel” – shouldn’t this be the right panel?
7. Same paragraph: in “the amplitude and phase of the applied voltage”, the voltage is DC, so it has a sign but it does not have a phase.
8. I find the passage between equations (3) and (4) to be quite violent; for many readers, this will be the point at which they stop understanding the details of the manuscript. Within this introductory-example section, this type of manipulation should be explained in more detail.
9. The order of introduction of figures in the text is somewhat jolting – fig. 4 is introduced before fig. 3. I think the order of the figures is largely fine, but they should be introduced in correct order inside the text. As it is, the discrepancy is distracting and this detracts from smooth reading of the manuscript.
10. In the expression for the bielliptical field (eq. (5) and onwards), why do the exponentials use negative frequencies (i.e. $e^{i\omega t}$ instead of $e^{-i\omega t}$)? This breaks the conventions of the field and makes the material harder to work with.
11. For the bielliptical experiment, the dependence of the ellipticity on the waveplate angle, $\epsilon(\theta)$, needs to be given explicitly.
12. The handling of the bielliptical perturbation parameters, $\delta_{1,2}$, is confusing. As initially presented they are real, but then they are given complex phases. Which is it?
13. There is a missing imaginary unit in the final exponent in equation (6).
14. In the specification $\mathbf{r}^{\pm} = \hat{x} \pm i \hat{y}$, after equation (6), what do the hats actually mean? I find the mixture of notation quite confusing. Wouldn’t it be better to use uniform and unambiguous notation for basis vectors, such as e.g. $\hat{\mathbf{e}}_x$, $\hat{\mathbf{e}}_y$ and $\hat{\mathbf{e}}_{\pm}$? Consider inserting a paragraph break before “As θ is detunes”, at the bottom of p. 6.
15. In p. 9, the text reads “In the examples above, Q is a vector...” before Q has appeared at all.
16. Some formatting issues: X in eq (9) and n under eq. (10) should not be bold. There is also a left-over summation over k,l,h,j in equation (10).
17. In figure 1, right-hand panel, the “bichromatic” pulse is depicted in red and blue, but it is not bichromatic (both components are at the same frequency).
18. In figures 2 and 3, the colour schemes used for the Lissajous figure need to be explained in the caption.
19. In figure 2, for the second and third rows, it feels to me like the symmetry breaking would be clearer if a smaller value of λ were used.

20. In figure 2, above the simulation results, the scheme claims the configuration in use contains “randomly oriented molecules”, instead of spherically-symmetric monatomic gas atoms.
21. In table I, the correspondence between rows on both columns is unclear. Presumably each row on the right corresponds to multiple rows on the left, at least at the top, but the which-to-which correspondence is highly unclear, making the table essentially useless.
22. Note missing publication details on reference 22.
23. In §II of the SI, the symmetries are introduced very sharply. While some of the newer symmetries are quite complex, I would argue that if there are clear, simple examples of field configuration with that symmetry, it would be good to mention them briefly, to provide a mental anchor which can be used to understand the results.
24. The supplementary information requires a round of careful copy-editing. Some issues I caught:
 - a. In eq. (II.5), a leftover (p), an unduly bold n, and a leftover summation over k,l,h,j
 - b. Throughout the SI, incorrectly upright symbols that should be italicized
 - c. Missing superscripts in (II.2) and indices in the summation symbols in (III.2) and (III.3).
 - d. Leftover (p) in (IV.7).
25. In eq. (II.50), I don’t feel the use of column-vector notation for the circular basis is appropriate. As written it is unambiguous if one reads the text in detail, but the potential for confusion is too high. An explicit basis summation would be highly preferable.
26. Relatedly, are circular-basis indices L/R or +/-? Eq. (II.50) uses the former, while (II.58) uses the latter.
27. For eq. (II.55), as an alternative to the notation $\exists z \in Z: (\text{quantity}) = Nz$, consider using the equivalent modulo notation, $(\text{quantity}) \equiv 0 \pmod{N}$.

I have also been asked to assess the authors’ response to the concerns of Referee 1. I am not really able to say whether Referee 1 would be satisfied by the response to the preamble, since I disagree with Referee 1 on the value provided by the manuscript. It is probably true that, as Referee 1 suspects, “many of these Taylor series showing various missing powers of theta could be derived from direct Taylor series of the time-dependent fields without any reference to this symmetry argument”, but that misses the point – even if such a derivation can be made on a case-by-case basis, there is still significant value in the general theory.

Regarding Referee 1’s detailed comments:

1. This seems to have been addressed fully.
2. This seems to have been addressed fully.
3. This seems to have been addressed fully.
4. I understand Referee 1’s point regarding the terminology, but this is ultimately a subjective matter. I feel that the authors’ reply lays out a valid justification for their use of the term.

Response to Reviewer #2 (Remarks to the Author):

We thank the referee for taking the time to review our manuscript, and for providing us with helpful comments, which have improved the quality of our paper. We also wish to thank the referee for recommending publication of our manuscript. Below we respond to the referee's comments point-by-point, where his/her comments are in black, and our answers and changes in the manuscript are written in blue.

Reviewer #2 (Remarks to the Author):

In this new version of the manuscript, the authors have significantly restructured the text, greatly improving its readability and clarity. Furthermore, I believe that they have adequately address all the issues I raised in the first review round.

However, I still believe that the employed notation for period and the frequency of the system is still confusing, as the same symbols have different meanings in each section. Case in point, in the new section "Experimental investigation of selection rules by real-synthetic DS", T is used to denote the period of the reference omega-field and T' is the true period of the bi-chromatic field. However, in section "General Theory" T is said to be the period of the Floquet system. Unless I am missing something, since in the experimental example, the perturbation does not alter the period of the Floquet system (the perturbation is change in polarization), we would have that T' of the section "Experimental investigation of selection rules by real-synthetic DS" corresponds to T of "General Theory". This is quite confusing and should be clarified.

If I understand correctly, there are 3 different periods relevant to the present formalism:

- 1) the (true) period of the unperturbed Floquet system;
- 2) the period of a reference field (the omega-field);
- 3) the (true) period of the Floquet perturbed system (which can be distinct from the period of the unperturbed system).

The text and notation should make a clear distinction between these 3 different periods and at present fails to do so. A possibility would be to use T_0 for the period of the unperturbed Floquet system, T_W for the perturbed Floquet system, and T_ω for the reference omega-field. The notation of angular frequency should also reflect this, using ω_0 , ω and ω_W , respectively. This is just a suggestion and any other notation that clearly distinguishes the 3 periods would be adequate, provided it is used consistently throughout the whole text.

Thank you for your comment. Your understanding is exactly correct – in the previous section, T' in the experimental section corresponds to T in the general theory section.

For the symmetry analysis employed in our manuscript, the relevant period is the (true) period of the Floquet perturbed system (item 3 in the above list). This is the period which we utilize to formulate the Floquet group symmetries on which the formalism relies.

When the period of the unperturbed Floquet system (item 1) is different than the period of the perturbed Floquet system (item 3), one should choose to work with the period of the perturbed system to consistently employ a Floquet treatment.

As for the period of a "reference field", if we correctly understand your meaning, it is simply the period of the unperturbed Floquet system.

According to the referee's suggestions, to keep all sections consistent with one another we have made the following changes:

- 1) In the experimental section, the period of the ω field is denoted by T_ω and it is emphasized repeatedly that the period of the bi-chromatic field is $T = 20T_\omega$.
- 2) In the general theory section, the period of the Floquet system is denoted by T , which is now consistent with the notation in the experimental section. Additionally, step by step detailed discussion of the correspondence between the table and the experiment is given in the SI, section III.

I have a few other question related to this problem.

1) In Eq. 8, is the variable s a rational number? From the previous version of the manuscript, I understood as much, but in the current version, that is not so clear. Please clarify this.

Yes, s is indeed a rational number. Otherwise, a Floquet theory with a single temporal dimension cannot be consistently applied (as discussed in the previous reply). In the revised manuscript, we explicitly constrain s to being rational after equation (8):

“

Here, ω is the fundamental frequency of the symmetric Floquet system and $s\omega = 2\pi s/T$ is the frequency of the symmetry breaking field \widehat{W} , where s is a rational number.

“

2) In Eq. 10, ω represents the fundamental frequency of the reference field, of the unperturbed Floquet system or of the perturbed Floquet system? I would expect it to be the fundamental frequency of the perturbed Floquet system. But this should be made clear.

In Eq.10, ω represents the fundamental frequency of the perturbed Floquet frequency. We have clarified this in the revised manuscript, right after equation 10:

“

Here, ω is the fundamental frequency of the perturbed Floquet system,...

“

I have a few suggestions:

a) I think that immediately above or below eq. 5, it should be clarified that the two fields are commensurate and state what is the periodicity of this bi-chromatic field, instead of just saying that further ahead in the text.

Thank you. In the revised manuscript, this is stated almost immediately after Eq.(5):

“

Eq. (5) describes two counter rotating elliptically polarized beams of ellipticity $\epsilon(\theta)$, at frequencies $\omega - 1.95\omega$ where $\omega \equiv 2\pi/T_\omega$, and Δ is the two-color amplitude ratio. For $\theta = 0$, the ellipticity is $\epsilon = 0$ and the field is in a “cross-linear” configuration. For $\theta = 45^\circ$, the pump ellipticities are $\epsilon = 1$ and the field is in a “bi-circular” configuration. Figure 3(b) shows Lissajous curves of the driving field for different values of θ . We note that Figure 3(b) depicts the Lissajous curves in the temporal window between 0 and T_ω , while the periodicity of the bi-chromatic field is $T = 20T_\omega$ (the complete Lissajous curves are presented in the SI, section V).

“

b) It would be helpful to clarify the meaning of “dressing laser fields” in the first line of the section “General Theory”.

The revised manuscript clarifies the meaning of this:

“

In this section we classify real synthetic symmetries imposed by dressing laser fields, using Floquet group theory. That is, we consider a Floquet system that initially exhibits some DS \hat{X} to be subject to an external laser field (the so-called dressing field), that transforms its DS \hat{X} to a real-synthetic symmetry $\hat{X} \cdot \hat{\zeta}$, and tabulate the corresponding selection rules by Floquet group theory.

“

All these questions/suggestions are minor issues. While these should be addressed by the authors, I can already recommend the paper to be published on Nature Communications.

I would also like to thank the authors for the detailed answer regarding my question on the use of quasi-periodic approximants. I look forward for their upcoming publication on incommensurate frequencies.

We thank the reviewer for his valuable comments, and for the time and effort he put in reviewing our manuscript. Your comments helped us to significantly improve the manuscript.

Response to Reviewer #3 (Remarks to the Author):

We would like to thank the reviewer for his many valuable comments in reviewing our manuscript. We sincerely appreciate the time and effort you took to evaluate our work, and to deeply understand and interact with the presented material. Your questions and comments helped us to significantly improve the manuscript. Below we respond to the referee's comments point-by-point, where his/her comments are in black, and our answers and changes in the manuscript are written in blue.

Reviewer #3 (Remarks to the Author):

The manuscript is vastly improved, and it is much closer to publishable form. The authors have successfully implemented the recommendation to rebuild the manuscript from the ground up, and as a result the manuscript is now readable and accessible and it presents a solid argument for its theses which will be easily approachable by its intended audience. I continue to think that the strength of the core results contained in this manuscript is sufficient to warrant publication in Nature Communications, and the improvements in presentation bring the paper very close to publishability. However, that said, there are still several critical issues remaining which must be addressed before publication.

Major comments

1. I am unable to reproduce the details of the authors' handling of the DS of the bielliptical field. It is possible that this is because the conventions in use have not been clearly communicated. I presume the sign conventions are such that $\hat{R}_{2.95} \mathbf{r}_{\pm} = e^{i2\pi/2.95} \mathbf{r}_{\pm}$ and $\hat{t}_{2.95} e^{i\omega t} = e^{i2\pi/2.95} e^{i\omega t}$, as this is the only way I can see which will retain the form of the first half of the field in eq. (6). (These sign conventions, including those two specific actions, should be included after that equation, to avoid ambiguity.)

Thank you for this very good comment. The transformation rules were laid out in (Neufeld, Podolsky and Cohen, 2019, *Nature Communications*). They are

$$\begin{aligned}\hat{R}_{2.95} \mathbf{r}^{\pm} &= e^{\pm 2\pi i/2.95} \mathbf{r}^{\pm} \\ \hat{t}_{2.95} e^{i\omega t} &= e^{2\pi i/2.95} e^{i\omega t}\end{aligned}$$

Let us employ these relations to show that $\mathbf{E}(t) = \hat{C}_{2.95} \mathbf{E}(t) = \hat{t}_{2.95} \cdot \hat{R}_{2.95} \mathbf{E}(t)$ for $\delta_1 = \delta_2 = 0$ (Eq.6).

For $\delta_1 = \delta_2 = 0$, the field is

$$\mathbf{E}(t) = \Re \frac{i}{2} \{ \eta(\mathbf{r}^+ e^{1.95\omega t} + \mathbf{r}^- e^{i\omega t}) \}$$

We operate with $\hat{R}_{2.95}$:

$$\begin{aligned}\hat{R}_{2.95} \mathbf{E}(t) &= \Re \frac{i}{2} \{ \eta(\hat{R}_{2.95} \mathbf{r}^+ e^{1.95\omega t} + \hat{R}_{2.95} \mathbf{r}^- e^{i\omega t}) \} \\ &= \Re \frac{i}{2} \{ \eta(e^{+2\pi i/2.95} \mathbf{r}^+ e^{1.95\omega t} + e^{-2\pi i/2.95} \mathbf{r}^- e^{i\omega t}) \}\end{aligned}$$

Next, we operate with $\hat{t}_{2.95}$ which cancels the phase factors added by the rotation:

$$\hat{t}_{2.95} \cdot \hat{R}_{2.95} \mathbf{E}(t) = \Re \frac{i}{2} \{ \eta(e^{1.95 \times 2\pi i/2.95} e^{+2\pi i/2.95} \mathbf{r}^+ e^{1.95\omega t} + e^{2\pi i/2.95} e^{-2\pi i/2.95} \mathbf{r}^- e^{i\omega t}) \}$$

$$\hat{t}_{2.95} \cdot \hat{R}_{2.95} \mathbf{E}(t) = \Re \frac{i}{2} \{ \eta(e^{(1.95+1) \times 2\pi i/2.95} \mathbf{r}^+ e^{1.95\omega t} + e^{(1-1) \times 2\pi i/2.95} \mathbf{r}^- e^{i\omega t}) \} = \mathbf{E}(t)$$

To avoid ambiguity, we have added the following sentence inside the revised version of the manuscript:

“polarized symmetry breaking fields which result in selection rule deviations as the symmetry is broken. The physical field in our experiment is represented by $\eta = 1 + \epsilon$ and $\delta_{1,2} = 1 - \epsilon \propto (\theta - 45^\circ)$. The field $\eta(\mathbf{r}^+ e^{1.95i\omega t} + \mathbf{r}^- e^{i\omega t})$ exhibits $\hat{C}_{2.95}$ DS (the relations $\hat{R}_{2.95}\mathbf{r}^\pm = e^{\pm 2\pi i/2.95}\mathbf{r}^\pm$ and $\hat{t}_{2.95}e^{i\omega t} = e^{2\pi i n/2.95}e^{i\omega t}$ are useful for verifying it.). In contrast, the fields of amplitudes $\delta_{1,2}$ do not exhibit..., hence they are symmetry breaking. However, ...“

Under this assumption, the action of the DS operator on the symmetry-breaking terms is then $\hat{C}_{2.95}\delta_1\mathbf{r}^+e^{i\omega t} = \delta_1e^{(1+1)2\pi i/2.95}\mathbf{r}^+e^{i\omega t}$ for the first term and $\hat{C}_{2.95}\delta_2\mathbf{r}^-e^{1.95i\omega t} = \delta_2e^{(-1+1.95)2\pi i/2.95}\mathbf{r}^-e^{1.95i\omega t}$ for the second one. This then implies that the action of the synthetic component of the new symmetry on the second term is correctly inferred as $\hat{\zeta}(\delta_2) = \delta_2e^{-0.95 \times 2\pi i/2.95}$, but for the first term the phase $e^{0.95 \times 2\pi i/2.95}$ seems to be incorrect – the same calculation indicates that the action should be $\hat{\zeta}(\delta_1) = \delta_1e^{-2 \times 2\pi i/2.95}$. This raises two possibilities: a. The derivation I have just made is incorrect, in which case the correct version should be laid out much more clearly in the manuscript. This entails giving unambiguous definitions in the main text, and a more detailed walkthrough in the SI. b. The derivations in the paper need to be corrected.

Thank you for this comment. In fact, both derivations (yours and ours) are entirely correct and result in identical phase factors. Explicitly, the phase factor $e^{-2 \times 2\pi i/2.95}$ is equal to the phase factor $e^{0.95 \times 2\pi i/2.95}$.

$$e^{-2 \times \frac{2\pi i}{2.95}} = 1 \times e^{-2 \times \frac{2\pi i}{2.95}} = e^{2.95 \times \frac{2\pi i}{2.95}} \times e^{-2 \times \frac{2\pi i}{2.95}} = e^{0.95 \times 2\pi i/2.95}$$

And therefore, the statement $\hat{\zeta}(\delta_1) = \delta_1e^{0.95 \times 2\pi i/2.95}$ is accurate and identical to the statement $\hat{\zeta}(\delta_1) = \delta_1e^{-2 \times 2\pi i/2.95}$. In order to enhance clarity, the revised manuscript includes both representations:

“

components are symmetric under the operation $\hat{C}_{2.95} \cdot \hat{\zeta}$, where $\hat{\zeta}$ phase shifts the symmetry breaking field components $\delta_1 \xrightarrow[\hat{\zeta}]{\hat{C}_{2.95}} \delta_1 e^{-2 \times 2\pi i/2.95} = \delta_1 e^{0.95 \times 2\pi i/2.95}$, $\delta_2 \xrightarrow[\hat{\zeta}]{\hat{C}_{2.95}} \delta_2 e^{-0.95 \times 2\pi i/2.95}$.

Figure 3(c) illustrates how each circularly polarized components of the driving field is transformed separately by the operation $\hat{C}_{2.95} \cdot \hat{\zeta}$. The resulting

“

We believe that these two changes remove possible ambiguity regarding phases accumulated by the symmetry operations (\hat{C} , \hat{t} , \hat{R} and $\hat{\zeta}$).

- I am unconvinced by the claims, at the end of the paragraph below equation (4), that even and odd harmonics scale linearly and quadratically, which are presented as categorical and unequivocal. To me, what this says is that the range of values of λ which is used for this calculation is too restricted. Given the structure of the problem, there is guaranteed to be a point in which λ is big enough that the pure linear/quadratic scalings are no longer operative, and the behaviour gives way to generic odd/even behaviour, starting with the inclusion of cubic/quartic terms. (Indeed, this is baked into the manuscript’s assertion that the selection rules under consideration are independent of perturbation theory.) I think the

manuscript's point would be carried across much more clearly if the range of λ in use here were expanded enough for that point to be reached.

Thank you for this valuable comment. We have updated figure 2, so that the range of λ is now $-0.75 < \lambda < 0.75$. Within this range, one already sees the effect of the cubic/quartic contributions and they are marked on top of the figure.

Additionally, we have changed the text to describe the situation more accurately. Instead of a claim for categorical linear or quadratic scaling, we explain that the lowest order is linear or quadratic, and that this is represented by the numerical fits:

“

Fitting even (odd) harmonic amplitudes to a linear (quadratic) λ -dependence, results in good agreement within the range $|\lambda| \leq 0.2$ (green shaded region; see individual and average R^2 values in Figure 2.(c,d)). This is in accordance with the analytical predictions of odd/even scaling with λ .

“

3. I am unconvinced by the authors' response regarding the use of the $\omega : 1.95\omega$ frequency ratio in the theoretical description of their experiment with bielliptical fields, which I initially raised as comment 12 of my first report.

In the following, we address your claims address point by point.

Our main claims in this context are:

- i. Using an $\omega : 1.95\omega$ frequency ratio in the theoretical analysis is justified, considering that the experiment employs an $\omega : 1.95\omega$ frequency ratio.
- ii. Our theoretical analysis is perfectly consistent with our experimental observation.
- iii. Our theory cannot be reduced to “sub-channel theory” (i.e., it cannot be derived directly from conservation laws). It provides many new rules that cannot be derived from conservation laws. This is known in the context of standard dynamical symmetries (Neufeld, Podolsky and Cohen, 2019, *Nature Communications*), and it is also true here, in the context of real-synthetic dynamical symmetries.
- iv. *When our theory overlaps with sub-channel analyses, they are consistent (and we have added an SI section that shows this).*
 - a.** As a first observation: there is a key missing piece of information regarding the experimental configuration, namely, the duration of the pulses used in the experiment. As a reader, the closest one can get is by following the manuscript's reference 29 (which is indicated as essentially the same configuration), where the duration of the initial pump laser is given as 27fs.

Firstly, we have included the pulse duration in the revised manuscript – it is 40fs FWHM for the intensity of the pulse, or 68fs full width for $1/e^2$ intensity (i.e. $1/e$ for the field).

For reference, one ω period is 2.66fs and one 1.95ω cycle is 1.36fs. The FWHM (or $1/e^2$) duration of the pulse includes 15 (or 25.5) ω cycles and ~ 29.5 (or ~ 50) cycles of the 1.95ω drive.

This amounts to 10 cycles of the fundamental beam, which has some severe implications: the “primed” dynamical symmetry (reflection on the y axis coupled with a time translation by 10T) does not have enough time to establish itself, and indeed translating in time by 10T does not give the same system – it takes the configuration

completely out of the duration of the pulse. This is in stark contrast with the authors' assertion (in their reply to my initial comment 12) that the reflection symmetry would be retained but the rotation symmetry would break.

Firstly, we would like to emphasize that the pulse duration is considerably longer than 10 cycles, such that pulse shape imperfections are not so severe (as clearly indicated by the experimental measurement fitting with the theory). Additionally, we wish to note that contrary to the referee's comment, we in fact did not assert the "breaking" of any of the results, all we asserted was that comparing an $\omega - 2\omega$ analysis to the experiment is an irrelevant test to the theory, as the experiment employs an $\omega - 1.95\omega$ frequency ratio.

Going by the **FWHM** (or $1/e^2$) definition for the pulse duration, the "primed" reflection symmetry directly preserves 5 (or 15) fundamental ω cycles. That is, approximately 10~30 recombination events have a counterpart recombination event 10T later, with good phase relations, so that they interfere in a way that is consistent with our theory. Additionally, in the experiment, the carrier envelope phase (CEP) is not stabilized, i.e., it 'slips' from pulse to pulse. As a result, our measurement represents the average of many light-matter interactions with different CEPs, diminishing the effect of the pulse envelope. For the rotational symmetry this is not an issue as the temporal displacement is sub-cycle.

The revised manuscript reads:

"In our set-up (Kfir *et al.*, 2016, *Applied Physics Letters*) (Fig. 4(a)), a bi-chromatic 40fs laser pulse (FWHM) with frequencies $\omega - 1.95\omega$ (corresponding to the wavelengths 800nm and 410nm, respectively)"

- b.** Generally, I think the core of the issue concerns the resolution of the experimentally observed spectrum (figure 4b) into individual subchannels as done in the original 2014 experiment. For the claimed frequency ratio, the allowed channels for harmonic ~ 19 split into a "base" channel at $18.7\omega = 7\omega + 6 \times 1.95\omega$ (allowed at full symmetry) together with the perturbation-driven subchannel at harmonic frequencies $18.75\omega = 9\omega + 5 \times 1.95\omega$ and $18.8\omega = 11\omega + 4 \times 1.95\omega$ (and so on) forbidden at full symmetry, with the HO18.75 subchannel possibly forbidden depending on the configuration. If the SH detuning from 2ω to 1.95ω is experimentally important, then one would expect to see all of the allowed subchannels present and clearly resolved in the measured spectra. However, I do not think this is the case: the presented spectra do show noticeable shifts in frequency depending on the value of θ , but these are not enough to clearly separate into individual subchannels.

We disagree with the referee on this point. The fact that the frequency detuning is important for our experiment is supported by quantitative evidence, laid out in the paper.

The quantitative evidence that the frequency detuning is important and that the channels are separated is

- (1) the experimental observations are consistent with an emission-channel analysis that relies on the fact that the channels are separated and detuning is important (revised SI, section IV) .
- (2) the experimental observations are consistent with an $\omega - 1.95\omega$ dynamical symmetry analysis that relies on the fact that detuning is important.
- (3) the experimental observations are **NOT** consistent with an $\omega - 2\omega$ dynamical symmetry analysis (and they should not be). This is laid out in the previous reply, as well as in this reply, section 3.c.
- (4) When both emission channel & dynamical symmetry analyses are applicable, they are consistent with one another (revised SI section IV).

Additionally, we note that in the previous version of the MS, we have shown 6 different predictions to be consistent with the $\omega - 1.95\omega$ analysis (3 spectral component, 2 predictions per component). In the revised SI, we have added 3 more spectral components, and so overall we now observe the consistency of 12 separate predictions with the experiment.

Considering this, the experimental observation shows unequivocally the spectral separation is important, even though the colormap may not look like the channels are perfectly separated. To us, it seems that the effect of the spectral separation is visually apparent in the spectrum (“noticeable spectral shifts”). But more significantly, 12 separate predictions are shown to be consistent with the observation.

Please also appreciate that it is experimentally challenging to obtain perfect separation of the channels which “looks” just like a TDSE or SFA simulation, but this does not mean that the channels are not separated or that the experiment is equivalent to an $\omega - 2\omega$ experiment.

- c.** This leads me to the authors’ assertion, in their reply to the initial comments, that if the simpler frequency ratio were used, the results in the paper would break, and the selection rules would change.

We emphasize that we do not claim that the predictions “break” in any way. All we claim is that comparing the $\omega - 2\omega$ theory to the $\omega - 1.95\omega$ observation is inconsistent and wrong. Further, **there is no reason to do so** because we have formulated a precise analytical theory that compares consistently with the observation.

The $\omega - 2\omega$ analysis would just be inconsistent with the observations because the experiment employs an $\omega - 1.95\omega$ frequency ratio. So, in any case the results in the paper would not ‘break’, as it would not be a relevant test to the theory if it is applied to the incorrect experimental set-up.

I am unable to follow the authors’ calculations in their reply, mostly because I do not understand the action of the DS as I mentioned in comment 1 above; I also do not understand the meaning of the multiple \pm signs used.

Firstly, we have already dedicated a section in the SI to derive the experimentally observed $\omega - 1.95\omega$ selection rules step by step. To further clarify these calculations, we have added the following clarifying sentences in section III of the SI, after equation III.3:

“

The action of $\hat{R}_{2.95}$ on the vector $\mathbf{E}_n^{(abcd)}$ can be obtained by representing it as a sum circularly polarized vectors, $\mathbf{E}_n^{(abcd)} = E_{Rn}^{(abcd)}(\hat{x} - i\hat{y}) + E_{Ln}^{(abcd)}(\hat{x} + i\hat{y})$ where \hat{x}, \hat{y} are Cartesian basis vectors. The vectors $\hat{x} \mp i\hat{y}$ are eigenvectors of the rotation operator $\hat{R}_{2.95}$ that exhibit $\hat{R}_{2.95}(\hat{x} \mp i\hat{y}) = e^{\mp 2\pi i/2.95}(\hat{x} \mp i\hat{y})$. Hence, the right-handed ($\hat{x} - i\hat{y}$) and left-handed ($\hat{x} + i\hat{y}$) polarization components of harmonic n satisfy...

“

The $\omega - 1.95\omega$ analysis is carried out in section S.III., and the results are presented in the main text together with the consistent experimental observation.

Now, let us repeat this calculation, step by step, for the $\omega - 2\omega$ configuration. We emphasize that this calculation does not represent our measurement, and it shouldn't. For an $\omega - 2\omega$ field, we have:

$$\mathbf{E}(t) = \Re \frac{i}{2} \left\{ \underbrace{\eta(\mathbf{r}^+ e^{2i\omega t} + \mathbf{r}^- e^{i\omega t})}_{\hat{C}_3 \text{ symmetric}} - \delta_1 \mathbf{r}^+ e^{i\omega t} + \delta_2 \mathbf{r}^- e^{2\omega t} \right\}$$

where $\mathbf{r}^\pm = \hat{x} \pm i\hat{y}$ where \hat{x} and \hat{y} are Cartesian basis vectors. This field exhibits the symmetry $\hat{C}_3 \cdot \hat{\zeta}$ where $\hat{\zeta}(\delta_1, \delta_2) = (\delta_1 e^{2\pi i/3}, \delta_2 e^{-2\pi i/3})$. As a result, the HHG emission adheres to

$$\mathbf{E}_{HHG}(t, \delta_1, \delta_2) = \hat{R}_3 \mathbf{E}_{HHG}(t + T/3, \delta_1 e^{2\pi i/3}, \delta_2 e^{-2\pi i/3})$$

By expanding $\mathbf{E}_{HHG}(t, \delta_1, \delta_2)$ to a power series in $\delta_1^a \delta_2^b$ we have

$$\mathbf{E}_{HHG}(t, \delta_1, \delta_2) = \sum_n \mathbf{E}_n^{(abcd)} e^{in\omega t} \delta_1^a \delta_2^b \delta_1^{-c} \delta_2^{-d}$$

We obtain

$$\begin{aligned} \hat{R}_3 \mathbf{E}_{HHG}(t + T/3, \delta_1 e^{2\pi i/3}, \delta_2 e^{-2\pi i/3}) \\ = \sum_n e^{2\pi i(a-b-c+d)/3} \hat{R}_3 \mathbf{E}_n^{(abcd)} e^{2\pi in/3} e^{in\omega t} \delta_1^a \delta_2^b \delta_1^{-c} \delta_2^{-d} \end{aligned}$$

The action of \hat{R}_3 on the vector $\mathbf{E}_n^{(abcd)}$ can be obtained by representing it in a circularly polarized basis, $\mathbf{E}_n^{(abcd)} = E_{n-}^{(abcd)} \mathbf{r}^- + E_{n+}^{(abcd)} \mathbf{r}^+$ where $\mathbf{r}^\mp = \hat{x} \mp i\hat{y}$. The vectors \mathbf{r}^\mp are eigenvectors of the rotation operator \hat{R}_3 that exhibit $\hat{R}_3 \mathbf{r}^\mp = e^{\mp 2\pi i/3} \mathbf{r}^\mp$. Hence, the right-handed ($\hat{x} - i\hat{y}$, denoted by a minus subscript) and left-handed ($\hat{x} + i\hat{y}$, denoted by a plus subscript) polarization components of harmonic n satisfy:

$$E_{n\mp}^{(abcd)} = e^{2\pi i \times [n + (a-b-c+d)\mp 1]/3} E_{n\mp}^{(abcd)}$$

Hence, $E_{n\pm}^{(abcd)} = 0$ unless there exists an integer q so that

$$n + (a - b - c + d) \pm 1 = 3q$$

For linear contributions, only one of a, b, c, d is non-zero and equal to one. Hence, linear contributions are characterized by $a - b - c + d = \pm 1$. Considering this, the condition for a particular harmonic order n to exhibit a linear contribution, is that there exists an integer q so that

$$n \pm 1 \pm 1 = 3q$$

Explicitly, a linear contribution exists for harmonic n if any of the following 4 conditions is met by an integer q :

$$n + 1 + 1 = 3q$$

$$n + 1 - 1 = 3q$$

$$n - 1 + 1 = 3q$$

$$n - 1 - 1 = 3q$$

For any integer n , there is an integer q that fulfills one of these conditions. Hence, all harmonic orders have linear contributions in their scaling.

For quadratic contributions, either two of a, b, c, d , are equal to 1 and the rest to zero, or, one is equal to 2 and the rest are 0. Hence, for quadratic contributions, $a - b - c + d$ can equate to 0 (e.g. through $a=1, b=1$), 2 (e.g. through $a=1, d=1$), or -2 (e.g. through $b=c=1$). Thus, we plug in $a - b - c + d = \pm(1 \pm 1)$ to represent that it can equate to 0, 2, or -2. The selection rule for quadratic contributions for harmonic n is

$$n \pm (1 \pm 1) \pm 1 = 3q$$

Or explicitly, a quadratic contribution exists for harmonic n if any of the following 6 conditions is met by an integer q :

$$n + 2 + 1 = 3q$$

$$n + 2 - 1 = 3q$$

$$n - 2 + 1 = 3q$$

$$n - 2 - 1 = 3q$$

$$n + 0 + 1 = 3q$$

$$n + 0 - 1 = 3q$$

Clearly, for any integer n there exists an integer q so that one of these conditions is met. Hence, all harmonic orders exhibit quadratic contributions as well.

This result does not describe in any way the experimental observation (and it shouldn't).

Nevertheless, it is very strange to claim that such a small change in how the situation is analysed will bring about such a massive change in the observed selection rules (i.e. the allowed scalings w.r.t. the perturbation for the chosen lines).

We respectfully disagree with this statement. Arguing that changing the frequency of the 2nd drive from 1.95ω to 2ω is a "small change", and therefore deeming the results "strange" is a subjective matter.

In our opinion, whether a parameter change is small or large is determined by the observation, and NOT by our own intuitive perceptions.

In the context of the scaling of a particular frequency component, a frequency change from 1.95ω to 2ω is not a small change at all, precisely because it introduces "such a massive change in the observed selection rule". Depending on the experimental configuration and observables of interest, a 5% change (or 2.5% of the SH field) could be insignificant in one

parameter (e.g., amplitude) and extremely significant in another (e.g., frequency, especially when bi-chromatic drives are employed).

To concretely show how much the difference between $\omega - 2\omega$ and $\omega - 1.95\omega$ is not “small”, we invite you to compare our measured spectra (harmonic order vs. QWP angle), to figure 2.b. and figure 3.d. in “A dynamical symmetry triad in high-harmonic generation revealed by attosecond recollision control”, which performs a similar experiment in an $\omega - 2\omega$ configuration. The spectra and its scaling with the QWP angle are visually different, despite the perception that 0.05ω is a small number. Any change that one can immediately see with naked eyes on the measured spectrum is not small.

The authors claim that the allowed scaling’s would “change” from cubic/quartic to linear/quadratic, and I find this implausible.

The scaling law of a particular frequency component of the emission with the QWP angle depends on the frequencies of the driving field. This is shown clearly and consistently by the theory and the observation. In particular, for an $w-2w$ experiment many subchannels interfere together, which masks the scaling of each channel individually. This is exactly what is revealed in the $w-1.95w$ experiment (as was shown e.g. for SAM conservation). Thus, a small frequency change can indeed change the appear scaling, because it reveals different channel contributions. Whether such change is plausible or implausible, is a matter of opinion and perspective, but regardless, that is irrelevant because that change is observed experimentally and exactly derived in our theory. We kindly ask that if the referee has further comments on this topic, that he use physical considerations supported by equations, rather than unsupported opinions.

Instead, it is much more likely that each line consists of multiple subchannels (as laid out above) and that each line contains one subchannel with linear/quadratic scaling, and that the authors are interested in additional subchannels which are detuned from that simpler linear/quadratic one. If this is the case, then it should be explained in depth, which implies adding a subsection in the SI exploring in detail the scaling of all the different subchannels of at least one chosen line of interest.

We do not refute that some of our predictions and observations can also be understood based on a subchannel analysis (i.e., derived using photonic conservation laws). Particularly, the harmonic scaling around 45 degrees in our experiment can be explained using a subchannel analysis, as you point out (and this shows consistency of the theory). Following the referee’s comments, we have added a section in the SI that shows this consistency (section IV). Fundamentally, this is because in **standard** Floquet group theory, $\hat{C}_{N,M}$ selection rules are equivalent to conservation of spin angular momentum. **It is not directly related to our work here on synthetic dimensions, but rather, it represents a general parallelism between dynamical symmetries and photonic conservation laws.** This parallelism was already discussed in our previous work (Neufeld, Podolsky and Cohen, 2019, *Nature Communications*):

“

Lastly, we discuss the link between DSs and selection rules to conservation laws. Although Noether’s theorem does not connect discrete DSs to conservation laws, all previous selection rules in HG were also derived from conservation laws: the appearance of only discrete harmonics in the spectrum (the selection rule due to time-periodicity ($\hat{\tau}_1$)) can be

derived from energy conservation, the \hat{C}_2 selection rule can be derived from parity conservation, and the \hat{C}_N selection rule can be derived from conservation of spin angular momentum of the interacting photons. This duality is believed to reflect an equivalence between the DS and photonic pictures in HG. Interestingly, we find that several of the DSs ($\hat{e}_{N,M}, \hat{P}_{N,M}, \hat{Z}, \hat{T}, \hat{Q}, \hat{D}$ and \hat{J}) lead to selection rules that have no analogue conservation law derivation. What does this result mean? It could indicate that the DS perspective is more general than the photonic perspective. Can conservation laws associated with the above DSs save the disparity between the two approaches?

“

While \hat{C}_{NM} (or $\hat{C}_{NM} \cdot \hat{\zeta}$) selection rules do have equivalent conservation law derivations, \hat{Z} (or $\hat{Z} \cdot \hat{\zeta}$) selection rules do not. Hence, the subchannel analysis you suggest as an alternative explanation to our observations is legitimate for the scaling around $\theta = 45^\circ$, but is not applicable for the scaling around $\theta = 0^\circ$, corresponding to $\hat{Z} \cdot \hat{\zeta}$ symmetry. Additionally, subchannel analysis (i.e., conservation law analysis) cannot directly explain other predictions obtained by our theory, i.e., selection rules for $\hat{T} \cdot \hat{\zeta}$, $\hat{Q} \cdot \hat{\zeta}$, $\hat{D} \cdot \hat{\zeta}$, and $\hat{H} \cdot \hat{\zeta}$. Instead, the physics that these rules reflect is the deep interplay between conservation laws and phase-sensitive interference effects.

Hence, the contrasting phrase “**Instead**, it is much more likely...” is inappropriate. It is not one or the other; dynamical symmetries and conservation laws are complementary concepts, which sometimes overlap. DSs cannot be replaced altogether with conservation laws, and this was already acknowledged in the original Floquet group theory paper (Neufeld, Podolsky and Cohen, 2019, *Nature Communications*). This is particularly true for our experiment. A subchannel analysis only overlaps with a small subset the results. Our observations and theory cannot be directly reduced to “sub-channel theory” (i.e., photonic conservation laws).

Following this discussion, we have added the following paragraph to the revised manuscript:

“

Finally, we note that the observed scaling of the harmonic amplitudes around $\theta = 45^\circ$ can also be obtained via emission-channel analysis that relies on conservation of energy, parity, and spin, (SI, section IV). In contrast, the scaling around $\theta = 0^\circ$ does not have an analogue conservation law derivation.

“

Additionally, we have added a new section in the revised SI (section IV) that shows explicitly that our approach is consistent with an emission channel analysis. This section also includes the analysis of three additional spectral components, for an overall of 6 spectral components. All 6 analyzed spectral components are analyzed with subchannel theory, real-synthetic symmetries, and are compared to the experimental observation.

- d.** As a minor, related question: at the end of p. 7, where the integration ranges are specified as $n\omega \pm 0.0225\omega$, is n an integer (as the notation would lead one to assume), or are we meant to infer $n = 18.8$ (etc.)?

Yes, indeed $n=18.8$. This sentence reads:

“

Figures 4(a-c) show the measured harmonic amplitudes of spectral components $n=18.8$, 19.75 and 20.7 as a function of the waveplate angle, θ , obtained by integrating the measured signal (Fig. 3(c)) in a range of $n\omega \pm 0.0225\omega$ and taking the square root.

“

4. I feel that the Lissajous figures reported in figure 3 do not accurately represent the fields in use. The Lissajous figures currently depicted correspond to the non-detuned frequency ratio $\omega:2\omega$, but if indeed the redshifting of the SH is essential, as the authors claim, then the Lissajous figures change to look something like the following, with the relative phase of the two fields slipping slightly on each period. I appreciate the need for simplicity, but this needs to be reported correctly (i.e. it should be at least indicated in the text that the Lissajous figures as shown consist only of a cut lasting a single period; I would still recommend in that case to include depictions of the full figures in the SI).

The Lissajous figures reported in Figure 3 are calculated using an $\omega - 1.95\omega$ frequency ratio, however, they only show a single ω cycle, as you have stated. This is the reason that the Lissajous figures do not “close” after the cycle (there is a small gap) – they will only close after 20 cycles of the fundamental ω frequency. We state this explicitly in the revised manuscript:

“

We note that Figure 3(b) depicts the Lissajous curves in the temporal window between 0 and T_ω , while the periodicity of the bi-chromatic field is $T = 20T_\omega$ (the complete Lissajous are given in the SI, section V).

“

Revised caption of Figure 3:

“

Figure 3: Real-synthetic symmetries in bi-elliptical HHG. (a) illustration of the experimental setup²⁹ (b) Lissajous curves of the driving field for different values of the QWP angle θ , in the temporal window $0 < t < T_\omega$. The phase of each Lissajous is illustrated by a color gradient...

“

Additionally, we have added a section in the SI that depicts the full Lissajous curves of the driving field (Section V)

Minor comments

5. The reference block [27,28] used for HHG is not particularly informative for readers who are unfamiliar with HHG. I would recommend adding a generic review here; a good example is Contemp. Phys. 59, p. 47 (2018).
Thank you for this comment. We have added Contemp. Phys. 59, p. 47 (2018) to this citation block.
6. In the first paragraph of the Results section, “broken in the systems in the left panel” – shouldn’t this be the right panel?

We have corrected it from left to right.

7. Same paragraph: in “the amplitude and phase of the applied voltage”, the voltage is DC, so it has a sign but it does not have a phase

We have corrected this. The revised manuscript just reads “the applied voltage”.

8. I find the passage between equations (3) and (4) to be quite violent; for many readers, this will be the point at which they stop understanding the details of the manuscript. Within this introductory-example section, this type of manipulation should be explained in more detail.

We have added the following paragraph between equations 3 and 4:

“

To reformulate this selection rule as a selection rule for harmonic amplitudes, we employ a Fourier transform $\mathbf{E}_{HHG}(t, \lambda) = \sum_n \mathbf{E}_n(\lambda) e^{in\omega t}$ where $\mathbf{E}_n(\lambda)$ is the complex amplitude of the n 'th harmonic. The operation \hat{R}_2 transforms a vector $\mathbf{E}_n(\lambda)$ as $\hat{R}_2 \mathbf{E}_n(\lambda) = -\mathbf{E}_n(\lambda)$, and $\hat{t}_2 e^{in\omega t} = e^{in\omega(t+T/2)} = (-1)^n e^{in\omega t}$. Consequently, the $\hat{C}_2 \cdot \hat{\zeta}$ selection rule for harmonic amplitude n is:

“

9. The order of introduction of figures in the text is somewhat jolting – fig. 4 is introduced before fig. 3. I think the order of the figures is largely fine, but they should be introduced in correct order inside the text. As it is, the discrepancy is distracting and this detracts from smooth reading of the manuscript.

Thank you. We have significantly rearranged the text as well as the figures and their content in this section.

The figure that depicts the experimental setup is now figure 3. This figure also includes the Lissajous of the driving field for different values of ϵ (previously figure 3.(a)), the real-synthetic symmetry operations, and the analytical predictions. The experimental observations and their comparison to the fitting models are shown in figure 4.

10. In the expression for the bielliptical field (eq. (5) and onwards), why do the exponentials use negative frequencies (i.e. $e^{-i\omega t}$ instead of $e^{i\omega t}$)? This breaks the conventions of the field and makes the material harder to work with.

This sign convention is employed to stay consistent with the sign conventions employed in the paper “Floquet group theory and its application to selection rules in harmonic generation” (Neufeld, Podolsky and Cohen, 2019, *Nature Communications*). We feel that this is the natural convention to follow, considering that our manuscript heavily utilizes Floquet group theory.

11. For the bielliptical experiment, the dependence of the ellipticity on the waveplate angle, $\epsilon(\theta)$, needs to be given explicitly.

Thank you. We have added this dependence in the SI, section V. The main text refers the reader to the corresponding section just before equation 5:

“

The rotation angle θ of the QWP controls the ellipticity $\epsilon(\theta)$ of the pumps (SI section V), resulting in the following field:...

“

12. The handling of the bielliptical perturbation parameters, $\delta_{1,2}$, is confusing. As initially presented, they are real, but then they are given complex phases. Which is it?
 $\delta_{1,2}$ are complex amplitudes of a general field given by equation 6. This general field exhibits real-synthetic symmetries that result in selection rules. These selection rules are observed in the experiment by varying $\delta_{1,2}$ on the real axis.
 In the revised manuscript, we have explicitly stated that $\delta_{1,2}$ are complex amplitude, to avoid confusion:
 “
 The parameters $\delta_{1,2}$ are the complex amplitudes of two circularly polarized symmetry breaking fields which result in selection rule deviations as the symmetry is broken.
 “
13. There is a missing imaginary unit in the final exponent in equation (6).
 Thank you. This is now corrected.
14. In the specification $\mathbf{r} \pm = \hat{x} \pm i \hat{y}$, after equation (6), what do the hats actually mean? I find the mixture of notation quite confusing. Wouldn't it be better to use uniform and unambiguous notation for basis vectors, such as e.g. \hat{e}_x , \hat{e}_y and \hat{e}_{\pm} ?
 We have chosen the \hat{x} and \hat{y} notation to remain consistent with (Neufeld, Podolsky and Cohen, 2019, *Nature Communications*), which our manuscript heavily relies on. We have eliminated all uses of \mathbf{r}^{\pm} in the SI, and instead use either explicitly $\hat{x} \pm i\hat{y}$, or \hat{e}_R and \hat{e}_L . Consider inserting a paragraph break before “As θ is detunes”, at the bottom of p. 6.
 We have added a paragraph break.
15. In p. 9, the text reads “In the examples above, \mathbf{Q} is a vector...” before \mathbf{Q} has appeared at all.
 Thank you. The revised manuscript defines \mathbf{Q} as a vector that contains the parameters of the perturbation:
 “
 \hat{W} breaks the symmetry \hat{X} such that $\hat{X}^\dagger \hat{H}_0 \hat{X} = \hat{H}_0$, but $\hat{X}^\dagger \hat{H} \hat{X} \neq \hat{H}$. Although \hat{X} is broken, it may still be exploited to formulate a symmetry of the form $\hat{X} \cdot \hat{\zeta}_{\hat{X}}$ in the symmetry broken system, where $\hat{\zeta}_{\hat{X}}$ operates on the internal degrees of freedom of \hat{W} denoted by the vector \mathbf{Q} , while leaving \hat{H}_0 unaffected.
 “
16. Some formatting issues: X in eq (9) and n under eq. (10) should not be bold. There is also a left-over summation over k,l,h,j in equation (10).
 Thank you. This is corrected in the revised manuscript.
17. 17. In figure 1, right-hand panel, the “bichromatic” pulse is depicted in red and blue, but it is not bichromatic (both components are at the same frequency).
 This is corrected in the revised manuscript.
18. In figures 2 and 3, the colour schemes used for the Lissajous figure need to be explained in the caption.
 Thank you, the captions of the revised MS read:
 “
The phase of each Lissajous is illustrated by a color gradient which transforms under temporal translations.
 “
19. In figure 2, for the second and third rows, it feels to me like the symmetry breaking would be clearer if a smaller value of λ were used.
 We have changed the value of λ to be smaller.

20. In figure 2, above the simulation results, the scheme claims the configuration in use contains “randomly oriented molecules”, instead of spherically-symmetric monatomic gas atoms. This is now corrected in the revised figure. The revised figure illustrates spherically symmetric atoms.
21. In table I, the correspondence between rows on both columns is unclear. Presumably each row on the right corresponds to multiple rows on the left, at least at the top, but the which-to-which correspondence is highly unclear, making the table essentially useless. We have split cells in table 1 so there is a one-to-one correspondence between the columns.
22. Note missing publication details on reference 22. This is now corrected.
23. In §II of the SI, the symmetries are introduced very sharply. While some of the newer symmetries are quite complex, I would argue that if there are clear, simple examples of field configuration with that symmetry, it would be good to mention them briefly, to provide a mental anchor which can be used to understand the results.

The symmetries and representative field examples were the focus of a separate paper (Neufeld, Podolsky and Cohen, 2019, *Nature Communications*). This paper extensively discussed Floquet group symmetries and presented exemplary fields for all DSs in its main text. In the revised SI, we refer the reader to (Neufeld, Podolsky and Cohen, 2019, *Nature Communications*) at the beginning of §II of the SI. Additionally, we provide an exemplary field for each symmetry and plot its Lissajous curve.

The revised SI reads:

“

The operations \$\hat{X}\$ were systematically tabulated in ref¹, which also provides explicit examples for \$\hat{X}\$ -symmetric Floquet systems for all DSs in \$(2 + 1)D\$ and \$(3 + 1)D\$. Figure SII.1 illustrates the Lissajous curves of these exemplary fields for all \$(2+1)D\$ Floquet group symmetries.

“

24. The supplementary information requires a round of careful copy-editing. Some issues I caught:
- In eq. (II.5), a leftover (p), an unduly bold n, and a leftover summation over k,l,h,j
This is corrected in the revised SI.
 - Throughout the SI, incorrectly upright symbols that should be italicized
This is corrected in the revised SI.
 - Missing superscripts in (II.2) and indices in the summation symbols in (III.2) and (III.3).
This is corrected in the revised SI.
 - Leftover (p) in (IV.7).
This is corrected in the revised SI.

Additionally, we have carefully proof-read the SI and corrected all the errors we spotted.

25. In eq. (II.50), I don't feel the use of column-vector notation for the circular basis is appropriate. As written it is unambiguous if one reads the text in detail, but the potential for confusion is too high. An explicit basis summation would be highly preferable. We have changed the notation to an explicit basis summation.

26. 26. Relatedly, are circular-basis indices L/R or +/-? Eq. (II.50) uses the former, while (II.58) uses the latter.

R/L vectors are right/left handed circularly polarized components, i.e., $\hat{x} \mp i\hat{y}$. The +/- vectors in Eq.(II.58) are right/left handed elliptically polarized vectors with ellipticity b , i.e., $\hat{x} \mp i\hat{y}$. This should be clear in the revised SI.

27. 27. For eq. (II.55), as an alternative to the notation $\exists z \in \mathbb{Z}: (\text{quantity}) = Nz$, consider using the equivalent modulo notation, $(\text{quantity}) \equiv 0 \pmod{N}$.

We have changed the notation to modulo notation in eq.(II.55) and the following sections.

I have also been asked to assess the authors' response to the concerns of Referee 1. I am not really able to say whether Referee 1 would be satisfied by the response to the preamble, since I disagree with Referee 1 on the value provided by the manuscript. It is probably true that, as Referee 1 suspects, "many of these Taylor series showing various missing powers of theta could be derived from direct Taylor series of the time-dependent fields without any reference to this symmetry argument", but that misses the point – even if such a derivation can be made on a case-by-case basis, there is still significant value in the general theory. Regarding Referee 1's detailed comments: 1. This seems to have been addressed fully. 2. This seems to have been addressed fully. 3. This seems to have been addressed fully. 4. I understand Referee 1's point regarding the terminology, but this is ultimately a subjective matter. I feel that the authors' reply lays out a valid justification for their use of the term.

REVIEWER COMMENTS

Reviewer #2 (Remarks to the Author):

To clarify, what I meant by reference field, is the field with period $T_{\{\omega\}} = 2\pi/\omega$ in the experimental section, which is not the period of the Floquet system, but simply a period that used as a unit of time.

Regarding the meaning of ω in equation 10, if in that equation ω is indeed the fundamental frequency of the perturbed Floquet system (which makes sense), shouldn't a different symbol be used to distinguish it from the ω of the experimental section? Perhaps use a capital Ω or ω_F , to make it clear that the ω of the experimental section is not the Floquet period (alternatively, always use ω for the frequency of the Floquet system, and use ω_0 for field with wavelength of 800nm in the experimental case). I apologize for being nitpicky regarding the notation used for the different frequencies, but lack of clarity in this really hinders the understanding of the paper, as the reader has to do some guess work.

A few remaining issues:

1. In Eq. 8 of the main text, when the perturbation W is discussed, it is written " s is a rational number". However, in the Supplementary Material when discussing Eq. II.1 it is written " s is an integer". I believe this should be corrected to " s in a rational number".
2. In the "Results" section of page 3 where it reads " σ_i is a spatial reflection relative to \hat{i} ", change it to: " σ_i is a spatial reflection relative to the i coordinate ($i = x, y, z$)"
3. In pag 7, where it reads "As θ is detunes from 45° ", it should probably be written "As θ is detuned from 45° "
4. In page 12, in Eq. 11, should s_1 and s_2 be commensurate with each other and the fundamental frequency of the unperturbed system? (in order for the system with the bi-chromatic perturbation to still be time-periodic)

I believe that the "Simple example" section really helps in the presentation of the general formalism. Overall, I think that by solving the simple issues I pointed out, the paper will be in a form suitable to be published in Nature Communications.

Reviewer #3 (Remarks to the Author):

The authors have responded to all my concerns. I recommend the manuscript for publication.

As a final recommendation, I would suggest that the parametric plots of Lissajous figures in Figure SII.1 should be plotted using aspect ratios that respect the natural lengths of the plots (i.e. such that the physical length on the page of 1 unit along the x axis is equal to the physical length of 1 unit on the y axis). This is essential for the direct visual interpretation of the image to correctly match the symmetry properties of the Lissajous figure depicted. As currently presented, none of panels (g), (h) or (i) appear symmetric, but if correctly presented (g) and (h) should look rotationally symmetric but (i) should not.

Response to reviewer #2

We thank the referee for reviewing our manuscript. The reviews assisted us to significantly improve our manuscript. Thank you. Below, we respond to the referee's comments. Our responses are written in blue

To clarify, what I meant by reference field, is the field with period $T_{\{\omega\}} = 2\pi/\omega$ in the experimental section, which is not the period of the Floquet system, but simply a period that used as a unit of time.

Regarding the meaning of ω in equation 10, if in that equation ω is indeed the fundamental frequency of the perturbed Floquet system (which makes sense), shouldn't a different symbol be used to distinguish it from the ω of the experimental section? Perhaps use a capital Ω or ω_F , to make it clear that the ω of the experimental section is not the Floquet period (alternatively, always use ω for the frequency of the Floquet system, and use ω_0 for field with wavelength of 800nm in the experimental case). I apologize for being nitpicky regarding the notation used for the different frequencies, but lack of clarity in this really hinders the understanding of the paper, as the reader has to do some guess work.

Thank you for this useful suggestion. Indeed ω in eq.10 is the frequency of the perturbed system. This is clarified in the revised version of the manuscript. In the experimental section, the frequency of the 800nm field is now denoted by ω_0 as you suggested. The notation was also updated in the figures and in the SI, so that whenever the experiment is discussed, the frequency of the 800nm beam is denoted by ω_0 and not ω .

Additionally, we have further emphasized in the main text the distinction between ω_0 , the reference field, and ω , the Floquet frequency. The revised manuscript reads:

“

We note that Figure 3(b) depicts the Lissajous curves in the temporal window between 0 and T_{ω_0} , while the periodicity of the bi-chromatic field is $T = 20T_{\omega_0}$ (the complete Lissajous are given in the SI, section V). **We further emphasize that the Floquet frequency of this system is given by $\omega = 2\pi/T$ and not $\omega_0 = 2\pi/T_{\omega_0}$, and it is the Floquet frequency ω that should be used when applying the general theory outlined in the next section.**

“

A few remaining issues:

1. In Eq. 8 of the main text, when the perturbation W is discussed, it is written "s is a rational number". However, in the Supplementary Material when discussing Eq. II.1 it is written "s is an integer". I believe this should be corrected to "s in a rational number".

Thank you. We have corrected the discussion that follows Eq.II.1 to "s is a rational number".

2. In the "Results" section of page 3 where it reads " σ_i is a spatial reflection relative to \hat{i} ", change it to: " σ_i is a spatial reflection relative to the i coordinate (i = x, y, z)"

We have clarified this sentence and changed the sentence to “and $\hat{\sigma}_i$ is a spatial reflection relative to \hat{i} axis (so that $\hat{\sigma}_x$ is the operation $y \rightarrow -y$)¹⁰”

3. In pag 7, where it reads "As θ is detunes from 45° ", it should probably be written "As θ is detuned from 45° "

We have corrected this sentence to “As θ detunes from $45^\circ \dots$ ”.

4. In page 12, in Eq. 11, should s_1 and s_2 be commensurate with each other and the fundamental frequency of the unperturbed system? (in order for the system with the bi-chromatic perturbation to still be time-periodic)

Yes, $s_1\omega$ and $s_2\omega$ should be commensurate with each other and with ω . This is stated in the revised MS following equation 11:

“

where $s_{1,2}$ determine the color of each perturbation, and $Q_{1,2}$ are their complex amplitudes, and $s_1\omega, s_2\omega$ and ω are mutually commensurate frequencies

“

I believe that the "Simple example" section really helps in the presentation of the general formalism. Overall, I think that by solving the simple issues I pointed out, the paper will be in a form suitable to be published in Nature Communications.

Reviewer #3 (Remarks to the Author):

We thank the referee for reviewing our manuscript and for recommending publication. The reviews and comments assisted us to rebuild and improve our manuscript. Thank you. Below, we respond to the referee's comments. Our responses are written in blue

The authors have responded to all my concerns. I recommend the manuscript for publication.

As a final recommendation, I would suggest that the parametric plots of Lissajous figures in Figure SII.1 should be plotted using aspect ratios that respect the natural lengths of the plots (i.e. such that the physical length on the page of 1 unit along the x axis is equal to the physical length of 1 unit on the y axis). This is essential for the direct visual interpretation of the image to correctly match the symmetry properties of the Lissajous figure depicted. As currently presented, none of panels (g), (h) or (i) appear symmetric, but if correctly presented (g) and (h) should look rotationally symmetric but (i) should not.

Thank you for pointing this out. The aspect ratio of the Lissajous figures is corrected in the revised SI.